# A framework for community curation of interspecies interactions literature

Alayne Cuzick[1]*, James Seager[1], Valerie Wood[2], Martin Urban[1], Kim Rutherford[2], Kim E Hammond-Kosack[1]*

[1]Strategic area: Protecting Crops and the Environment, Rothamsted Research, Harpenden, United Kingdom; [2]Department of Biochemistry, University of Cambridge, Cambridge, United Kingdom

**Abstract** The quantity and complexity of data being generated and published in biology has increased substantially, but few methods exist for capturing knowledge about phenotypes derived from molecular interactions between diverse groups of species, in such a way that is amenable to data-driven biology and research. To improve access to this knowledge, we have constructed a framework for the curation of the scientific literature studying interspecies interactions, using data curated for the Pathogen–Host Interactions database (PHI-base) as a case study. The framework provides a curation tool, phenotype ontology, and controlled vocabularies to curate pathogen–host interaction data, at the level of the host, pathogen, strain, gene, and genotype. The concept of a multispecies genotype, the 'metagenotype,' is introduced to facilitate capturing changes in the disease-causing abilities of pathogens, and host resistance or susceptibility, observed by gene alterations. We report on this framework and describe PHI-Canto, a community curation tool for use by publication authors.

## Editor's evaluation

Focused on host-pathogen interactions, this valuable study presents a useful resource for unifying language(s) and rules used in biology experiments, with a new ontology and tool called PHI-Canto. The framework enables using UniProtKB IDs to curate proteins and eventually derive 'metagenotypes', an important concept that may incidentally help shrinking proliferating names and acronyms for genes, processes, and interactions. This important framework builds on established standards and methods and was rigorously tested with a variety of publications, providing a system that may eventually capture complex information hidden in the data, such as metagenotypes.

*For correspondence:
alayne.cuzick@rothamsted.ac.uk (AC);
kim.hammond-kosack@rothamsted.ac.uk (KEH-K)

Competing interest: The authors declare that no competing interests exist.

## Introduction

Recent technological advancements across the biological sciences have resulted in an increasing volume of peer-reviewed publications reporting experimental data and conclusions. To increase the value of this highly fragmented knowledge, biocurators manually extract the data from publications and represent it in a standardized and interconnected way following the FAIR (Findable, Accessible, Interoperable, and Reusable) Data Principles (*International Society for Biocuration, 2018*; *Wilkinson et al., 2016*). Curated functional data is then made available in online databases, either organism- or clade-specific (e.g. model organism databases) or those supporting multiple kingdoms of life (e.g. PHI-base *Urban et al., 2022*), Alliance of Genomes Resources (*Agapite et al., 2020*), or UniProt (*Bateman et al., 2021*). Due to the complexity of the biology and the specificity of the curation requirements, manual biocuration is currently the most reliable way to capture information about function and phenotype in databases and knowledge bases (*Wood et al., 2022*). For pathogen–host

**eLife digest** The increasingly vast amount of data being produced in research communities can be difficult to manage, making it challenging for both humans and computers to organise and connect information from different sources. Currently, software tools that allow authors to curate peer-reviewed life science publications are designed solely for single species, or closely related species that do not interact.

Although most research communities are striving to make their data FAIR (Findable, Accessible, Interoperable and Reusable), it is particularly difficult to curate detailed information based on interactions between two or more species (interspecies), such as pathogen-host interactions. As a result, there was a lack of tools to support multi-species interaction databases, leading to a reliance on labour-intensive curation methods.

To address this problem, Cuzick et al. used the Pathogen-Host Interactions database (PHI-base), which curates knowledge from the text, tables and figures published in over 200 journals, as a case study. A framework was developed that could capture the many observable traits (phenotype annotations) for interactions and link them directly to the combination of genotypes involved in those interactions across multiple scales – ranging from microscopic to macroscopic. This demonstrated that it was possible to build a framework of software tools to enable curation of interactions between species in more detail than had been done before.

Cuzick et al. developed an online tool called PHI-Canto that allows any researcher to curate published pathogen-host interactions between almost any known species. An ontology – a collection of concepts and their relations – was created to describe the outcomes of pathogen-host interactions in a standardised way. Additionally, a new concept called the 'metagenotype' was developed which represents the combination of a pathogen and a host genotype and can be easily annotated with the phenotypes arising from each interaction.

The newly curated multi-species FAIR data on pathogen-host interactions will enable researchers in different disciplines to compare and contrast interactions across species and scales. Ultimately, this will assist the development of new approaches to reduce the impact of pathogens on humans, livestock, crops and ecosystems with the aim of decreasing disease while increasing food security and biodiversity. The framework is potentially adoptable by any research community investigating interactions between species and could be adapted to explore other harmful and beneficial interspecies interactions.

interactions, the original publications do not provide details of specific strains, variants, and their associated genotypes and phenotypes, nor the relative impact on pathogenicity and virulence, in a standardized machine-readable format. The expert curator synergizes knowledge from different representations (text, graphs, images) into clearly defined machine-readable syntax. The development of curation tools with clear workflows supporting the use of biological ontologies and controlled vocabularies has standardized curation efforts, reduced ambiguity in annotation, and improved the maintenance of the curated corpus as biological knowledge evolves (*International Society for Biocuration, 2018*).

The pathogen–host interaction research communities are an example of a domain of the biological sciences exhibiting a literature deluge (*Figure 1*). PHI-base (phi-base.org) is an open-access FAIR biological database containing data on bacterial, fungal, and protist genes proven to affect (or not to affect) the outcome of pathogen–host interactions (*Rodríguez-Iglesias et al., 2016*; *Urban et al., 2020*; *Urban et al., 2022*). Since 2005, PHI-base has manually curated phenotype data associated with underlying genome-level changes from peer-reviewed pathogen–host interaction literature. Information is also provided on the target sites of some anti-infective chemistries (*Urban et al., 2020*). Knowledge related to pathogen–host interaction phenotypes is increasingly relevant, as infectious microbes continually threaten global food security, human health across the life course, farmed animal health and wellbeing, tree health, and ecosystem resilience (*Brown et al., 2012*; *Fisher et al., 2018*; *Fisher et al., 2012*; *Fisher et al., 2022*; *Smith et al., 2019*). Rising resistance to antimicrobial compounds, increased globalization, and climate change indicate that infectious microbes will present ever-greater economic and societal threats (*Bebber et al., 2013*; *Chaloner et al., 2021*; *Cook et al., 2021*). In

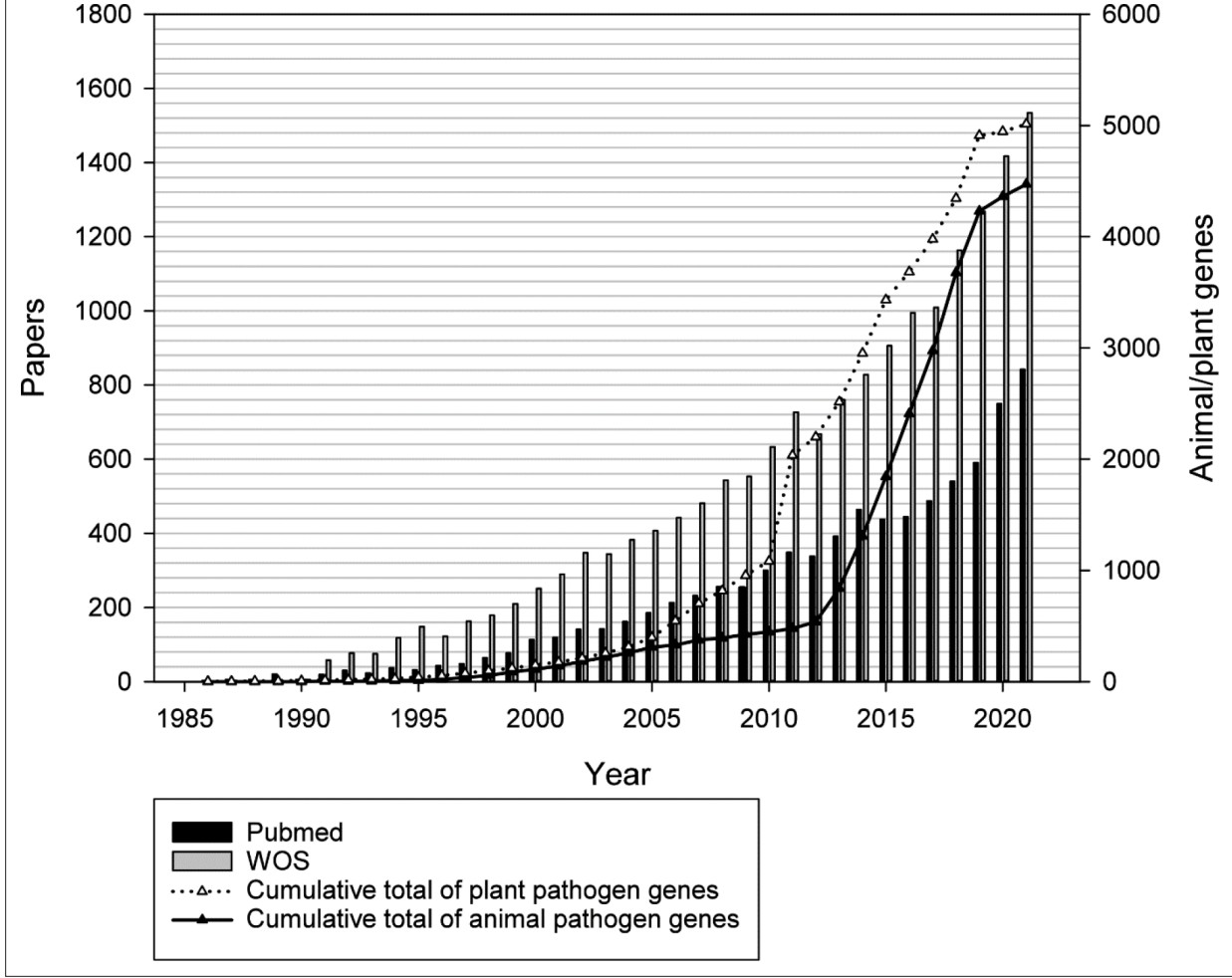

**Figure 1.** Increase of molecular pathogen-host interaction publications and gene-phenotype information during the last 35 years curated in the Pathogen–Host Interactions database (PHI-base). Gray bars show the number of publications in the Web of Science Core Collection database retrieved with search terms '(fung* or yeast) and (gene or factor) and (pathogenicity or virulen* or avirulence gene*).' Black vertical bars show the number of articles retrieved from PubMed (searching on title and abstract). White and black triangles show the number of curated plant and animal pathogen genes, respectively.

order to curate relevant publications into PHI-base (version 4), professional curators have, since 2011, entered 81 different data types into a text file (*Urban et al., 2017*). However, increasing publication numbers and data complexity required more robust curation procedures and greater involvement from publication authors.

We were unable to locate any curation frameworks or tools capable of capturing the interspecies interactions required for PHI-base. PomBase, the fission yeast (*Schizosaccharomyces pombe*) database developed Canto, a web-based tool supporting curation by both professional biocurators and publication authors (*Rutherford et al., 2014*). Canto already had support for annotating genes from multiple species in the same curation session, but it could not support annotation of the interactions between species, nor the annotation of genes from naturally occurring strains. We extended and customized Canto to support the annotation of multiple strains of multiple species, and the modeling and annotation of interspecies interactions between pathogens and hosts, to create a new tool: PHI-Canto (the Pathogen–Host Interaction Community Annotation Tool). Likewise, there were no existing biomedical ontologies that could accurately describe pathogen–host interaction phenotypes at the depth and breadth required for PHI-base. Infectious disease formation depends on a series of complex and dynamic interactions between pathogenic species and their potential hosts, and also requires the correct biotic and/or abiotic environmental conditions (*Scholthof, 2007*), as illustrated

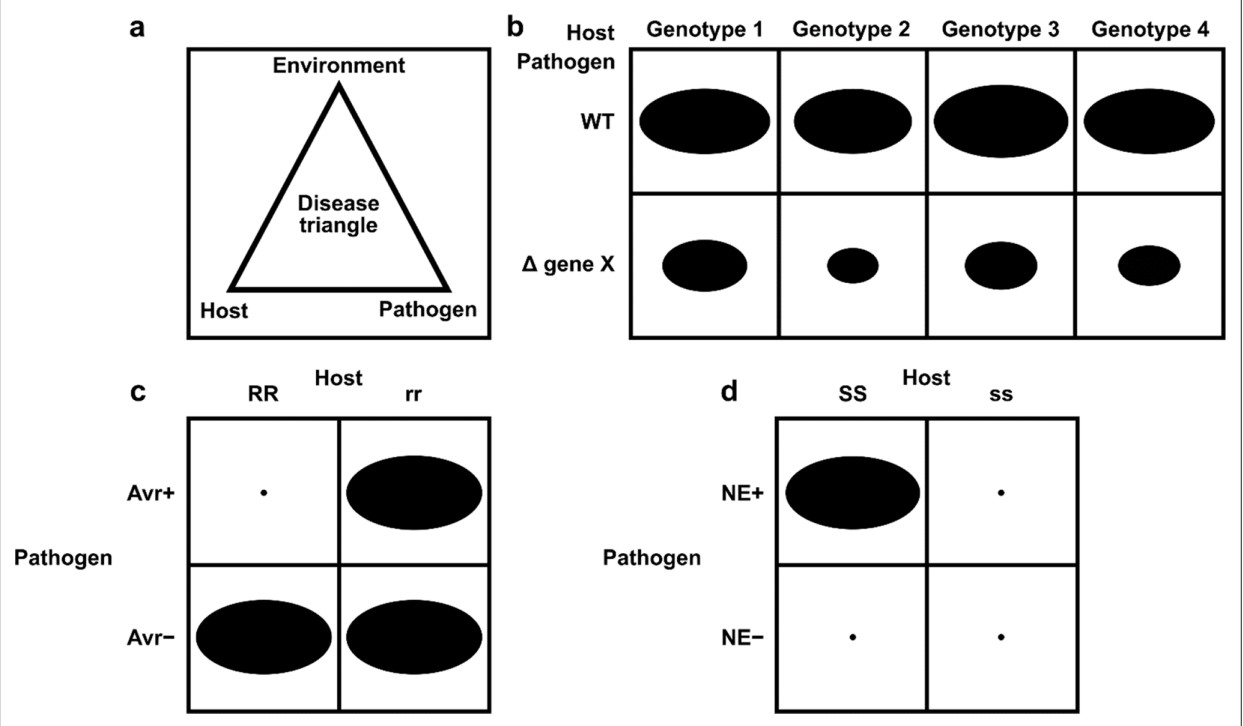

**Figure 2.** Schematic representation of pathogen–host interactions. (**a**) The disease triangle illustrates the requirement for the correct abiotic and biotic environmental conditions to ensure disease when an adapted pathogen encounters a suitable host. (**b**) A non-gene-for-gene genetic relationship where compatible interactions result in disease on all host genotypes (depicted as genotypes 1–4), but the extent of disease formation is influenced to a greater or lesser extent by the presence or absence of a single pathogen virulence gene product X. In host genotypes 1 and 3, the pathogen gene product X is the least required for disease formation. The size of each black oval in each of the eight genetic interactions indicates the severity of the disease phenotype observed, with a larger oval indicating greater severity. (**c**) A gene-for-gene genetic relationship. In this genetic system, considerable specificity is observed, which is based on the direct or indirect interaction of a pathogen avirulence (*Avr*) effector gene product with a host resistance (*R*) gene product to determine specific recognition (an incompatible interaction), which is typically observed in biotrophic interactions (*Jones and Dangl, 2006*). In one scenario, the product of the *Avr* effector gene binds to the product of the *R* gene (a receptor) to activate host resistance mechanisms. In another scenario, the product of the *Avr* effector gene binds to an essential host target which is guarded by the product of the *R* gene (a receptor). Once *Avr* effector binding is detected, host resistance mechanisms are activated. The absence of the *Avr* effector product or the absence of the *R* gene product leads to susceptibility (a compatible interaction). The small black dot indicates no disease formation, and the large black oval indicates full disease formation. (**d**) An inverse gene-for-gene genetic relationship. Again, considerable specificity is observed based on the interaction of a pathogen necrotrophic effector (*NE*) with a host susceptibility (*S*) target to determine specific recognition. The product of the pathogen *NE* gene binds to the product of the *S* gene (a receptor) to activate host susceptibility mechanisms.

by the concept of the 'disease triangle' (*Figure 2*). All these interrelated factors must be recorded in order to sufficiently describe a pathogen–host interaction.

In this study, three key issues were addressed in order to develop the curation framework for interspecies interactions: first, to support the classification of genes as 'pathogen' or 'host,' and enable the variations of the same gene in different strains to be captured; second, formulating the concept of a 'metagenotype' to represent the interaction between specific strains of both a pathogen and a host within a multispecies genotype; and thirdly, developing supporting ontologies and controlled vocabularies, including the generic Pathogen–Host Interaction Phenotype Ontology (PHIPO), to annotate phenotypes connected to genotypes at the level of a single species (pathogen or host) and multiple species (pathogen–host interaction phenotypes). Leading on from these advances, we discuss how the overall curation framework described herein, the concept of annotating metagenotypes, and ongoing generic ontology development, is a suitable approach for adoption and use by a wide range of research communities in the life sciences focused on different types of interspecies interactions occurring within or across kingdoms in different environments and at multiple (micro to macro) scales.

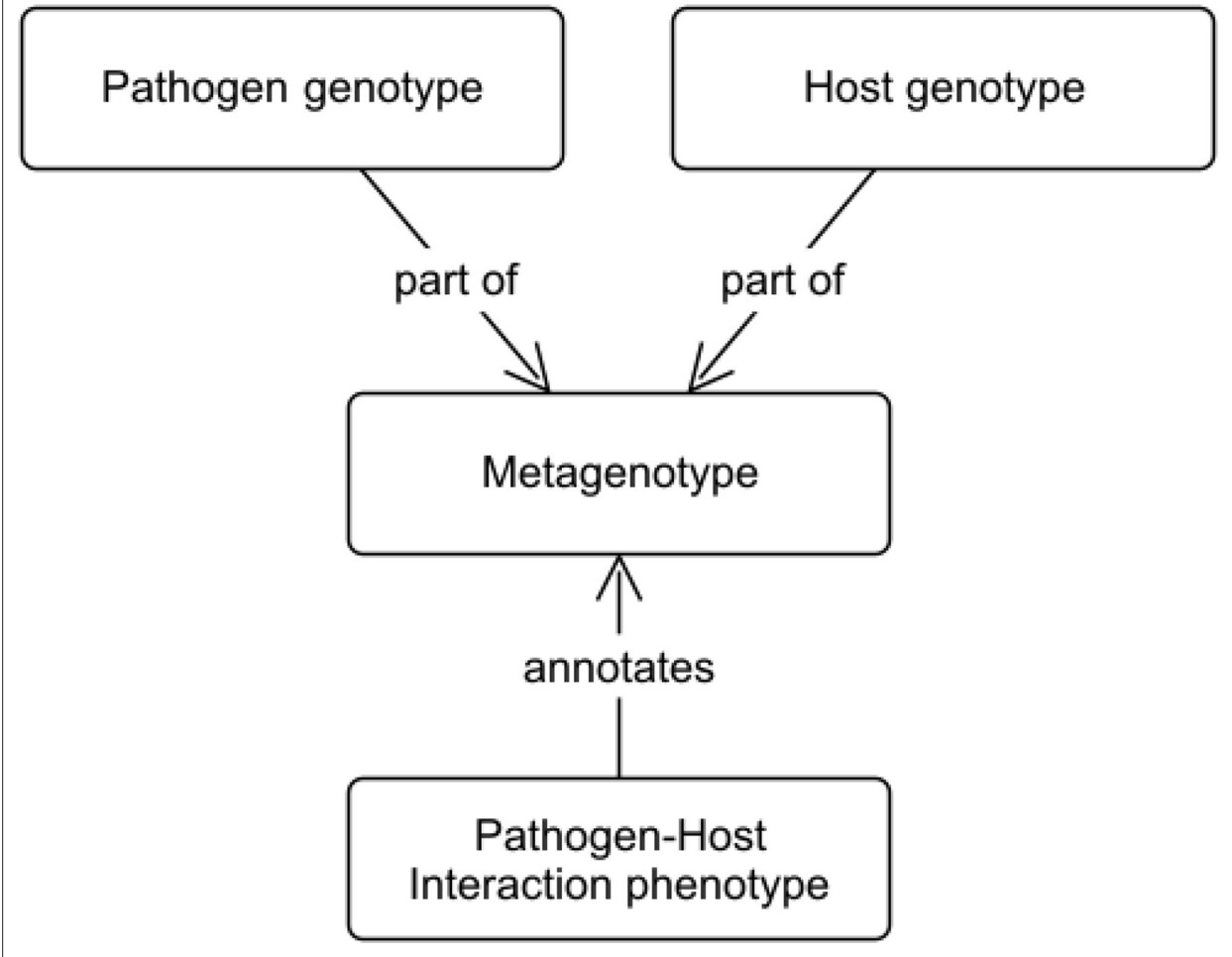

**Figure 3.** Conceptual model showing the relationship between metagenotypes, genotypes, and annotations. The curator selects a pathogen genotype and a host genotype to combine into a metagenotype. The metagenotype can be annotated with pathogen–host interaction phenotypes from PHIPO (the Pathogen–Host Interaction Phenotype Ontology).

The online version of this article includes the following figure supplement(s) for figure 3:

**Figure supplement 1.** Canto entity-relationship model.

**Figure supplement 2.** Entity–relationship model for the main Canto database.

**Figure supplement 3.** Entity–relationship model for a Canto curation session database.

## Results
### Enabling multispecies curation with UniProtKB accessions
In any curation context, stable identifiers are required for annotated entities. The UniProt Knowledge-base (UniProtKB) (*Bateman et al., 2021*) is universally recognized, provides broad taxonomic protein coverage, and manually curates standard nomenclature across protein families. Protein sequences are both manually and computationally annotated in UniProtKB, providing a wealth of data on catalytic activities, protein structures, and protein–protein interactions, Gene Ontology (GO) annotations, and links to PHI-base phenotypes (*Ashburner et al., 2000*; *Carbon et al., 2021*; *Urban et al., 2022*). To improve interoperability with other resources, we used UniProtKB accession numbers for retrieving protein entities, gene names, and species information for display in PHI-Canto. PHI-Canto accesses the UniProtKB API to automatically retrieve the entities and their associated data.

### Developing the metagenotype to capture interspecies interactions
To enable the annotation of interspecies interactions, we developed the concept of a 'metageno-type,' which represents the combination of a pathogen genotype and a host genotype (*Figure 3*). A

metagenotype is created after the individual genotypes from both species are created. Each metagenotype can be annotated with pathogen–host interaction phenotypes to capture changes in pathogenicity (caused by alterations to the pathogen) and changes in virulence (caused by alterations to the host and/or the pathogen). Pathogenicity is a property of the pathogen that describes the ability of the pathogen to cause an infectious disease in another organism. When a pathogenic organism causes disease, the severity of the disease that occurs is referred to as 'virulence' and this can also be dependent upon the host organism. Metagenotypes must always include at least one named pathogen gene with a genotype of interest, but need not include a host gene if none is referenced in a given experiment: instead, the wild-type host species and strain may be used for the host part of the metagenotype.

## Annotation types and annotation extensions in PHI-Canto

In PHI-Canto, 'annotation' is the task of relating a specific piece of knowledge to a biological feature. Three types of biological features can be annotated in PHI-Canto: genes, genotypes, and metagenotypes. Genotypes can be further specified as pathogen genotypes or host genotypes. Each of these biological features has a corresponding set of annotation types. The relation between biological features, annotation types, and the values that can be used for annotation are shown in *Table 1*. To capture additional biologically relevant information associated with an annotation, curators use the concept of annotation extensions (which include Gene Ontology annotations described by *Huntley et al., 2014*) to extend the primary annotation. For Canto and PHI-Canto, the meaning of 'annotation extension' was broadened to capture additional properties related to the annotation, such as the metagenotype used as an experimental control. The aforementioned additional properties are simply referred to as 'annotation extensions (AEs)' in this study (*Table 1*, *Supplementary file 1* and *Supplementary file 2*). Descriptions of the new AEs for PHI-Canto and the core collection of AEs from Canto are available in the PHI-Canto user documentation (see the Code availability section).

Metagenotypes can be annotated with terms from an ontology or controlled vocabulary following either the 'pathogen–host interaction phenotype,' 'gene-for-gene phenotype,' or 'disease name' annotation types (*Table 1*). Phenotype annotations on metagenotypes can be supported by AEs providing additional qualifying information required to fully interpret the experiment, such as the infected tissue of the host.

Phenotypes can also be curated for single-species experiments involving either the pathogen or host, following the 'single species phenotype' annotation workflow (*Table 1*). Single species phenotype annotations have a selection of AEs available, including the protein assayed in the experiment and the severity of the observed phenotype (see an example from PMID:22314539 in Appendix 1).

PHI-Canto also supports the annotation of gene and gene product attributes to represent the evolved functional role of a gene product, described here as the 'gene annotation' workflow (*Table 1*). The Gene Ontology is used for the annotation of a gene product's molecular functions, biological processes, and cellular components, while PSI-MOD is used for the annotation of protein modifications (*Montecchi-Palazzi et al., 2008*), and BioGRID experiment types are used to capture genetic and physical interactions (*Oughtred et al., 2021*). GO annotations are submitted to the EBI GO Annotation Database (GOA), from where they are propagated to the main GO knowledge base (*Carbon et al., 2021*; *Huntley et al., 2015*).

## Trial curation of interspecies interaction publications

Ten publications covering a wide range of typical plant, human, and animal pathogen–host interactions were selected for trial curation in PHI-Canto before the tool was made available to publication authors and communities to add further publications (*Table 2*). These publications included experiments with early-acting pathogen virulence proteins, the first host targets of pathogen effectors, and resistance to antifungal chemistries. These publications guided the development of the ontology terms and controlled vocabulary terms that were required for PHI-Canto, as well as the curation methods required for different experiments. Major curation problems and their solutions are summarized in *Table 3*, and example annotations are described below and in Appendix 1 and Appendix 2.

**Table 1.** Annotation types and annotation extensions in the Pathogen–Host Interaction Community Annotation Tool (PHI-Canto), grouped by the biological feature being annotated.

| Annotation type | Annotation extensions * | Annotation value |
|---|---|---|
| **Annotation types for the *gene* biological feature** [†] | | |
| Gene Ontology annotation | | Gene Ontology term |
| | with host species | NCBI Taxonomy ID |
| | with symbiont species | NCBI Taxonomy ID |
| Wild-type expression | | PomBase Gene Expression ontology term |
| | during | Gene Ontology biological process term [‡] |
| | in presence of | Chemical entity (ChEBI ontology) |
| | tissue type | BRENDA Tissue Ontology term |
| **Annotation types for the *genotype* biological feature** | | |
| Single species phenotype (Pathogen phenotype or Host phenotype) | | PHIPO term (single-species phenotype branch) |
| | affected proteins | UniProtKB accession number (one for each affected protein) |
| | assayed RNA [§] | UniProtKB accession number |
| | assayed protein | UniProtKB accession number |
| | observed in organ | BRENDA Tissue Ontology term [¶] |
| | penetrance | Qualitative value (low, normal, high, complete) or quantitative value (percentage) |
| | severity | Qualitative value (low, normal, high, variable) or quantitative value (percentage) |
| **Annotation types for the *metagenotype* biological feature** | | |
| Pathogen–host interaction phenotype or Gene-for-gene phenotype | | PHIPO term (pathogen–host interaction phenotype branch) |
| | affected proteins | UniProtKB accession number (one for each affected protein) |
| | assayed protein | UniProtKB accession number |
| | assayed RNA | UniProtKB accession number |
| | compared to control metagenotype | Metagenotype ** |
| | extent of infectivity [††] | PHIPO term |
| | gene-for-gene interaction [‡‡] | PHIPO Extension (PHIPO_EXT) ontology term |
| | host tissue infected | BRENDA Tissue Ontology term |
| | inverse gene-for-gene interaction [‡‡] | PHIPO Extension (PHIPO_EXT) ontology term |
| | outcome of interaction [††] | PHIPO term |
| | penetrance | Qualitative value (low, normal, high, complete) or quantitative value (percentage) |
| | severity | Qualitative value (low, normal, high, variable) or quantitative value (percentage) |
| Disease name | | PHIDO term [§§] |
| | host tissue infected | BRENDA Tissue Ontology term |

*PHI-Canto uses 44 annotation extension (AE) relations, of which nine are unique to PHI-base, while the remaining 35 are shared with PomBase.

[†]Additional AEs shared with PomBase for the gene annotation types are available in ***Supplementary file 2***.

[‡]Restricted to GO:0022403, GO:0033554, GO:0072690, GO:0051707 and their descendant terms.

[§]AE relates to mRNA.

[¶]Restricted to BTO:0001489, BTO:0001494, BTO:0001461 and their descendant terms.

**Metagenotypes are selected from those already added to the curation session.

[††]AE only applies to pathogen–host interaction phenotypes.

[‡‡]AE only applies to gene-for-gene phenotypes.

[§§]Curated list of disease names.

**Table 2.** Publications selected for trial curation using the Pathogen–Host Interaction Community Annotation Tool (PHI-Canto).

| Subject of publication | PMID | Publication title | Genotype * annotated with | Metagenotype † annotated with |
|---|---|---|---|---|
| Bacteria–human interaction | 28715477 ‡ | The RhlR quorum-sensing receptor controls *Pseudomonas aeruginosa* pathogenesis and biofilm development independently of its canonical homoserine lactone autoinducer. | Pathogen phenotype | unaffected pathogenicity, altered pathogenicity or virulence |
| Fungal–human interaction/novel antifungal target | 28720735 § | A nonredundant phosphopantetheinyl transferase, PptA, is a novel antifungal target that directs secondary metabolite, siderophore, and lysine biosynthesis in *Aspergillus fumigatus* and is critical for pathogenicity. | Pathogen phenotype | unaffected pathogenicity, altered pathogenicity or virulence |
| Secondary metabolite clusters required for pathogen virulence | 30459352 § | Phosphopantetheinyl transferase (Ppt)-mediated biosynthesis of lysine, but not siderophores or DHN melanin, is required for virulence of *Zymoseptoria tritici* on wheat. | Pathogen phenotype | unaffected pathogenicity, altered pathogenicity or virulence |
| Early acting virulence proteins | 29020037 §, ¶ | A conserved fungal glycosyltransferase facilitates pathogenesis of plants by enabling hyphal growth on solid surfaces. | Pathogen phenotype | altered pathogenicity or virulence |
| Mutualism interaction | 16517760 ** | Reactive oxygen species play a role in regulating a fungus-perennial ryegrass mutualistic interaction | Pathogen phenotype | mutualism |
| First host targets of pathogen effectors | 31804478 §, †† | An effector protein of the wheat stripe rust fungus targets chloroplasts and suppresses chloroplast function. | N/A | altered pathogenicity or virulence a pathogen effector |
| Receptor decoys | 30220500 †† | Suppression of plant immunity by fungal chitinase-like effectors. | Pathogen phenotype | a pathogen effector |
| R-Avr interactions | 20601497 ‡‡, §§ | Activation of an *Arabidopsis* resistance protein is specified by the *in planta* association of its leucine-rich repeat domain with the cognate oomycete effector. | Host phenotype | a pathogen effector a gene-for-gene interaction |
| Fungal toxins required for virulence on plants | 22241993 ¶¶ | The cysteine rich necrotrophic effector SnTox1 produced by *Stagonospora nodorum* triggers susceptibility of wheat lines harboring Snn1. | N/A | a pathogen effector a gene-for-gene interaction (inverse) |
| Resistance to antifungal chemistries | 22314539 *** | The T788G mutation in the cyp51C gene confers voriconazole resistance in *Aspergillus flavus* causing aspergillosis. | Pathogen phenotype Pathogen chemistry phenotype | N/A |

*Single species genotypes could be annotated with either a pathogen phenotype, a pathogen chemistry phenotype, or a host phenotype. Genotypes are annotated with *in vitro* or *in vivo* phenotypes from PHIPO, using either the Pathogen phenotype or Host phenotype annotation type workflow.

†Metagenotype comprises of a pathogen and a host genotype in combination. Phenotypes from PHIPO can be annotated to metagenotypes using either the 'Pathogen–Host Interaction Phenotype' or 'Gene-for-Gene Phenotype' annotation type workflow.

‡Example of curating 'unaffected pathogenicity' available in Appendix 1.

§Example of curating 'altered pathogenicity or virulence' available in Appendix 1 and Appendix 2.

¶Example of '*in vitro* pathogen phenotype' available in Appendix 1.

**Example of curating 'mutualism' available in Appendix 1. Although 'mutualism interactions' are generally out of scope for PHI-base, PHI-Canto can be used to curate these publications if required. In this study, the fungal gene mutation altered the interaction from mutualistic to antagonistic.

††Example of curating 'a pathogen effector' available in Appendix 1.

‡‡Example of curating 'a gene-for-gene interaction' available in Appendix 1.

§§Example of '*in vivo* host phenotype' available in Appendix 1.

¶¶Example of curating 'an inverse gene-for-gene interaction' available in Appendix 1.

***Example of '*in vitro* pathogen chemistry phenotype' available in Appendix 1.

## Curating an experiment with a metagenotype

A large proportion of the curation in PHI-Canto requires the use of metagenotypes: one of the simpler cases involves early-acting virulence proteins, where a genetically modified pathogen is inoculated onto a host (without a host gene being specified). A metagenotype is created to connect the genotypes of both species and is annotated with a phenotype term. These experiments are curated following the 'pathogen–host interaction phenotype' workflow, including any relevant AEs (*Table 1*). This two-step curation process is illustrated by PMID:29020037 curation (*Table 2*, Appendix 1 and Appendix 2) where the *GT2* gene is deleted from the fungal plant pathogen *Zymoseptoria tritici* and inoculated onto wheat plants; the observed phenotype 'absence of pathogen-associated host lesions'

**Table 3.** Issues encountered whilst curating ten example publications with the Pathogen–Host Interaction Community Annotation Tool (PHI-Canto).

| Curated feature | Problem description | Solution | Context in PHI-Canto | Example |
|---|---|---|---|---|
| Species strain | UniProtKB sequence information is commonly from a reference genome strain. This sequence may differ from the experimental strain curated in PHI-Canto. | Develop a selectable list of strains for curators to assign to the genotype (and metagenotype). | Strain selected after UniProtKB entry on gene entry page. Strain used within genotype creation. | URL[1] All phenotype annotation examples in Appendix 1 contain a 'strain name' within the genotype/metagenotype. |
| Delivery mechanism | Pathogen–host interaction experiments use a wide array of mechanisms to deliver the treatment of choice (to cells, tissues, and host and non-host species) which are required for experimental interpretation. | Develop terms prefixed with 'delivery mechanism' in the Pathogen–Host Interaction Experimental Conditions Ontology (PHI-ECO). | Selection of experimental conditions whilst making a phenotype annotation to a metagenotype. | URL[2] Examples in Appendix 1 PMID:20601497, PMID:31804478 and PMID:22241993. |
| Physical interaction | Physical interactions (i.e. protein–protein interactions) could only be annotated between proteins of the same species, so it was not possible to annotate interactions between a pathogen effector and its first host target. | Adapt the 'Physical Interaction' annotation type to store gene and species information from two organisms (instead of one). | Physical Interaction annotation type. | URL[3] |
| Pathogen effector | There was no available ontology term to describe a 'class' pathogen effector (a 'transferred entity from pathogen to host'), because effectors have heterogeneous functions (specific enzyme inhibitors, modulating host immune responses, and targeting host gene-silencing mechanisms). Effector is not a phenotype, and so did not fit into the Pathogen–Host Interaction Phenotype Ontology (PHIPO). | Develop new Gene Ontology (GO) biological process terms (and children), to group 'effector-mediated' processes. | GO Biological Process annotation on a pathogen gene. | URL[4] Example in Appendix 1 PMID:31804478. |
| Wild-type control phenotypes | Natural sequence variation between strains of both pathogen and host organisms can alter the phenotypic outcome within an interaction. The wild-type metagenotype phenotype needs to be curated so that the phenotype of an altered metagenotype is informative. | Allow creation of metagenotypes containing wild-type genes. Develop a new annotation extension (AE) property 'compared to control,' used in annotation of altered metagenotypes. | Annotation of phenotypes and AEs to metagenotypes (using the 'PHI phenotype' or 'Gene for gene phenotype' annotation type). | URL[5] Examples in Appendix 1 PMID:28715477, PMID:16517760, PMID:29020037, PMID:20601497, PMID:22241993. |
| Chemistry | How to record chemicals for resistance or sensitivity phenotypes. | Follow PomBase model to pre-compose PHIPO terms to include chemical names from the ChEBI ontology. | Annotation of phenotypes to single species genotypes. | URL[4] Example in Appendix 1 PMID:22314539. |

*Table 3 continued on next page*

*Table 3 continued*

| Curated feature | Problem description | Solution | Context in PHI-Canto | Example |
|---|---|---|---|---|
| Gene for gene interactions | Complex gene-for-gene interactions within plant pathogen–host interactions required additional detail to describe the function of the pathogen and host genes within the metagenotype (including the specified strains). | Develop the additional metagenotype curation type 'Gene for gene phenotype.' Develop two new AEs, 'gene_for_gene_ interaction' and 'inverse gene_for_gene_ interaction,' using PHIPO_EXT terms describing three components of the interaction.* | Annotation of phenotypes and AEs to metagenotypes using the 'Gene for gene phenotype' annotation type. | URL[4] Examples in Appendix 1 PMID:20601497 and PMID:22241993. |
| Nine high-level legacy terms (from PHI-base 4) | PHI-base should incorporate legacy data from PHI-base 4 into new PHI-base 5 gene-centric pages. | Maintain the nine high level terms as 'tags' within the new PHI-base 5 user interface. Develop mapping methods to enable this. | Three locations described in *Supplementary file 3*. | *Urban et al., 2015* NAR (PMID:25414340). |

URL[1] https://canto.phi-base.org/docs/getting_started#adding_strains.
URL[2] https://canto.phi-base.org/docs/phipo_annotation#experimental_conditions.
URL[3] https://canto.phi-base.org/docs/physical_interaction_annotation.
URL[4] https://canto.phi-base.org/docs/phipo_annotation#pathogen_host_interaction_phenotypes.
URL[5] https://canto.phi-base.org/docs/genotypes#metagenotype_management.
*Namely, (i) the compatibility of the interaction (ii) the functional status of the pathogen gene, and (iii) the functional status of the host gene.

(PHIPO:0000481) is annotated to the metagenotype; and the AE for 'infective ability' is annotated with 'loss of pathogenicity' compared to the unaltered pathogen.

## Curating pathogen effector experiments

A pathogen effector is defined as an entity transferred between the pathogen and the host that is known or suspected to be responsible for either activating or suppressing a host process commonly involved in defense (*Houterman et al., 2009*; *Jones and Dangl, 2006*; *Figure 2*). To curate an effector experiment, a metagenotype is created and annotated with a phenotype term. To indicate that the pathogen gene functions as an effector, it is necessary to make a concurrent gene annotation (*Table 1*) with the GO biological process term 'effector-mediated modulation of host process' (GO:0140418) or an appropriate descendant term. This GO term (GO:0140418) and its descendant terms were created in collaboration with the Gene Ontology Consortium (GOC) and are used for pathogen effectors in PHI-base (version 5) (*Supplementary file 3*). Reported activities of pathogen effectors can also be curated with GO molecular function terms. An example of curation of a pathogen effector experiment is illustrated using PMID:31804478 (*Table 2* and Appendix 1) where the pathogen effector Pst_12806 from *Puccinia striiformis* suppresses pattern-triggered immunity in a tobacco leaf model. Here, the metagenotype is annotated with the phenotype 'decreased level of host defense-induced callose deposition' (PHIPO:0001015) and the effector is annotated with 'effector-mediated suppression of host pattern-triggered immunity' (GO:0052034). A further experiment demonstrated that the pathogen effector protein was able to bind to the natural host (wheat) protein PetC and inhibit its enzyme activity, resulting in a GO molecular function annotation 'enzyme inhibitor activity' (GO:0004857) for Pst_12806, with PetC captured as the target protein (see Appendix 1).

## Curating experiments with a gene-for-gene relationship

For a gene-for-gene pathogen–host interaction type, the 'gene-for-gene phenotype' metagenotype workflow is followed (a gene-for-gene interaction is when a known genetic interaction is conferred by a specific pathogen avirulence gene product and its cognate host resistance gene product) (*Figure 2c and d*, further described in the figure legend *Flor, 1956*; *Jones and Dangl, 2006*; *Kanyuka et al., 2022*). The metagenotypes and phenotype annotations are made in the same way as the standard 'pathogen–host interaction phenotype' workflow, but with different supporting data. A new AE was created to indicate the following three components of the interaction: (i) the compatibility of the interaction, (ii) the functional status of the pathogen gene, and (iii) the functional status of the host gene. An example of an annotation for a biotrophic pathogen gene-for-gene interaction has been illustrated

with PMID:20601497 (*Table 2* and Appendix 1). Inverse gene-for-gene relationships occur with necrotrophic pathogens, where the pathogen necrotrophic effector interacts with a gene product from the corresponding host susceptibility locus and activates a host response that benefits the pathogen (a compatible interaction). If the necrotrophic effector cannot interact with the host target, then no disease occurs (an incompatible interaction) (*Breen et al., 2016*). An example of an inverse gene-for-gene interaction using the appropriate AEs is illustrated with PMID:22241993 (*Table 2* and Appendix 1).

## Curating an experiment with a single species genotype in the presence or absence of a chemical

Single species genotypes (pathogen or host) can also be annotated with phenotypes following the 'single species phenotype' workflow (*Table 1*). This is illustrated using PMID:22314539 in *Table 2* (and Appendix 1) with an example of an *in vitro* pathogen chemistry phenotype, where a single nucleotide mutation in the *Aspergillus flavus* cyp51C gene confers 'resistance to voriconazole' (PHIPO:0000590), an antifungal agent.

## Supporting curation of legacy information

PHI-Canto's curation workflows maintain support for nine high-level terms that describe phenotypic outcomes essential for taxonomically diverse interspecies comparisons, which were the primary annotation method used in previous versions of PHI-base (*Urban et al., 2015*) and which are displayed in the Ensembl Genomes browser (*Yates et al., 2022*). For example, the 'infective ability' AE can be used to annotate the following subset of high-level terms: 'loss of pathogenicity,' 'unaffected pathogenicity,' 'reduced virulence,' 'increased virulence,' and 'loss of mutualism' (formerly 'enhanced antagonism'). The mapping between the nine high-level terms and the PHI-Canto curation process is further described in *Supplementary file 3*.

## Resolving additional problems with curating complex pathogen–host interactions

*Table 3* shows a selection of the problems encountered during the development of PHI-Canto and the solutions we identified: for example, recording the delivery mechanism used within the pathogen–host interaction experiment. New experimental condition terms were developed with a prefix of 'delivery mechanism': for example, 'delivery mechanism: agrobacterium,' 'delivery mechanism: heterologous organism,' and 'delivery mechanism: pathogen inoculation.' Another issue encountered was how to record a physical interaction between two proteins from different species, especially for the curation of pathogen effectors and their discovered first host targets. This was resolved by adapting the existing Canto module for curating physical interactions to support two different species.

## Development of the Pathogen–Host Interaction Phenotype Ontology and additional data lists

To support the annotation of phenotypes in PHI-Canto, PHIPO was developed. PHIPO is a species-neutral phenotype ontology that describes a broad range of pathogen–host interaction phenotypes. Terms in PHIPO were developed following a pre-compositional approach, where the term names and semantics were composed from existing terms from other ontologies, in order to make the curation process easier. For example, the curator annotates 'resistance to penicillin' (PHIPO:0000692) instead of annotating 'increased resistance to chemical' (PHIPO:0000022) and 'penicillin' (CHEBI:17334) separately. Terms in PHIPO have logical definitions that follow design patterns from the uPheno ontology (*Shefchek et al., 2020*), and mapping PHIPO terms to the uPheno patterns is an ongoing effort. These logical definitions provide relations between phenotypes in PHIPO and terms in other ontologies, such as PATO, GO, and ChEBI. PHIPO is available in OWL and OBO formats from the OBO Foundry (*Jackson et al., 2021*).

PHI-Canto uses additional controlled vocabularies derived from data in PHI-base. To enable PHI-Canto to distinguish between pathogen and host organisms, we extracted a list of >250 pathogen species and >200 host species from PHI-base (*Supplementary file 4*). A curated list of strain names and their synonyms for the species currently curated in PHI-base was also developed for use in PHI-Canto (*Supplementary files 4 and 5*). PHI-base uses 'strain' as a grouping term for natural pathogen

isolates, host cultivars, and landraces, all of which are included in the curated list. The curation of pathogen strain designations was motivated by the NCBI Taxonomy's decision to discontinue the assignment of strain-level taxonomic identifiers (*Federhen et al., 2014*) and a lack of standardized nomenclature for natural isolates of non-model species. New strain designations can be requested by curators and are reviewed by an expert prior to inclusion to ensure that each describes a novel strain designation rather than a new synonym for an existing strain.

Annotations in PHI-Canto include experimental evidence, which is specified by a term from a subset of the Evidence & Conclusion Ontology (ECO) (*Giglio et al., 2019*). Experimental evidence codes specific to pathogen–host interaction experiments have been developed and submitted to ECO. Phenotype annotations also include experimental conditions that are relevant to the experiment being curated, which are sourced from the PHI-base Experimental Conditions Ontology (PHI-ECO).

PHI-Canto includes a 'disease name' annotation type (*Table 1*) for annotating the name of the disease caused by an interaction between the pathogen and host specified in a wild-type metagenotype (this annotation type is described in the PHI-Canto user documentation and in Appendix 2). Diseases are specified by a controlled vocabulary of disease names (called PHIDO), which was derived from disease names curated in previous versions of PHI-base (*Urban et al., 2022*). PHIDO was developed as a placeholder to allow disease names to be annotated on a wide variety of pathogen interactions, including those on plant, human, animal, and invertebrate hosts, especially where such diseases were not described in any existing ontology.

## Summary of the PHI-Canto curation process

The PHI-Canto curation process is outlined in *Figure 4*, *Figure 4—figure supplement 1*, the PHI-Canto user documentation, a detailed worked example provided in Appendix 2 and curation tutorials on the PHI-base YouTube channel (https://www.youtube.com/@PHI-base), under the playlist 'PHI-Canto tutorial videos.' Each curation session is associated with one publication (using its PubMed identifier). One or more curators can collaborate on curating the same publication. An instructional email is sent by PHI-Canto to curators when they begin a new curation session, and PHI-base provides further guidelines on what information is needed to curate a publication in PHI-Canto (*Figure 4—figure supplement 2*) and how to identify UniProtKB accession numbers from reference proteomes (*Figure 4—figure supplement 3*).

The curator first adds genes from the publication, then creates alleles from genes, genotypes from alleles, and metagenotypes from pathogen and host genotypes. Pathogen genotypes and host genotypes are created on separate pages, that only include genes from the relevant pathogen or host. A genotype can consist of multiple alleles, therefore, a metagenotype can contain multiple alleles from both the pathogen and the host. A 'copy and edit' feature allows the creation of multiple similar annotations.

To make annotations, the curator selects a gene, genotype, or metagenotype to annotate, then selects a term from a controlled vocabulary, adds experimental evidence, experimental conditions, AEs (where available), and any additional comments. The curator can also specify a figure or table number from the original publication as part of the annotation. Curators can use a term suggestion feature to suggest new terms for any controlled vocabulary used by PHI-Canto, and experimental conditions can be entered as free text if no suitable condition is found in PHI-ECO. Subsequently, new condition suggestions are reviewed and approved by expert curators. The curation session can be saved and paused at various stages during the curation process. Once the curation process is complete, the curator submits the session for review by a nominated species expert.

## Display and interoperability of data

The process of incorporating FAIR principles fully into the PHI-base curation process will promote interoperability between data resources (*Wilkinson et al., 2016*). *Figure 5* illustrates the internal and external resource dependencies for curation in PHI-Canto. URLs and descriptions of the use of each resource are provided in *Figure 5—figure supplement 1*. All data curated in PHI-Canto will be displayed in the new gene-centric version 5 of PHI-base, introduced in *Urban et al., 2022*. Additional detail on the data types displayed in PHI-base 5 is available in *Table 4*. Reciprocally, components of the interspecies curation framework (*Figure 6a*) will provide data to other resources (*Figure 6b*). For example, GO terms will be used in curation with PHI-Canto and these annotations will be made

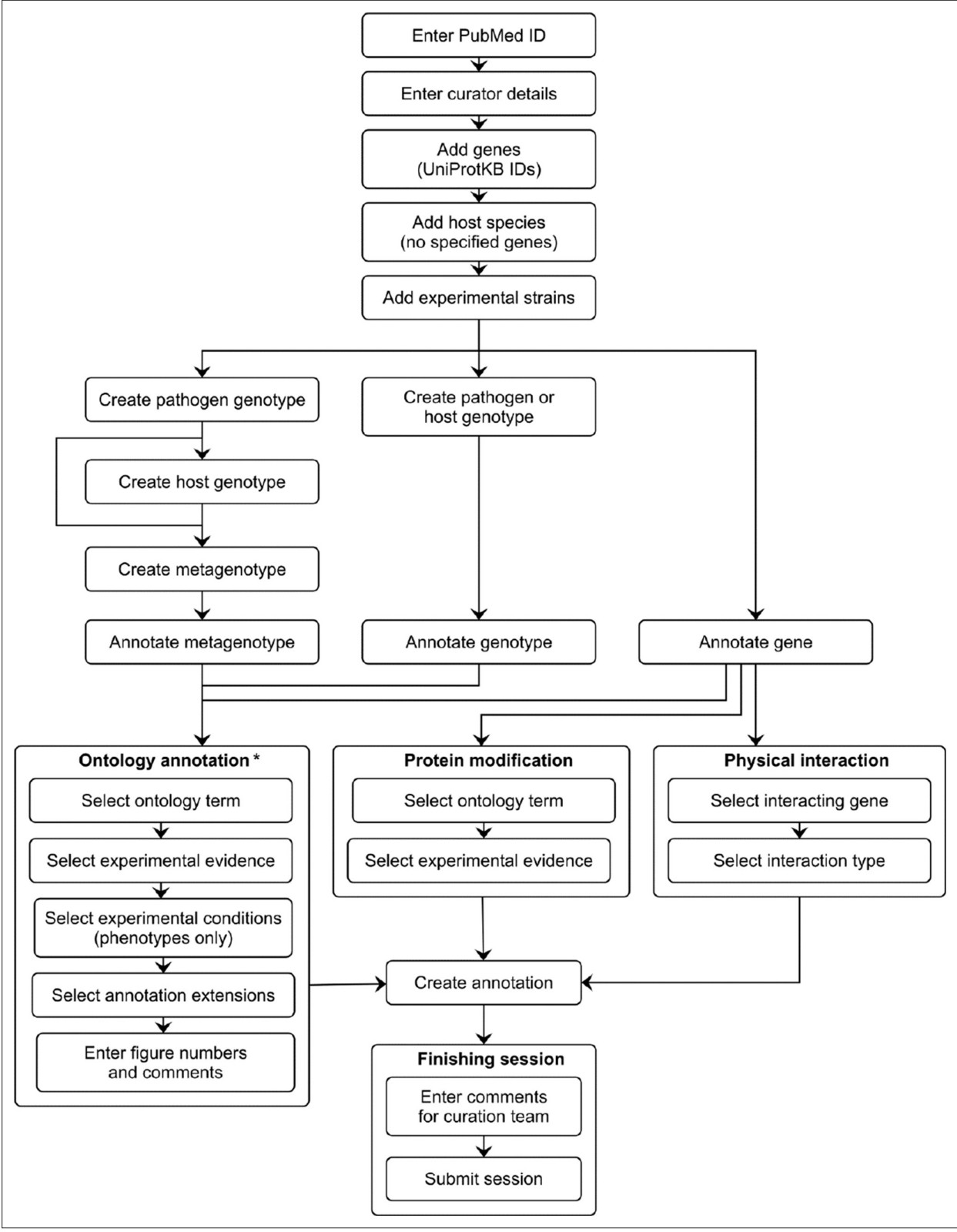

**Figure 4.** Pathogen–Host Interaction Community Annotation Tool (PHI-Canto) curation workflow diagram. This diagram shows the curation workflow from the start of a curation session to its submission. The PubMed ID of the publication to be curated is entered and the title is automatically retrieved. The curator enters their name, email address, and ORCID iD. On the species and genes page, the experimental pathogen and host genes are entered using UniProtKB accession numbers, and for experiments where a mutant pathogen genotype is assayed on a wild-type host with no specified genes,

*Figure 4 continued on next page*

*Figure 4 continued*

there is the option to select the host species from an autocomplete menu. Information on the specific experimental strains used for each species is entered. After entering this initial information, the curator follows one of three distinct workflows depending on the biological feature the user wants to annotate (metagenotype, genotype, or gene annotation type). Except for genes, biological features are created by composing less complex features: genotypes from alleles (generated in the pathogen or host genotype management pages), and metagenotypes from genotypes (generated in the metagenotype management page). Biological features are annotated with terms from a controlled vocabulary (usually an ontology), plus additional information that varies based on the annotation type. The curator has the option to generate further annotations after creating one, but this iterative process is not represented in the diagram for the sake of brevity. After all annotations have been made, the session is submitted into the Pathogen–Host Interactions database (PHI-base) version 5. * Note that the 'Ontology annotation' group covers multiple annotation types, all of which annotate biological features with terms from an ontology or controlled vocabulary. These annotation types are described in *Table 1*.

The online version of this article includes the following figure supplement(s) for figure 4:

**Figure supplement 1.** Alternative curation step workflow.

**Figure supplement 2.** What you need to curate a publication using the Pathogen–Host Interaction Community Annotation Tool (PHI-Canto).

**Figure supplement 3.** Instructions on how to look up a UniProt Knowledgebase (UniProtKB) ID.

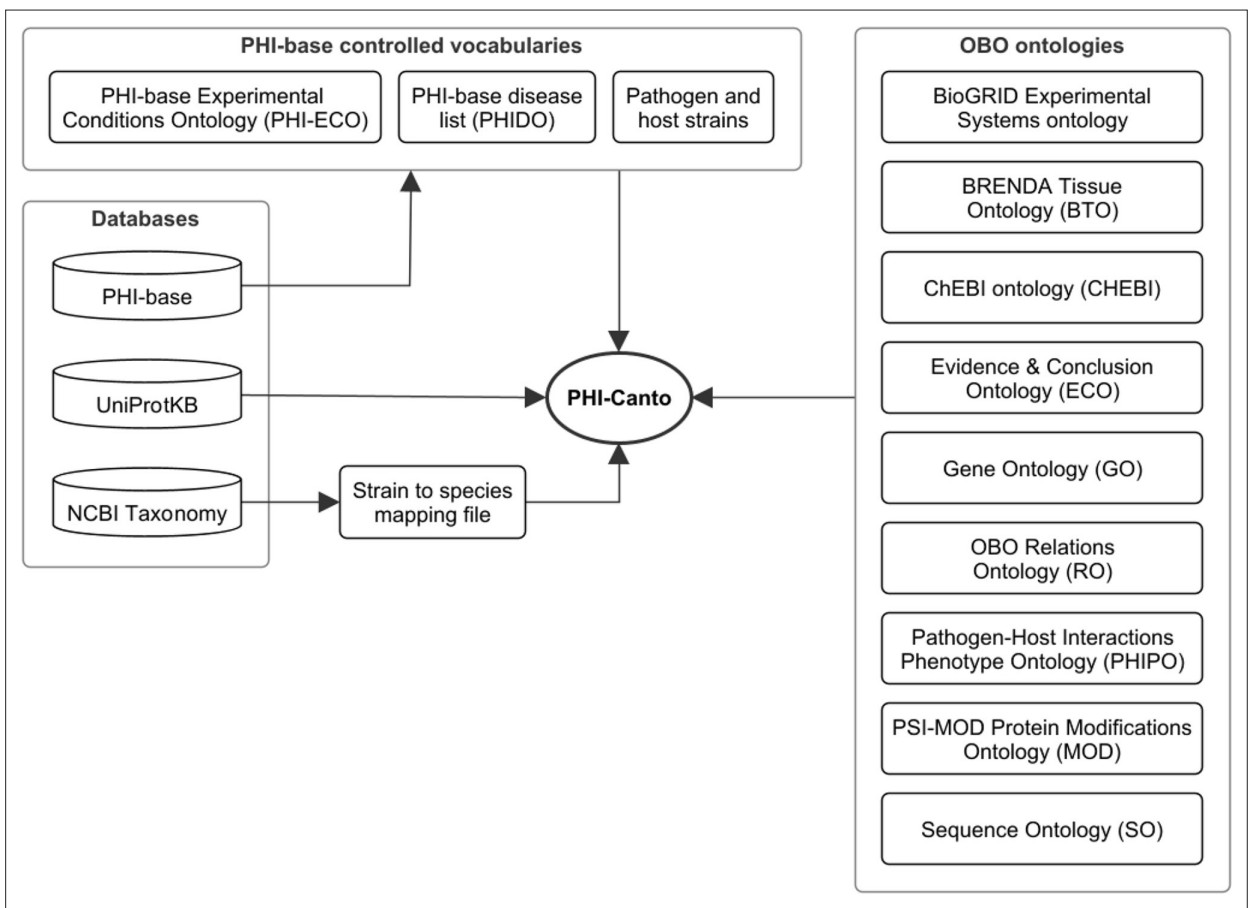

**Figure 5.** Network diagram showing the data resources used by the Pathogen–Host Interaction Community Annotation Tool (PHI-Canto). Of the databases shown, the Pathogen–Host Interactions database (PHI-base) provides data (experimental conditions, disease names, and species strain names) used to create terms in the PHI-base controlled vocabularies; the UniProt Knowledgebase (UniProtKB) provides accession numbers for proteins that PHI-Canto uses to identify genes; and the NCBI Taxonomy database is used to generate a mapping file relating taxonomic identifiers lower than species rank to their nearest taxonomic identifiers at species rank. The OBO ontologies group contains ontologies in the OBO format that PHI-Canto uses for its annotation types. The parenthesized text after the ontology name indicates the term prefix for the ontology.

The online version of this article includes the following figure supplement(s) for figure 5:

**Figure supplement 1.** Resources relied upon by the Pathogen–Host Interaction Community Annotation Tool (PHI-Canto).

**Table 4.** Automatically and manually curated types of data displayed in the gene-centric version 5 of the Pathogen–Host Interactions database (PHI-base).

| Data type | Data source |
| --- | --- |
| **Metadata** | |
| Entry Summary * | UniProtKB [†] |
| Pathogen species | NCBI Taxonomy [†] |
| Pathogen strain | PHI-base strain list |
| Host species | NCBI Taxonomy [†] |
| Host strain | PHI-base strain list |
| Publication | PubMed [†] |
| | |
| **Phenotype annotation sections** | |
| Pathogen–Host Interaction Phenotype | PHIPO [‡] pathogen–host interaction phenotype branch |
| Gene-for-Gene Phenotype | PHIPO pathogen–host interaction phenotype branch |
| Pathogen Phenotype | PHIPO single species phenotype branch |
| Host Phenotype | PHIPO single species phenotype branch |
| | |
| **Other annotation sections** | |
| Disease name | PHIDO |
| GO Molecular Function | GO [§] |
| GO Biological Process | GO |
| GO Cellular Component | GO |
| Wild-type RNA level [¶] | FYPO_EXT [**] |
| Wild-type Protein level | FYPO_EXT |
| Physical Interaction | BioGRID [††] |
| Protein Modification | PSI-MOD [‡‡] |

*The Entry Summary section includes information on which gene is being displayed in the gene-centric results page. The UniProtKB accession number is used to automatically retrieve the name and function of the protein, plus any cross-referenced identifiers from Ensembl Genomes and NCBI GenBank. The section also displays the PHI-base 5 gene identifier (PHIG) and any of the high-level terms (**Supplementary file 3**) annotated to the gene.

[†]Data from UniProtKB, NCBI Taxonomy, and PubMed are automatically retrieved, while all other data are manually curated.

[‡]PHIPO is the Pathogen–Host Interaction Phenotype Ontology.

[§]GO is the Gene Ontology.

[¶]This relates to mRNA.

[**]FYPO_EXT is the Fission Yeast Phenotype Ontology Extension.

[††]BioGRID is the Biological General Repository for Interaction Datasets.

[‡‡]PSI-MOD is the Human Proteome Organization (HUPO) Proteomics Standards Initiative (PSI) Protein Modifications Ontology.

available in the GO knowledge base via submission to the GOA Database (*Carbon et al., 2021*; *Huntley et al., 2015*). PHI-base is a member of ELIXIR, an organization that aims to unite leading life science resources and is a major proponent of FAIR data (*Durinx et al., 2016*).

## Discussion

Scalable and accurate curation of data within the scientific literature is of paramount importance due to the increasing quantity of publications and the complexity of experiments within each publication.

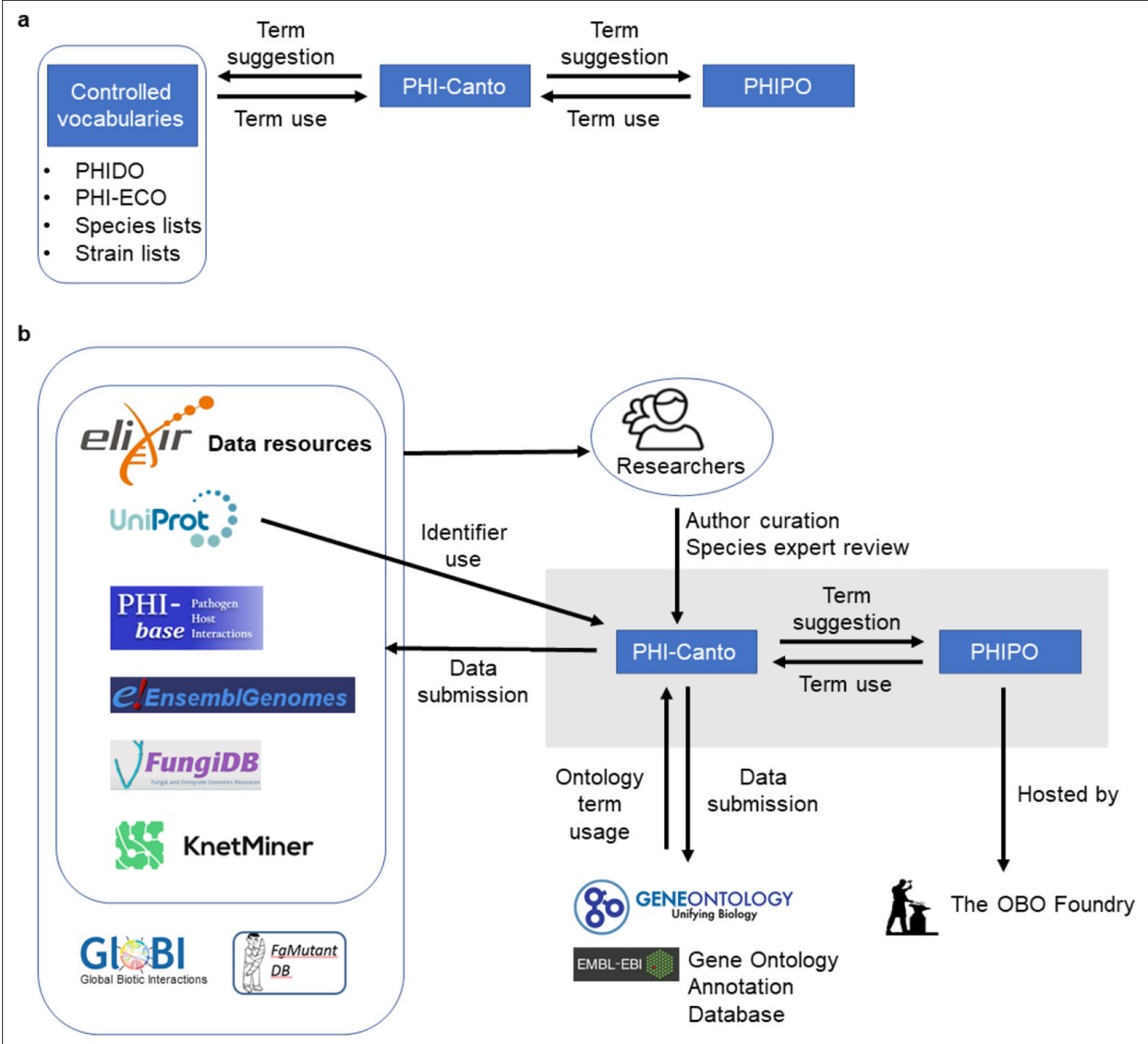

**Figure 6.** The interspecies curation framework and the interoperability of the Pathogen–Host Interaction Community Annotation Tool (PHI-Canto). (a) The interspecies curation framework consists of three main components. First, a curation tool called PHI-Canto, second, a new species-neutral phenotype ontology called PHIPO (the Pathogen–Host Interaction Phenotype Ontology), and thirdly, a selection of additional controlled vocabularies for disease names (PHIDO), experimental conditions (PHI-ECO), pathogen and host species, and natural strains associated with each species. The two-way arrows indicate that terms from the ontology and controlled vocabularies are used in curation with PHI-Canto, and that new terms required for curation may be suggested for inclusion within the ontology and controlled vocabularies. (b) The PHI-Canto and PHIPO content curation framework (gray box) uses persistent identifiers and cross-referenced information from UniProt, Ensembl Genomes, and the Gene Ontology. PHIPO is made available at the OBO Foundry. Newly minted wild-type gene annotations are suggested for inclusion into the Gene Ontology via the EBI Gene Ontology Annotation database. Data curated in PHI-Canto, following expert review, is then shared with ELIXIR data resources such as UniProtKB, Ensembl Genomes, FungiDB, and KnetMiner, and provided on request to other databases (FgMutantDB, GloBI). Researchers can look up curated information via the Pathogen–Host Interactions database (PHI-base) web interface or can download the whole dataset from PHI-base for inclusion in their bioinformatics pipelines. Authors can submit data to PHI-base by curating their publications into PHI-Canto. The origin of data is indicated by directional arrows.

PHI-base is an example of a freely available, manually curated database, which has been curating literature using professional curators since 2005 (*Winnenburg et al., 2006*).

Here, we have described the development of PHI-Canto to allow the curation of the interspecies pathogen–host interaction literature by professional curators and publication authors. This curated data is then made available on the new gene-centric version 5 of PHI-base, where all information (i.e.

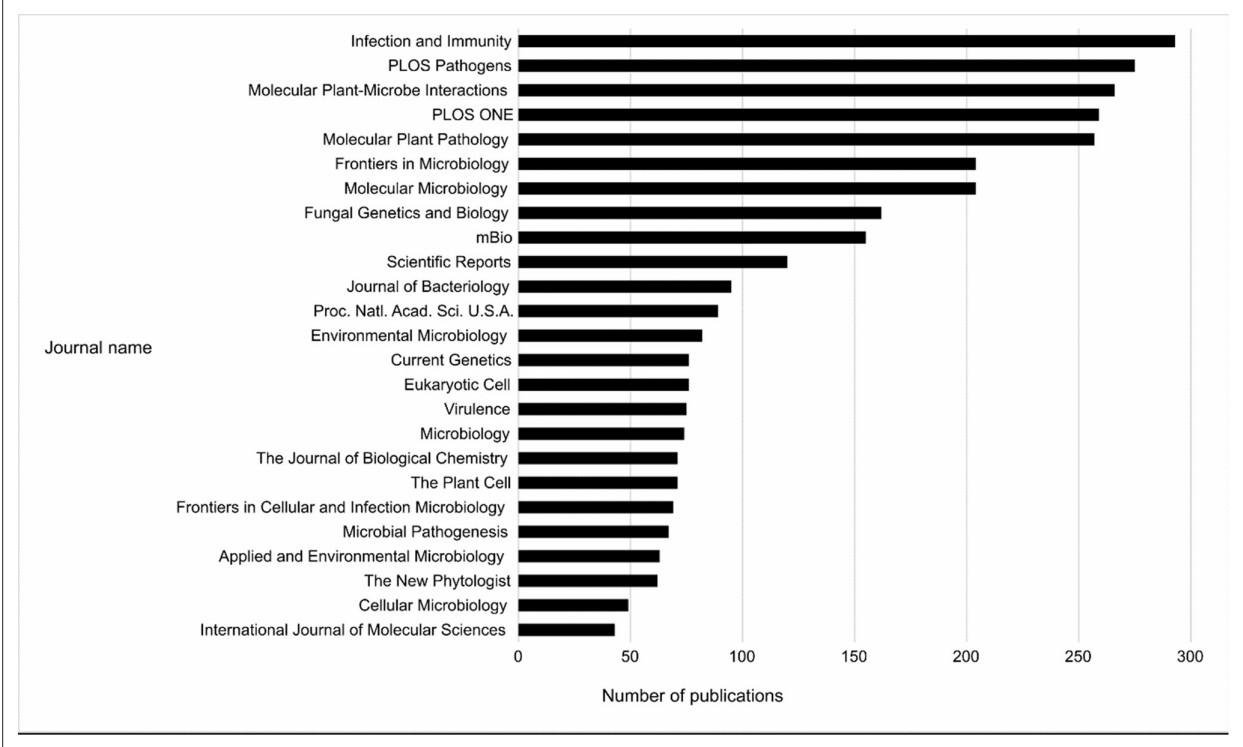

**Figure 7.** Top 25 Journals in the Pathogen–Host Interactions database (PHI-base). Bar chart showing the top 25 journals by number of publications curated in PHI-base, as of version 4.13 (published May 9, 2022). Publication counts were generated by extracting every unique PubMed identifier (PMID) from PHI-base, then using the Entrez Programming Utilities (E-Utilities) to retrieve the journal name for each PMID, and finally summing the count of journal names. The total number of journals in version 4.13 of PHI-base was 291.

new and existing) on a single gene from several publications is presented on a single page, with links to external resources providing information on interacting genes, proteins, and other entities.

Several adaptations to the original single-species community annotation tool, Canto (*Rutherford et al., 2014*), were required to convert this tool for interspecies use. Notably, the need to annotate an interaction involving two different organisms necessitated the development of a novel concept, the 'metagenotype' (*Figure 3*), in order to record a combined experimental genotype involving both a pathogen and a host. This is, to our knowledge, the first example of such an approach to interspecies interaction curation.

Curation of pathogen–host interactions in PHI-Canto also necessitated the development of a new phenotype ontology (PHIPO) to annotate pathogen–host interaction phenotypes in sufficient detail across the broad range of host species that were curated in PHI-base ($n$=234 in version 4.14 of PHI-base). The functional annotation of genes involved in interspecies interactions is a complex and challenging task, requiring ongoing modifications to the Gene Ontology and occasional major refactoring to deprecate legacy terms (*Carbon et al., 2021*). PHIPO development and maintenance will also be an ongoing task, with both authors and professional curators requesting new terms and edits to existing terms and the ontology structure. Maintenance will be made more sustainable by the incorporation of logical definitions that are aligned across phenotype ontologies in collaboration with the uPheno project (*Shefchek et al., 2020*).

To improve the efficiency of the curation process, we are suggesting that authors follow an author checklist during manuscript preparation (Appendix 3). This will improve the presentation of key information (e.g. species names, gene identifiers, etc.) in published manuscripts, thus enabling more efficient and comprehensive curation that is human- and machine-readable. The annotation procedures described here using PHI-Canto can be used to extract data buried in small-scale publications and increase the accessibility of the curated article to a wider range of potential users, for example, computational biologists, thereby improving the FAIR status of the data. The current data in PHI-base has been obtained from >200 journals (*Figure 7*) and, therefore, represents highly fragmented

knowledge which is exceptionally difficult to use by professionals in other disciplines. The feasibility of scalable community curation with Canto is evidenced by PomBase (**Lock et al., 2020**), where Canto *S. pombe* annotations from over 1000 publications are provided by publication authors, with the data made available within 24 hr of review (https://curation.pombase.org/pombe/stats/annotation).

With regard to our focus on manual curation, we recognize that great progress has been made with machine learning (ML) approaches in recent times. However, **Wood et al., 2022** note that the data being curated from publications are 'categorical, highly complex, and with hundreds of thousands of heterogeneous classes, often not explicitly labeled.' There are no published examples of ML approaches outperforming an expert curator in accuracy, which is paramount in the medical field. However, curation by experts could provide a highly reliable corpus that could be used for training ML systems. Our aspiration is that ML and expert curators can collaborate in a virtuous cycle whereby expert curators continually review and refine the ML models, while the manual work of finding publications and entity recognition is handled by the ML system.

Our future intentions are twofold: first, a graph-based representation of the data will be enabled by integration with knowledge network generation tools, such as Knetminer (**Hassani-Pak et al., 2021**), where subgraphs of the knowledge graph could be embedded into each gene-centric page on the PHI-base 5 website. Second, within PHI-Canto, we intend to address the issues associated with maximizing the inherent value of the natural sequence variation between species strains, and the associated altered phenotypic outcomes observed at multiple scales, in different types of interactions and/or environments. PHI-base already contains information on numerous species with multiple experimental strains, and natural sequence variation between strains can result in alterations at the genome level that affect the subsequently observed phenotypes. Strain-specific sequence variation is not captured in the reference proteomes stored by UniProt, even though accession numbers from these proteomes are often used in PHI-Canto. Currently, when a curator enters a gene with a taxonomic identifier below the species rank, PHI-Canto maps the identifier to the corresponding identifier at the species rank (thus removing any strain details from the organism name), and the curator specifies a strain to differentiate gene variants in naturally occurring strains. However, this does not change the taxonomic identifier linked to the UniProtKB accession number (nor its sequence), so the potential for inaccuracy remains. To mitigate this, the future plan is to record the strain-specific sequence of the gene using an accession number from a database from the International Nucleotide Sequence Database Collaboration (**Arita et al., 2021**).

The release of PHI-Canto to the community will occur gradually through various routes. Community curation will be promoted by working with journals to capture the publication data at the source, at the point of manuscript acceptance. We will also target specific research communities (e.g. those working on a particular pathogen and/or research topic) by inviting authors to curate their own publications. Authors may contact us directly to request support while curating their publications in PHI-Canto.

PHI-Canto, PHI-base, and PHIPO were devised and built over the past seven years to serve the research needs of a specific international research community interested in exploring the wide diversity of common and species-specific mechanisms underlying pathogen attack and host defense in plants, animals, humans, and other host organisms caused by fungi, protists and bacteria. However, it should be noted that the underlying developments to Canto's data model – especially the concept of annotating metagenotypes – could be of use to communities focused on different types of interspecies interactions. Possible future uses of the PHI-Canto schema could include insect–plant interactions (both beneficial and detrimental), endosymbiotic relationships such as mycorrhiza–plant rhizosphere interactions, nodulating bacteria–plant rhizosphere interactions, fungi–fungi interactions, plant–plant interactions or bacteria–insect interactions, and non-pathogenic relationships in natural environments such as bulk soil, rhizosphere, phyllosphere, air, freshwater, estuarine water or seawater, and human–animal, animal–bird, human–insect, animal–insect, bird–insect interactions in various anatomical locations (e.g. gut, lung, and skin). The schema could also be extended to situations where phenotype–genotype relations have been established for predator–prey relationships or where there is competition in herbivore–herbivore, predator–predator or prey–prey relationships in the air, on land, or in the water. Finally, the schema could be used to explore strain-to-strain interactions within a species when different biological properties have been noted. Customizing Canto to use other ontologies and controlled vocabularies is as simple as editing a configuration file, as shown in Source code 1.

## Methods

### Changes to the Canto data model and configuration

PHI-Canto stores its data in a series of relational databases using the SQLite database engine. A primary database stores data shared across all curation sessions, and each curation session also has its own database to store data related to a single publication (such as genes, genotypes, metagenotypes, etc.). PHI-Canto can export its data as a JSON file or in more specialized formats, for example, the GO Annotation File (GAF) format.

To implement PHI-Canto several new entities were added to the Canto data model in order to support pathogen–host curation, as well as new configuration options (the new entities are illustrated in *Figure 3—figure supplement 1*). These entities were 'strain,' 'metagenotype,' and 'metagenotype annotation.' The complete data model for PHI-Canto is illustrated in *Figure 3—figure supplements 2 and 3*.

### Pathogen and host roles

Genotype entities in PHI-Canto's data model were extended with an attribute indicating their status as a pathogen genotype or a host genotype. Genotypes inherit their status (as pathogen or host) from the organism, which in turn is classified as a pathogen or host based on a configuration file that contains the NCBI Taxonomy ID (taxid) (*Schoch et al., 2020*) of each host species in PHI-base. Only host taxids need to be specified since PHI-Canto defaults to classifying a species as a pathogen if its taxid is not found in the configuration file.

PHI-Canto also loads lists of pathogen and host species that specify the scientific name, taxid, and common name (if any) of each species. These species lists are used to specify which host species can be added as a component of the metagenotype in the absence of a specific studied gene, and to override the scientific name provided by UniProtKB in favor of the name used by a scientific community studying the species (for example, to control whether the anamorph or teleomorph name of a fungal species is displayed in PHI-Canto's user interface).

### Metagenotype implementation

Metagenotypes were implemented by adding a 'metagenotype' entity to PHI-Canto's data model. The metagenotype is the composition of two genotype entities. We also changed the data model to allow annotations to be related to metagenotypes (previously, only genes and genotypes could be related to annotations).

### Strain implementation

Support for strain curation was implemented by adding a 'strain' entity to PHI-Canto's data model. Strains are related to an organism entity and its related genotype entities. In the user interface, PHI-Canto uses the taxid of the organism to filter an autocomplete system, such that only the strains of the specified organism are suggested. The autocomplete system can also use synonyms in the strain list to suggest a strain based on its synonymous names. Unknown strains are represented by a preset value of 'Unknown strain.'

### Ontologies

PHIPO was developed using the Protégé ontology editor (*Musen and Team, 2015*). PHIPO uses OBO namespaces to allow PHI-Canto to filter the terms in the ontology by annotation type, ensuring that genotypes are annotated with single-species phenotypes and metagenotypes with pathogen–host interaction phenotypes.

PHI-ECO was also developed using Protégé, starting from a list of experimental conditions originally developed by PomBase. PHIDO was initially derived from a list of diseases already curated in PHI-base and is now maintained as a flat file that is converted into an OBO file using ROBOT (*Jackson et al., 2019*).

### Data availability

Pathogen–Host Interaction Phenotype Ontology: http://purl.obolibrary.org/obo/phipo.owl.

PHI-base: Experimental Conditions Ontology: (*Cuzick and Seager, 2022a*) https://github.com/PHI-base/phi-eco.

PHIDO: the controlled vocabulary of disease names: (*Cuzick and Seager, 2022b*) https://github.com/PHI-base/phido.

PHIPO Extension Ontology for gene-for-gene phenotypes: (*Cuzick and Seager, 2022c*) https://github.com/PHI-base/phipo_ext.

Location of species and strain lists used by PHI-Canto: (*Cuzick et al., 2022d*) https://github.com/PHI-base/data.

PHI-Canto approved curation sessions (December 2022): https://doi.org/10.5281/zenodo.7428788.

## Code availability

PHI-Canto's source code is available on GitHub, at https://github.com/PHI-base/canto, (copy archived at swh:1:rev:dd310334974d9471c1916c0ac080550bfd153707). PHI-Canto is freely licensed under the GNU General Public License version 3, with no restrictions on copying, distributing, or modifying the code, for commercial use or otherwise, provided any derivative works are licensed under the same terms. PHI-base provides an online demo version of PHI-Canto at https://demo-canto.phi-base.org/ which can be used for evaluating the tool. The demo version and the main version of PHI-Canto will remain freely available online.

Canto's source code is available on GitHub, at https://github.com/pombase/canto, (copy archived at swh:1:rev:2f8fe11c217b52a69251cb589abdf798dab3767b). Canto is also freely licensed under the GNU General Public License version 3.

The source code for PHI-Canto's user documentation is available on GitHub, at https://github.com/PHI-base/canto-docs, (copy archived at swh:1:rev:a134c04d8fb59769678456fb41d02fd169be7b06). The user documentation is licensed under the MIT license. The published format of the user documentation is available online at https://canto.phi-base.org/docs/index.

The source code for PHIPO is available on GitHub under a Creative Commons Attribution 3.0 license, at https://github.com/PHI-base/phipo, (copy archived at swh:1:rev:fbb0af482869744e085e829c463d4eb0c6afafd2).

## Acknowledgements

We thank the late post-doctoral PHI-base team member Dr. Alistair Irvine for adding chemical entries to ChEBI. Dr. Paul Kersey, formerly the non-vertebrate Ensembl team leader, is thanked for helpful discussions and ideas on community engagement. We thank Dr. Midori Harris (formerly of the University of Cambridge, UK) for providing valuable input into the development of PHIPO based on her extensive knowledge of FYPO. Dr. Pascale Gaudet (Swiss-Prot, Swiss Institute of Bioinformatics) is thanked for the generation and editing of GO terms involved in interspecies interactions. We also thank Drs. Chris Stephens and Ana Machado-Wood (both formerly of Rothamsted Research) for completing the trial curation of articles into beta versions of PHI-Canto and providing invaluable feedback and suggestions for further improvements. We thank Dr. Melina Velasquez (based at Rothamsted Research) for preparing the PHI-Canto tutorial videos. The Molecular Connections team based in Bangalore India while developing the PHI-base 5 website, provided useful feedback on data interoperability between PHI-Canto and the new gene-centric version of PHI-base. PHI-base is funded by the UK Biotechnology and Biological Sciences Research Council (BBSRC) Grants BB/S020020/1 and BB/S020098/1 and previously by the BBSRC National Capability Grant (2012–2017). Rothamsted authors MU and KHK receive additional BBSRC grant-aided support as part of the Institute Strategic Programme (ISP) Designing Future Wheat Grant (BB/P016855/1) and Delivering Sustainable Wheat (DSW) (BB/X011003/1). In addition, author AC receives BBSRC ISP DSW (BB/X011003/1) support and authors AC and JS receive BBSRC ISP Growing Health (BB/X010953/1) support. This work was conducted using the Protégé resource, which is supported by grant GM10331601 from the National Institute of General Medical Sciences of the United States National Institutes of Health.

# Additional information

## Funding

| Funder | Grant reference number | Author |
|---|---|---|
| Biotechnology and Biological Sciences Research Council | BB/S020020/1 | Alayne Cuzick<br>Martin Urban<br>Kim E Hammond-Kosack<br>James Seager |
| Biotechnology and Biological Sciences Research Council | BB/S020098/1 | Alayne Cuzick<br>Martin Urban<br>Kim E Hammond-Kosack |
| Biotechnology and Biological Sciences Research Council | BB/X011003/1 | Alayne Cuzick<br>Martin Urban<br>Kim E Hammond-Kosack |
| Biotechnology and Biological Sciences Research Council | BB/X010953/1 | Alayne Cuzick<br>James Seager |
| Biotechnology and Biological Sciences Research Council | BB/P016855/1 | Martin Urban<br>Kim E Hammond-Kosack |

The funders had no role in study design, data collection and interpretation, or the decision to submit the work for publication.

## Author contributions

Alayne Cuzick, Conceptualization, Data curation, Validation, Investigation, Visualization, Methodology, Writing – original draft, Writing – review and editing; James Seager, Resources, Software, Validation, Visualization, Methodology, Writing – review and editing; Valerie Wood, Data curation, Supervision, Validation, Investigation, Methodology, Writing – review and editing; Martin Urban, Resources, Visualization, Methodology, Writing – review and editing; Kim Rutherford, Resources, Software, Supervision, Methodology, Writing – review and editing; Kim E Hammond-Kosack, Conceptualization, Supervision, Funding acquisition, Methodology, Project administration, Writing – review and editing

## Author ORCIDs

Alayne Cuzick http://orcid.org/0000-0001-8941-3984
James Seager http://orcid.org/0000-0001-7487-610X
Valerie Wood http://orcid.org/0000-0001-6330-7526
Martin Urban http://orcid.org/0000-0003-2440-4352
Kim Rutherford http://orcid.org/0000-0001-6277-726X
Kim E Hammond-Kosack http://orcid.org/0000-0002-9699-485X

## Decision letter and Author response

Decision letter https://doi.org/10.7554/eLife.84658.sa1
Author response https://doi.org/10.7554/eLife.84658.sa2

# Additional files

## Supplementary files

• Supplementary file 1. Mapping display name to relation name for Annotation Extensions in the Pathogen–Host Interaction Community Annotation Tool (PHI-Canto).

• Supplementary file 2. PomBase annotation extensions used in the Pathogen–Host Interaction Community Annotation Tool (PHI-Canto).

• Supplementary file 3. Pathogen–Host Interactions database (PHI-base) nine high-level term mapping to the Pathogen–Host Interaction Community Annotation Tool (PHI-Canto).

• Supplementary file 4. The Pathogen–Host Interaction Community Annotation Tool (PHI-Canto) species and strain lists for pathogens and hosts.

• Supplementary file 5. Mapping between strains in the Pathogen–Host Interactions database (PHI-

base) and the Pathogen–Host Interaction Community Annotation Tool (PHI-Canto).

• MDAR checklist

• Source code 1. Main configuration file for the Pathogen–Host Interaction Community Annotation Tool (PHI-Canto). This is the main configuration file for PHI-Canto. Much of the configuration is inherited from Canto, the original curation application from which PHI-Canto is derived. Lines containing custom configuration for PHI-Canto have been indicated with comments

### Data availability

Datasets generated for use within the curation framework are available as GitHub links in the manuscript section 'Data availability'. Code is available as GitHub links in the manuscript section 'Code availability'. PHI-Canto curated data is available here https://doi.org/10.5281/zenodo.7428788.

The following dataset was generated:

| Author(s) | Year | Dataset title | Dataset URL | Database and Identifier |
|---|---|---|---|---|
| Cuzick A, Wood V, Velasquez M, Wilkes JM | 2022 | PHI-Canto approved curation sessions: December 2022 | https://doi.org/10.5281/zenodo.7428788 | Zenodo, 10.5281/zenodo.7428788 |

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

## Appendix 1

### How to use annotation extensions

This file provides information on Annotation Extensions (AE) and how to use them in PHI-Canto to curate a standard selection of experiments (*Table 2*). The first section provides four examples of using AEs for curating metagenotypes with pathogen-host interaction phenotypes. The second section provides examples of curating metagenotypes using the gene-for-gene phenotype workflow, including using the AEs for gene-for-gene interactions and inverse gene-for-gene interactions. The third section of this file illustrates three examples of using AEs for curating single-species phenotypes.

Further information on how to use PHI-Canto to make annotations can be found in PHI-Canto's user documentation, available at https://canto.phi-base.org/docs/index.

Contents:

SECTION 1: Annotation Extensions for curating pathogen-host interaction phenotypes on metagenotypes

- Section 1A: If you have a metagenotype phenotype recording 'unaffected pathogenicity' (corresponds to footnote ‡ in *Table 2*)
- Section 1B: If you have a metagenotype phenotype recording 'altered pathogenicity or virulence' (corresponds to footnote § in *Table 2*)
- Section 1C: If you have a metagenotype phenotype recording 'mutualism' (corresponds to footnote ** in *Table 2*)
- Section 1D: If you have a metagenotype phenotype recording 'a pathogen effector' (corresponds to footnote †† in *Table 2*)

SECTION 2: Annotation Extensions for curating gene-for-gene phenotypes on metagenotypes

- Section 2A: If you have a metagenotype phenotype recording 'a gene-for-gene interaction' (corresponds to footnote ‡‡ in *Table 2*)
- Section 2B: If you have a metagenotype phenotype recording 'an inverse gene-for-gene interaction' (corresponds to footnote ¶¶ in *Table 2*)

SECTION 3: Annotation Extensions for curating single species phenotypes (pathogen phenotypes or host phenotypes)

- Section 3A: Example of an *in vitro* pathogen phenotype (corresponds to footnote ¶ in *Table 2*)
- Section 3B: Example of an *in vitro* pathogen chemistry phenotype (corresponds to footnote *** in *Table 2*)
- Section 3C: Example of an *in vivo* host phenotype (corresponds to footnote §§ in *Table 2*)

## Section 1: Annotation Extensions for curating pathogen-host interaction phenotypes on metagenotypes

When creating and annotating metagenotypes, it is advisable to also create and annotate a wild-type control metagenotype where possible. This enables a better understanding of annotations made to altered metagenotypes.

(Note: It is also possible to use several of the AEs in the table documenting single species phenotype AEs, e.g. *penetrance* and *affected protein*).

## Section 1 A: If you have a metagenotype phenotype recording 'unaffected pathogenicity' (corresponds to footnote ‡ in *Table 2*)

**Appendix 1—table 1.** Annotation extensions (AE) summary for 'unaffected pathogenicity'.

| AE name | Cardinality | Available terms |
| --- | --- | --- |
| compared to control genotype | 0, 1 | Metagenotype identifier |
| extent of infectivity | 0, 1 | 'unaffected pathogenicity' |
| host tissue affected | 0, $n$ | BRENDA Tissue Ontology term |
| outcome of interaction | 0, 1 | 'disease present,' 'disease absent' |

Example publication: The RhlR quorum-sensing receptor controls *Pseudomonas aeruginosa* pathogenesis and biofilm development independently of its canonical homoserine lactone autoinducer (PMID:28715477).

## Control metagenotype

| Pathogen genotype | Host genotype | Term ID | Term name | Evidence code | Conditions | Figure | Annotation extension |
|---|---|---|---|---|---|---|---|
| rhlI+[WT level] *P. aeruginosa* (PA14) | wild type *C. elegans* (N2) | PHIPO:0001069 | death of host organism with pathogen | Cell growth assay | agar plates | Fig 6a | **infects_tissue** whole body , **has_penetrance** 50% , **interaction_outcome** disease present |

## Altered metagenotype

| Pathogen genotype | Host genotype | Term ID | Term name | Evidence code | Conditions | Figure | Annotation extension |
|---|---|---|---|---|---|---|---|
| rhlIΔ *P. aeruginosa* (PA14) | wild type *C. elegans* (N2) | PHIPO:0001069 | death of host organism with pathogen | Cell growth assay | agar plates | Fig 6a | **infects_tissue** whole body , **infective_ability** unaffected pathogenicity , **has_penetrance** 50% , **compared_to_control** rhlI+[WT level] Pseudomonas aeruginosa (PA14) / wild type Caenorhabditis elegans (N2) , **interaction_outcome** disease present |

**Appendix 1—figure 1.** Pathogen-host interaction phenotype for 'unaffected pathogenicity'. Note: Phenotype annotations use evidence codes modeled on the Evidence & Conclusion Ontology (ECO). Evidence code 'Cell growth assay' corresponds to 'cell growth assay evidence' (ECO:0001563).

## Section 1B: If you have a metagenotype phenotype recording 'altered pathogenicity or virulence' (corresponds to footnote § in *Table 2*)

**Appendix 1—table 2.** Annotation extensions (AE) summary for 'altered pathogenicity or virulence'.

| AE name | Cardinality | Available terms |
|---|---|---|
| compared to control genotype | 0, 1 | Metagenotype identifier |
| extent of infectivity | 0, 1 | 'loss of pathogenicity,' 'reduced virulence,' 'increased virulence' |
| host tissue affected | 0, *n* | BRENDA Tissue Ontology term |
| outcome of interaction | 0, 1 | 'disease present,' 'disease absent' |

Example publication: A conserved fungal glycosyltransferase facilitates pathogenesis of plants by enabling hyphal growth on solid surfaces (PMID:29020037).

A training video is available for the curation of this publication at https://youtu.be/44XGoi6Ijqk?t=1738.

**Control metagenotype**

| Pathogen genotype | Host genotype | Term ID | Term name | Evidence code | Conditions | Figure | Annotation extension |
|---|---|---|---|---|---|---|---|
| GT2+[WT level] *Z. tritici* (IPO323) | wild type *T. aestivum* (cv. Riband) | PHIPO:0000480 | presence of pathogen-associated host lesions | Macroscopic observation (qualitative observation) | 14 days post inoculation | Figure 2E | **infects_tissue** leaf , **interaction_outcome** disease present |

**Altered metagenotype**

| Pathogen genotype | Host genotype | Term ID | Term name | Evidence code | Conditions | Figure | Annotation extension |
|---|---|---|---|---|---|---|---|
| ΔGT2-19(deletion) *Z. tritici* (IPO323) | wild type *T. aestivum* (cv. Riband) | PHIPO:0000481 | absence of pathogen-associated host lesions | Macroscopic observation (qualitative observation) | 14 days post inoculation | Figure 2E | **infects_tissue** leaf , **infective_ability** loss of pathogenicity , **compared_to_control** GT2+[WT level] Zymoseptoria tritici (IPO323) / wild type Triticum aestivum (cv. Riband) , **interaction_outcome** disease absent |

**Appendix 1—figure 2.** Pathogen-host interaction phenotype for 'altered pathogenicity or virulence'. Note: Phenotype annotations use evidence codes modeled on the Evidence & Conclusion Ontology (ECO). Evidence code 'Macroscopic observation (qualitative observation)' corresponds to the new ECO term 'qualitative macroscopy evidence' (ECO:0006342).

## Section 1 C: If you have a metagenotype phenotype recording 'mutualism' (corresponds to footnote ** in *Table 2*)

**Appendix 1—table 3.** Annotation extensions (AE) summary for 'mutualism'.

| AE name | Cardinality | Available terms |
|---|---|---|
| compared to control genotype | 0, 1 | Metagenotype identifier |
| extent of infectivity | 0, 1 | 'mutualism present,' 'mutualism absent,' 'loss of mutualism' |
| host tissue affected | 0, $n$ | BRENDA Tissue Ontology term |

Note: The 'Outcome of interaction' AE is not relevant in this mutualism interaction.

Example publication: Reactive oxygen species play a role in regulating a fungus-perennial ryegrass mutualistic interaction (PMID:16517760).

**Control metagenotype**

| Pathogen genotype | Host genotype | Term ID | Term name | Evidence code | Figure | Annotation extension |
|---|---|---|---|---|---|---|
| noxA+[WT level] *E. festucae* (Fl1) **bkg:** GFP | wild type *L. perenne* (Unknown strain) | PHIPO:0000954 | presence of pathogen growth within host | Microscopy | Figure 1c | **infects_tissue** leaf , **infective_ability** mutualism present |

**Altered metagenotype**

| Pathogen genotype | Host genotype | Term ID | Term name | Evidence code | Figure | Annotation extension |
|---|---|---|---|---|---|---|
| noxA::pAN7-1(disruption)[Not assayed] *E. festucae* (Fl1) **bkg:** GFP | wild type *L. perenne* (Unknown strain) | PHIPO:0000368 | increased pathogen growth within host | Microscopy | Figure 1d | **infects_tissue** leaf , **infective_ability** loss of mutualism , **compared_to_control** noxA+[WT level] Epichloe festucae (Fl1) / wild type Lolium perenne (Unknown strain) |

**Appendix 1—figure 3.** Pathogen-host interaction phenotype: Example 1 Illustrating a phenotype associated with the pathogen component within the Pathogen-Host Interaction. Note: Phenotype annotations use evidence codes modeled on the Evidence & Conclusion Ontology (ECO). Evidence code 'Microscopy' corresponds to 'microscopy evidence' (ECO:0001098).

## Control metagenotype

| Pathogen genotype | Host genotype | Term ID | Term name | Evidence code | Figure | Annotation extension |
|---|---|---|---|---|---|---|
| noxA+[WT level] *E. festucae* (Fl1) | wild type *L. perenne* (Unknown strain) | PHIPO:0001005 | normal host morphology during pathogen invasion | Macroscopic observation (qualitative observation) | Figure 1a, 5c | **infects_tissue** whole plant , **infective_ability** mutualism present |

## Altered metagenotype

Note: in this case, two separate annotations were made to the same metagenotype.

| Pathogen genotype | Host genotype | Term ID | Term name | Evidence code | Figure | Annotation extension |
|---|---|---|---|---|---|---|
| noxA::pAN7-1(disruption)[Not assayed] *E. festucae* (Fl1) | wild type *L. perenne* (Unknown strain) | PHIPO:0001130 | stunted host growth during pathogen colonization | Macroscopic observation (qualitative observation) | Figure 1a | **infects_tissue** whole plant , **infective_ability** loss of mutualism , **compared_to_control** noxA+[WT level] Epichloe festucae (Fl1) / wild type Lolium perenne (Unknown strain) |

| Pathogen genotype | Host genotype | Term ID | Term name | Evidence code | Figure | Annotation extension |
|---|---|---|---|---|---|---|
| noxA::pAN7-1(disruption)[Not assayed] *E. festucae* (Fl1) | wild type *L. perenne* (Unknown strain) | PHIPO:0001131 | increased number of host side shoots during pathogen colonization | Macroscopic observation (qualitative observation) | Figure 1a | **infects_tissue** whole plant , **infective_ability** loss of mutualism , **compared_to_control** noxA+[WT level] Epichloe festucae (Fl1) / wild type Lolium perenne (Unknown strain) |

**Appendix 1—figure 4.** Pathogen-host interaction phenotype: Example 2 Illustrating a phenotype associated with the <u>host</u> component within the Pathogen-Host Interaction.

## Section 1D: If you have a metagenotype phenotype recording 'a pathogen effector' (corresponds to footnote †† in *Table 2*)

If you have a biotrophic or necrotrophic plant pathogen effector which is involved in a gene-for-gene interaction, please see the AEs for the 'gene-for-gene interaction' or 'inverse gene-for-gene interaction' workflow (Section 2).

Annotate the pathogen effector with the GO Biological Process term 'effector-mediated modulation of host process by symbiont' (GO:0140418) or a descendant. If the GO Molecular Function term is known, then this can also be annotated and linked to the relevant GO effector term via an annotation extension.

**Appendix 1—table 4.** Annotation extensions (AE) summary for 'a pathogen effector'.

| AE name | Cardinality | Available terms |
|---|---|---|
| compared to control genotype | 0, 1 | Metagenotype identifier |
| extent of infectivity | 0, 1 | 'unaffected pathogenicity,' 'loss of pathogenicity,' 'reduced virulence,' 'increased virulence' |
| host tissue affected | 0, *n* | BRENDA Tissue Ontology term |
| outcome of interaction | 0, 1 | 'disease present,' 'disease absent' |

Example publication: An effector protein of the wheat stripe rust fungus targets chloroplasts and suppresses chloroplast function (PMID:31804478).

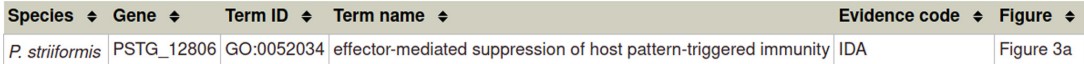

| Species | Gene | Term ID | Term name | Evidence code | Figure |
|---|---|---|---|---|---|
| *P. striiformis* | PSTG_12806 | GO:0052034 | effector-mediated suppression of host pattern-triggered immunity | IDA | Figure 3a |

**Appendix 1—figure 5.** Gene Ontology (GO) biological process annotation for 'a pathogen effector'. Note: 'Effector-mediated suppression of host pattern-triggered immunity' (GO:0052034) is a descendant term of 'effector-mediated modulation of host process by symbiont' (GO:0140418). Note: GO annotations use GO evidence codes (http://geneontology.org/docs/guide-go-evidence-codes/).

| Species ⇕ | Gene ⇕ | Term ID ⇕ | Term name ⇕ | Evidence code ⇕ | With | Figure ⇕ | Annotation extension ⇕ |
|---|---|---|---|---|---|---|---|
| *P. striiformis* | PSTG_12806 | GO:0005515 | protein binding | IPI | petC | Figure 5 | **part_of** effector-mediated suppression of host pattern-triggered immunity |
| *P. striiformis* | PSTG_12806 | GO:0004857 | enzyme inhibitor activity | IPI | petC | Figure 5 | **has_regulation_target** petC , **occurs_at** host cell chloroplast , **part_of** effector-mediated suppression of host pattern-triggered immunity |

**Appendix 1—figure 6.** Gene Ontology (GO) molecular function annotation for 'a pathogen effector'. Please note that in the case of a physical interaction (protein–protein interaction) between the pathogen and host gene products (PSTG_12806 and PetC in the example above, respectively) this information can be curated using the Physical Interaction curation workflow, documented in https://canto.phi-base.org/docs/physical_interaction_ annotation.

### Altered metagenotype

| Pathogen genotype | Host genotype | Term ID ⇕ | Term name ⇕ | Evidence code ⇕ | Conditions | Figure ⇕ | Annotation extension ⇕ |
|---|---|---|---|---|---|---|---|
| Pst_12806ΔSP(1-23)[Not assayed] *P. striiformis* (f. sp. tritici strain CYR32) | wild type *N. benthamiana* (Unknown strain) | PHIPO:0001015 | decreased level of host defense-induced callose deposition | Microscopy | delivery mechanism: agrobacterium, + PTI inducer flg22 | Figure 3a, b | **infects_tissue** leaf , **infective_ability** increased virulence |
| Pst_12806ΔSP(1-23)[Not assayed] *P. striiformis* (f. sp. tritici strain CYR32) | wild type *N. benthamiana* (Unknown strain) | PHIPO:0001128 | effector-mediated suppression of host PAMP-triggered immunity present | Microscopy | delivery mechanism: agrobacterium, + PTI inducer flg22 | Figure 3a, b, c | **infective_ability** increased virulence |

**Appendix 1—figure 7.** Pathogen-host interaction phenotypes for 'a pathogen effector'. In this case, there are no metagenotype control annotations. This is because it is not possible to create and annotate a metagenotype comprising of an empty vector control within the pathogen component of the metagenotype.

## Altered metagenotype

### Section 2: Annotation extensions for curating gene-for-gene phenotypes on metagenotypes

Section 2 A: If you have a metagenotype phenotype recording 'a gene-for-gene interaction' (corresponds to footnote ‡‡ in **Table 2**)

Annotate the pathogen effector with the GO Biological process term 'effector-mediated modulation of host process by symbiont' (GO:0140418) or a descendant. If the GO Molecular Function term is known, then this can also be annotated and linked to the relevant GO effector term via an annotation extension.

**Appendix 1—table 5.** Annotation extensions (AE) summary for 'a gene-for-gene interaction'.

| AE name | Cardinality | Available terms |
|---|---|---|
| compared to control genotype | 0, 1 | Metagenotype identifier |
| gene-for-gene phenotype | 0, 1 | 'incompatible interaction, recognizable pathogen effector present, functional host resistance gene present' 'incompatible interaction, recognizable pathogen effector present, gain of functional host resistance gene' 'incompatible interaction, gain of recognizable pathogen effector, gain of functional host resistance gene' 'incompatible interaction, gain of recognizable pathogen effector, functional host resistance gene present' 'compatible interaction, recognizable pathogen effector present, functional host resistance gene absent' 'compatible interaction, recognizable pathogen effector absent, functional host resistance gene present' 'compatible interaction, recognizable pathogen effector present, compromised host resistance gene' 'compatible interaction, recognizable pathogen effector absent, functional host resistance gene absent' 'compatible interaction, recognizable pathogen effector absent, compromised functional host resistance gene' 'compatible interaction, compromised recognizable pathogen effector, functional host resistance gene present' 'metagenotype outcome overcome by external condition' |
| host tissue affected | 0, *n* | BRENDA Tissue Ontology term |

Example publication: Activation of an *Arabidopsis* resistance protein is specified by the *in planta* association of its leucine-rich repeat domain with the cognate oomycete effector (PMID:20601497).

| Species ⬥ | Gene ⬥ | Term ID ⬥ | Term name ⬥ | Evidence code ⬥ |
|---|---|---|---|---|
| *H. arabidopsidis* | ATR1 | GO:0140418 | effector-mediated modulation of host process by symbiont | IMP |

**Appendix 1—figure 8.** Gene Ontology (GO) biological process annotation for 'a pathogen effector' within 'a gene-for-gene interaction'.

| Species ⬥ | Gene ⬥ | Term ID ⬥ | Term name ⬥ | Evidence code ⬥ | With | Figure ⬥ | Annotation extension ⬥ |
|---|---|---|---|---|---|---|---|
| *H. arabidopsidis* | ATR1 | GO:0005515 | protein binding | IPI | RPP1 | Figure 4, 5 | **part_of** effector-mediated modulation of host process by symbiont |

**Appendix 1—figure 9.** Gene Ontology (GO) molecular function annotation for 'a pathogen effector' within 'a gene-for-gene interaction'.

## Control metagenotypes

### Incompatible control

| Pathogen genotype | Host genotype | Term ID ⬥ | Term name ⬥ | Evidence code ⬥ | Conditions | Figure ⬥ | Annotation extension ⬥ |
|---|---|---|---|---|---|---|---|
| ATR1-Δ51(1-51)[Not assayed] *H. arabidopsidis* (Maks9) **bkg:** Citrine tag | RPP1+[Not assayed] *A. thaliana* (ecotype Ws-0) **bkg:** HA tag | PHIPO:0000192 | presence of host-defense induced lesion by host hypersensitive response | Macroscopic observation (qualitative observation) | delivery mechanism: agrobacterium, heterologous species tobacco | Figure 3a | **infects_tissue** leaf , **gene_for_gene_interaction** incompatible interaction, recognizable pathogen effector present, functional host resistance gene present |

### Compatible control

| Pathogen genotype | Host genotype | Term ID ⬥ | Term name ⬥ | Evidence code ⬥ | Conditions | Figure ⬥ | Annotation extension ⬥ |
|---|---|---|---|---|---|---|---|
| ATR1-Δ51(1-51)[Not assayed] *H. arabidopsidis* (Maks9) **bkg:** Citrine tag | RPP1+[Not assayed] *A. thaliana* (ecotype Nd-0) **bkg:** HA tag | PHIPO:0000182 | absence of host-defense induced lesion by host hypersensitive response | Macroscopic observation (qualitative observation) | delivery mechanism: agrobacterium, heterologous species tobacco | Figure 3b, 8b | **infects_tissue** leaf , **gene_for_gene_interaction** compatible interaction, recognizable pathogen effector absent, functional host resistance gene present |

### Altered metagenotype (shift from compatible to incompatible interaction)

| Pathogen genotype | Host genotype | Term ID ⬥ | Term name ⬥ | Evidence code ⬥ | Conditions | Figure ⬥ | Annotation extension ⬥ |
|---|---|---|---|---|---|---|---|
| ATR1-Δ51-D191G(1-51, D191G)[Not assayed] *H. arabidopsidis* (Maks9) **bkg:** Citrine tag | RPP1+[Not assayed] *A. thaliana* (ecotype Nd-0) **bkg:** HA tag | PHIPO:0000192 | presence of host-defense induced lesion by host hypersensitive response | Macroscopic observation (qualitative observation) | delivery mechanism: agrobacterium, heterologous species tobacco | Figure 8b | **infects_tissue** leaf , **gene_for_gene_interaction** incompatible interaction, gain of recognizable pathogen effector, functional host resistance gene present , **compared_to_control** ATR1-delta51(1-51)[Not assayed] Hyaloperonospora arabidopsidis (Maks9) / RPP1+[Not assayed] Arabidopsis thaliana (ecotype Nd-0) |

**Appendix 1—figure 10.** Gene-for-gene phenotype.

## Section 2B: If you have a metagenotype phenotype recording 'an inverse gene-for-gene interaction' (corresponds to footnote ¶¶ in *Table 2*)

Annotate the pathogen effector with the GO Biological process term 'effector-mediated modulation of host process by symbiont' (GO:0140418) or a descendant. If the GO Molecular Function term is known, then this can also be annotated and linked to the relevant GO effector term via an annotation extension.

**Appendix 1—table 6.** Annotation extensions (AE) summary for 'an inverse gene-for-gene interaction'.

| AE name | Cardinality | Available terms |
|---|---|---|
| compared to control genotype | 0, 1 | Metagenotype identifier |
| inverse gene-for-gene phenotype | 0, 1 | 'compatible interaction, functional pathogen necrotrophic effector present, functional host susceptibility locus present' <br> 'compatible interaction, functional pathogen necrotrophic effector present, gain of functional host susceptibility locus' <br> 'compatible interaction, gain of functional pathogen necrotrophic effector, functional host susceptibility locus present' <br> 'incompatible interaction, functional pathogen necrotrophic effector present, functional host susceptibility locus absent' <br> 'incompatible interaction, functional pathogen necrotrophic effector absent, functional host susceptibility locus present' <br> 'incompatible interaction, functional pathogen necrotrophic effector present, functional host susceptibility locus compromised' <br> 'incompatible interaction, compromised functional pathogen necrotrophic effector, functional host susceptibility locus present' <br> 'incompatible interaction, gain of functional pathogen necrotrophic effector, functional host susceptibility locus compromised' <br> 'metagenotype outcome overcome by external condition' |
| host tissue affected | 0, n | BRENDA Tissue Ontology term |

Example publication: The cysteine-rich necrotrophic effector SnTox1 produced by Stagonospora nodorum triggers susceptibility of wheat lines harboring Snn1 (PMID:22241993).

| Species ⇕ | Gene ⇕ | Term ID ⇕ | Term name ⇕ | Evidence code ⇕ |
|---|---|---|---|---|
| *P. nodorum* | Tox1 | GO:0080185 | effector-mediated induction of plant hypersensitive response by symbiont | EXP |

**Appendix 1—figure 11.** Gene Ontology (GO) biological process annotation for 'a pathogen necrotrophic effector' within 'an inverse gene-for-gene interaction'.

| Species ⇕ | Gene ⇕ | Term ID ⇕ | Term name ⇕ | Evidence code ⇕ | Annotation extension ⇕ |
|---|---|---|---|---|---|
| *P. nodorum* | Tox1 | GO:0140295 | pathogen-derived receptor ligand activity | EXP | **has_input** Snn1 , **part_of** effector-mediated induction of plant hypersensitive response by symbiont |

**Appendix 1—figure 12.** Gene Ontology (GO) molecular function annotation for 'a pathogen necrotrophic effector' within 'an inverse gene-for-gene interaction'.

**Control metagenotypes** Compatible control

| Pathogen genotype | Host genotype | Term ID ⬍ | Term name ⬍ | Evidence code ⬍ | Conditions | Figure ⬍ | Annotation extension ⬍ |
|---|---|---|---|---|---|---|---|
| Tox1+[WT level] *P. nodorum* (SN15) | Snn1+[WT level] *T. aestivum* (cv. Chinese Spring) | PHIPO:0000480 | presence of pathogen-associated host lesions | Macroscopic observation (qualitative observation) | delivery mechanism: culture infiltration | Figure 1 | **infects_tissue** leaf , **inverse_gene_for_gene** compatible interaction, functional pathogen necrotrophic effector present, functional host susceptibility locus present |

Incompatible control

| Pathogen genotype | Host genotype | Term ID ⬍ | Term name ⬍ | Evidence code ⬍ | Conditions | Figure ⬍ | Annotation extension ⬍ |
|---|---|---|---|---|---|---|---|
| Tox1-(no endogenous copy)[Not assayed] *P. nodorum* (Sn79-1087) | Snn1+[WT level] *T. aestivum* (cv. Chinese Spring) | PHIPO:0000481 | absence of pathogen-associated host lesions | Macroscopic observation (qualitative observation) | delivery mechanism: pathogen spore inoculation | Figure 5b | **infects_tissue** leaf , **inverse_gene_for_gene** incompatible interaction, functional pathogen necrotrophic effector absent, functional host susceptibility locus present |

**Altered metagenotypes** Shift from compatible to incompatible interaction

| Pathogen genotype | Host genotype | Term ID ⬍ | Term name ⬍ | Evidence code ⬍ | Conditions | Figure ⬍ | Annotation extension ⬍ |
|---|---|---|---|---|---|---|---|
| Tox1+[WT level] *P. nodorum* (SN15) | Snn1-ems237(unknown)[Not assayed] *T. aestivum* (cv. Chinese Spring) | PHIPO:0000481 | absence of pathogen-associated host lesions | Macroscopic observation (qualitative observation) | delivery mechanism: culture infiltration | Figure 1 | **infects_tissue** leaf , **compared_to_control** Tox1+[WT level] Parastagonospora nodorum (SN15) / Snn1+[WT level] Triticum aestivum (Chinese Spring) , **inverse_gene_for_gene** incompatible interaction, functional pathogen necrotrophic effector present, functional host susceptibility locus compromised |

Shift from incompatible to compatible interaction

| Pathogen genotype | Host genotype | Term ID ⬍ | Term name ⬍ | Evidence code ⬍ | Conditions | Figure ⬍ | Annotation extension ⬍ |
|---|---|---|---|---|---|---|---|
| +Sn15Tox1A1(transformant, no endogenous copy)[Not assayed] *P. nodorum* (Sn79-1087) | Snn1+[WT level] *T. aestivum* (cv. Chinese Spring) | PHIPO:0000480 | presence of pathogen-associated host lesions | Macroscopic observation (qualitative observation) | delivery mechanism: pathogen spore inoculation | Figure 5b | **infects_tissue** leaf , **compared_to_control** Tox1-(no endogenous copy)[Not assayed] Parastagonospora nodorum (Sn79-1087) / Snn1+[WT level] Triticum aestivum (Chinese Spring) , **inverse_gene_for_gene** compatible interaction, gain of functional pathogen necrotrophic effector, functional host susceptibility locus present |

No shift compared to control, still an incompatible interaction, despite alteration to both pathogen and host genotypes

| Pathogen genotype | Host genotype | Term ID ⬍ | Term name ⬍ | Evidence code ⬍ | Conditions | Figure ⬍ | Annotation extension ⬍ |
|---|---|---|---|---|---|---|---|
| +Sn15Tox1A1(transformant, no endogenous copy)[Not assayed] *P. nodorum* (Sn79-1087) | Snn1-ems237(unknown)[Not assayed] *T. aestivum* (cv. Chinese Spring) | PHIPO:0000481 | absence of pathogen-associated host lesions | Macroscopic observation (qualitative observation) | delivery mechanism: pathogen spore inoculation | Figure 5b | **infects_tissue** leaf , **compared_to_control** Tox1-(no endogenous copy)[Not assayed] Parastagonospora nodorum (Sn79-1087) / Snn1+[WT level] Triticum aestivum (Chinese Spring) , **inverse_gene_for_gene** incompatible interaction, gain of functional pathogen necrotrophic effector, functional host susceptibility locus compromised |

**Appendix 1—figure 13.** Gene-for-gene phenotype annotations for 'an inverse gene-for-gene interaction'. Note: The Annotation extensions (AEs) capture the detail of what has occurred within the pathogen-host interactions.

## Section 3: Annotation extensions for curating single species phenotypes (pathogen phenotypes or host phenotypes)

**Appendix 1—table 7.** Annotation extensions (AE) summary for 'curating single species phenotypes'.

| AE name | Cardinality | Available terms |
|---|---|---|
| affected proteins | 2 | UniProtKB accession number |
| assayed RNA | 0, 1 | UniProtKB accession number |
| assayed protein | 0, 1 | UniProtKB accession number |
| penetrance | 0, 1 | qualitative terms ('high,' 'medium,' 'low,' or 'complete') or a quantitative value (a percentage) |
| severity | 0, 1 | 'high,' 'medium,' 'low,' 'variable severity' |
| observed in organ | 0, 1 | BRENDA Tissue Ontology term |

## Section 3 A: Example of an *in vitro* pathogen phenotype (corresponds to footnote ¶ in *Table 2*)

Example publication: A conserved fungal glycosyltransferase facilitates pathogenesis of plants by enabling hyphal growth on solid surfaces (PMID:29020037).

A training video is available for the curation of this publication at https://youtu.be/44XGoi6Ijqk?t=1738.

| Species (strain) ⬍ | Genes ⬍ | Genotype (allele and expression) ⬍ | Term ID ⬍ | Term name ⬍ | Evidence code ⬍ | Conditions ⬍ | Figure ⬍ |
|---|---|---|---|---|---|---|---|
| *Z. tritici* (IPO323) | GT2 | ΔGT2-19(deletion) | PHIPO:0001212 | decreased hyphal growth | Cell growth assay | water medium, agar plates | Figure 2E |

**Appendix 1—figure 14.** Pathogen phenotype. Please note that in this curation example, no AEs were required.

## Section 3B: Example of an *in vitro* pathogen chemistry phenotype (corresponds to footnote *** in *Table 2*)

Example publication: The T788G mutation in the cyp51C gene confers voriconazole resistance in *Aspergillus flavus* causing aspergillosis. (PMID:22314539).

| Species (strain) ⬍ | Genes ⬍ | Genotype (allele and expression) ⬍ | Term ID ⬍ | Term name ⬍ | Evidence code ⬍ | Conditions ⬍ | Figure ⬍ | Annotation extension ⬍ |
|---|---|---|---|---|---|---|---|---|
| *A. flavus* (NRRL 3357) | cyp51c | cyp51C-T788G(aaS240A)[WT level] | PHIPO:0000590 | resistance to voriconazole | Cell growth assay | liquid culture, minimal medium, + voriconazole | Table 3 (footnote d), text page 2602 | **has_severity** high |
| *A. flavus* (NRRL 3357) | cyp51c | cyp51C-T161C(aaM54T)[WT level] | PHIPO:0001219 | normal growth on voriconazole | Cell growth assay | liquid culture, minimal medium, + voriconazole | text on page 2602 | |

**Appendix 1—figure 15.** Pathogen chemistry phenotype.

## Section 3 C: Example of an *in vivo* host phenotype (corresponds to footnote §§ in *Table 2*)

Example publication: Activation of an *Arabidopsis* resistance protein is specified by the *in planta* association of its leucine-rich repeat domain with the cognate oomycete effector. (PMID:20601497).

| Species (strain) ◆ | Genes ◆ | Background | Genotype (allele and expression) ◆ | Term ID ◆ | Term name ◆ | Evidence code ◆ | Conditions | Figure ◆ | Annotation extension ◆ |
|---|---|---|---|---|---|---|---|---|---|
| *A. thaliana* (ecotype Ws-0) | RPP1 | GFP | RPP1-TIR(266-1221)[Not assayed] | PHIPO:0000467 | presence of effector-independent host hypersensitive response | Macroscopic observation (qualitative observation) | delivery mechanism: agrobacterium, heterologous species tobacco | Figure 7a, c | **observed_organ** leaf |
| *A. thaliana* (ecotype Ws-0) | RPP1 | GFP | RPP1-TIRNBS(590-1221)[Not assayed] | PHIPO:0001180 | absence of effector-independent host hypersensitive response | Macroscopic observation (qualitative observation) | delivery mechanism: agrobacterium, heterologous species tobacco | Figure 7a | **observed_organ** leaf |
| *A. thaliana* (ecotype Ws-0) | RPP1 | GFP | RPP1-TIR E158A(266-1221, E158A)[Not assayed] | PHIPO:0001180 | absence of effector-independent host hypersensitive response | Macroscopic observation (qualitative observation) | delivery mechanism: agrobacterium, heterologous species tobacco | Figure 7c | **observed_organ** leaf |

**Appendix 1—figure 16.** Host phenotype.

## Appendix 2

## Worked example of a curation session

This document provides a worked example of the curation process in PHI-Canto for the publication by *King et al., 2017*, *A conserved fungal glycosyltransferase facilitates pathogenesis of plants by enabling hyphal growth on solid surfaces* (PMID:29020037).

The research study confirms the hypothesis that the *GT2* gene is required for the fungal pathogens *Zymoseptoria tritici* and *Fusarium graminearum* to cause disease in wheat (*Triticum aestivum*). The curation session in PHI-Canto captures this conclusion by annotating a pathogen–host interaction between *Z. tritici* and *T. aestivum* to show that deletion of the *GT2* gene causes loss of pathogenicity in the pathogen, and an absence of pathogen-associated lesions in the host. The wild-type interaction between *Z. tritici* and *T. aestivum* is annotated to indicate the presence of disease (and lesions), and a corresponding pathogen–host interaction between *F. graminearum* and *T. aestivum* is annotated to show that deleting *GT2* again causes a loss of pathogenicity and the absence of pathogen-associated lesions in the host.

The example starts with the entry of the publication into PHI-Canto (https://canto.phi-base.org/) and ends with the submission of the curation session for review by curators at PHI-base. The information curated from this publication is available on the new gene-centric PHI-base 5 website (http://phi5.phi-base.org, search for PHIG:308 and PHIG:307).

## Entering the publication

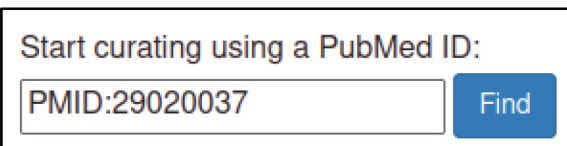

**Appendix 2—figure 1.** The Pathogen–Host Interaction Community Annotation Tool (PHI-Canto) homepage provides a text field where publications can be entered by providing their PubMed ID (PMID). The PMID in this case is 29020037.

| **Publication details** | |
|---|---|
| **ID** | PMID:29020037 |
| **Title** | A conserved fungal glycosyltransferase facilitates pathogenesis of plants by enabling hyphal growth on solid surfaces. |
| **Authors** | King R, Urban M, Lauder RP, Hawkins N, Evans M, Plummer A, Halsey K, Lovegrove A, Hammond-Kosack K, Rudd JJ |
| **Abstract** | Pathogenic fungi must extend filamentous hyphae across solid surfaces to cause diseases of plants. However, the full inventory of genes which support this is incomplete and many may be currently concealed due to their essentiality for the hyphal growth form. During a random T-DNA mutagenesis screen performed on the pleomorphic wheat (Triticum aestivum) pathogen Zymoseptoria tritici, we acquired a mutant unable to extend hyphae specifically when on solid surfaces. In contrast "yeast-like" |

**Appendix 2—figure 2.** The Pathogen–Host Interaction Community Annotation Tool (PHI-Canto) will automatically retrieve details of the publication from PubMed so that the curator can confirm that they have entered the correct PubMed ID (PMID).

**Curator details**

Before you start curating, please confirm your name and email address:

Name Martin Urban

Email martin.urban@rothamsted.ac.uk

Your ORCID (optional but recommended):

0000-0003-2440-4352

Why we collect ORCIDs

**Appendix 2—figure 3.** After accepting the publication, the curator is prompted for their name, email address, and (optionally) an ORCID ID, which are used to attribute the curation to the curator, and to contact the curator in case of problems with the curation session.

## Specifying genes and species

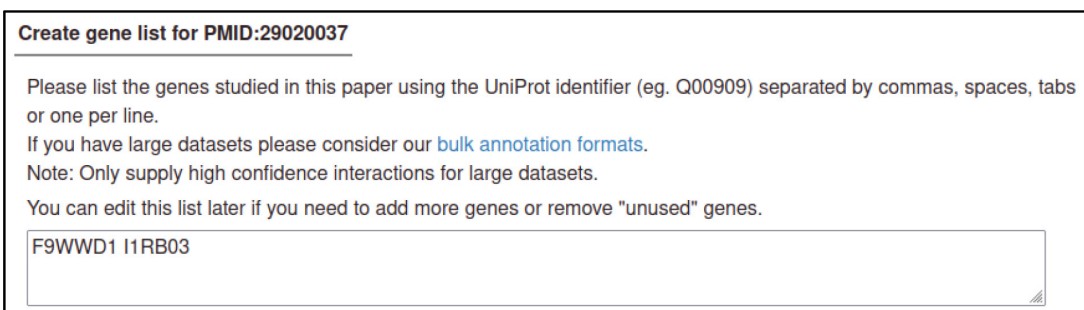

**Create gene list for PMID:29020037**

Please list the genes studied in this paper using the UniProt identifier (eg. Q00909) separated by commas, spaces, tabs or one per line.
If you have large datasets please consider our bulk annotation formats.
Note: Only supply high confidence interactions for large datasets.
You can edit this list later if you need to add more genes or remove "unused" genes.

F9WWD1 I1RB03

**Appendix 2—figure 4.** The gene is the most basic unit of annotation in the Pathogen–Host Interaction Community Annotation Tool (PHI-Canto): every other biological feature that can be annotated involves a gene, so genes are entered first. PHI-Canto uses accession numbers from the UniProt Knowledgebase (UniProtKB) to uniquely identify proteins for the genes of interest in the curated publication. The UniProtKB accession numbers for the publication are shown.

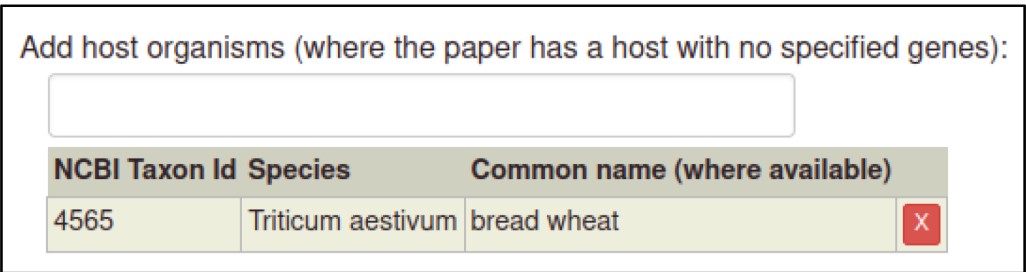

Add host organisms (where the paper has a host with no specified genes):

| NCBI Taxon Id | Species | Common name (where available) | |
|---|---|---|---|
| 4565 | Triticum aestivum | bread wheat | X |

**Appendix 2—figure 5.** Since this publication describes a wild-type host species (*T. aestivum*) with no specified genes of interest, the curator must add the host to the session by entering its NCBI Taxonomy ID in a separate field.

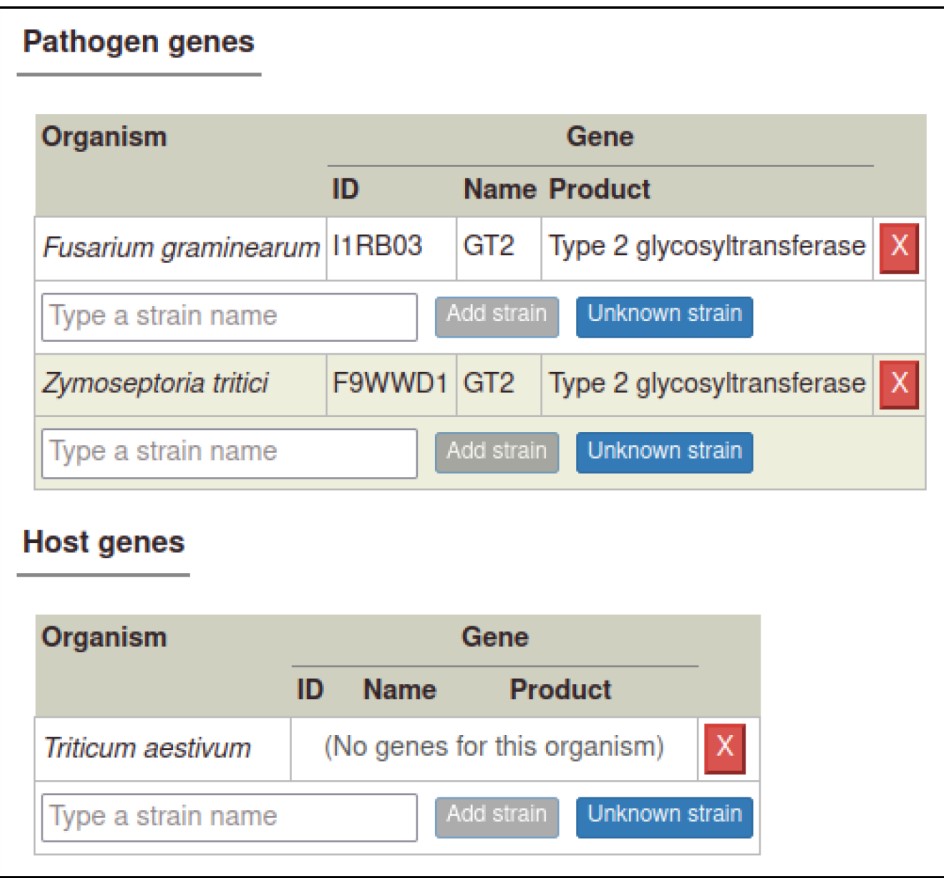

**Appendix 2—figure 6.** The Pathogen–Host Interaction Community Annotation Tool (PHI-Canto) automatically retrieves details of the proteins from UniProtKB, including the gene name, gene product, and taxonomy (e.g. the species name).

## Specifying strains

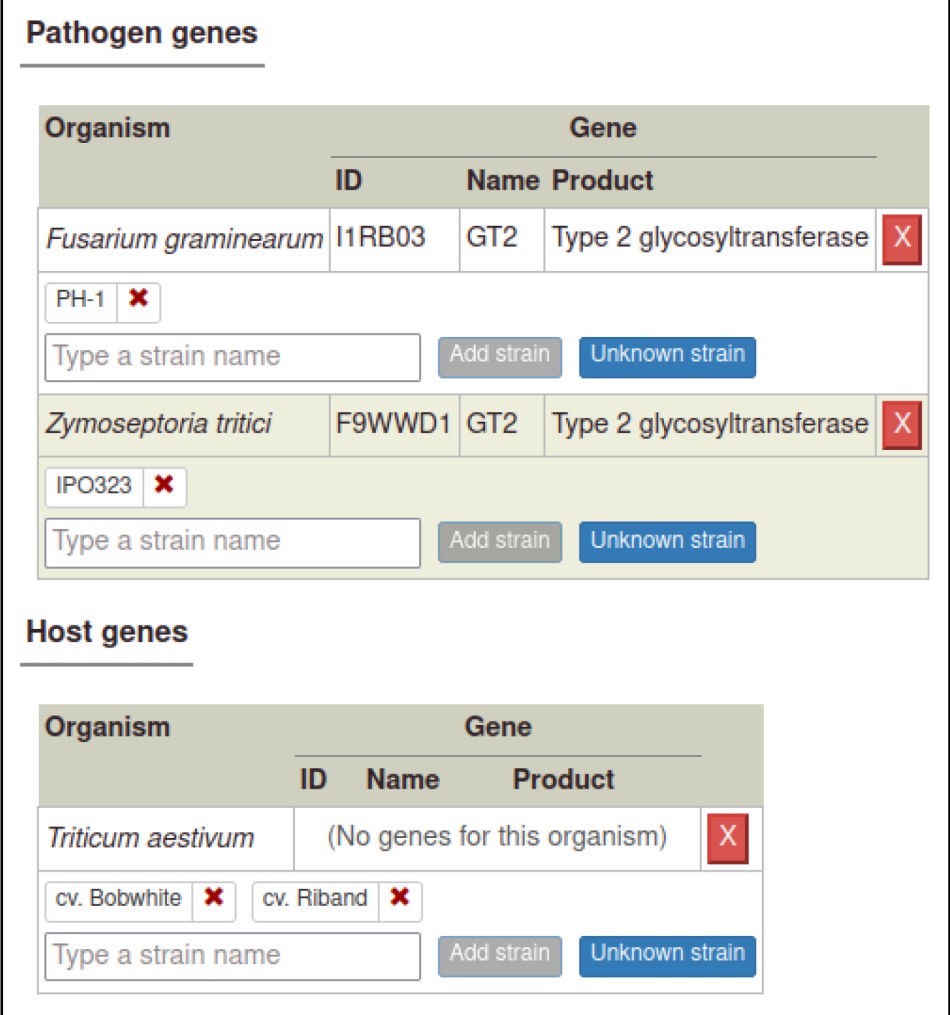

**Appendix 2—figure 7.** The curator must enter the strains for each organism studied in the publication or must specify when the strain was not known (or not specified in the publication). The Pathogen–Host Interaction Community Annotation Tool (PHI-Canto) provides a pre-populated list of strains for many species that the curator can select from, though they also have the option to specify a strain not in the list as free text. In this publication, the pathogen strains are PH-1 for *F. graminearum* and IPO323 for *Z. tritici*. Two cultivars of *T. aestivum* were used: cv. Bobwhite and cv. Riband.

## Creating alleles and genotypes

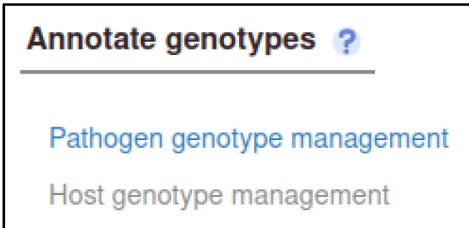

**Appendix 2—figure 8.** In order to show that deleting *GT2* in the pathogen causes a loss of pathogenicity, the curator must annotate the interaction between the mutant pathogen and its host with a phenotype, meaning the interaction must be added to the curation session. In the Pathogen–Host Interaction Community Annotation Tool *Appendix 2—figure 8 continued on next page*

*Appendix 2—figure 8 continued*
(PHI-Canto), interactions are represented as *metagenotypes*, which are the combined genotypes of the pathogen and host species. Before the curator can create a metagenotype, they must first create a genotype. Genotypes are composed of alleles (except in the case of wild-type host genotypes with no specified genes, as described later), and metagenotypes are composed from genotypes. So, the curator must first create an allele from a gene, then a genotype from an allele, then a metagenotype from two genotypes. The curator starts from the Pathogen genotype management page, following a link from the Curation summary page.

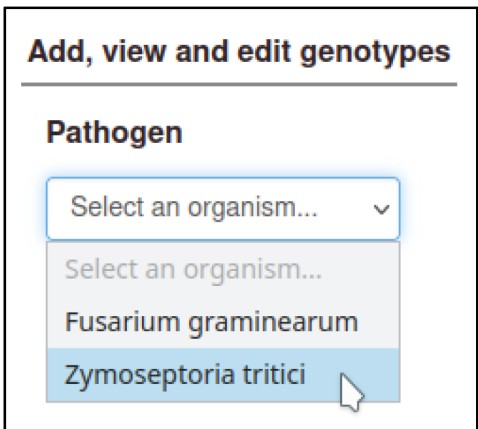

**Appendix 2—figure 9.** The curator then selects a pathogen species (*Z. tritici*) from a drop-down menu.

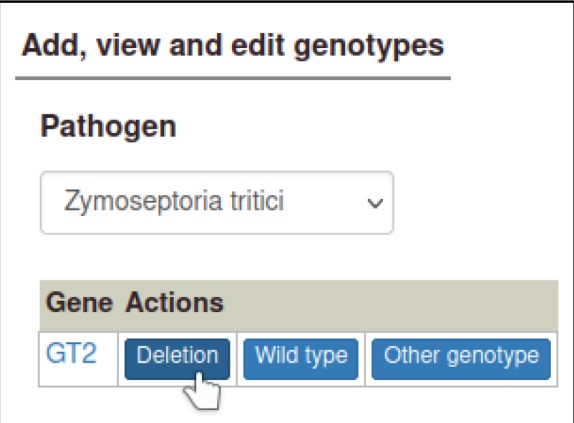

**Appendix 2—figure 10.** Selecting a pathogen species shows a list of genes for the species, with buttons to create types of alleles. Here, the curator selects 'Deletion' for a deletion allele.

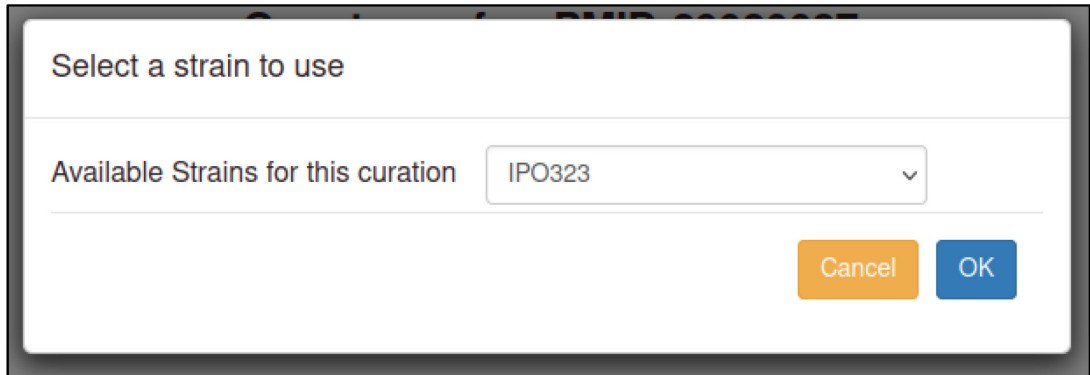

**Appendix 2—figure 11.** The curator is prompted for the strain the deletion occurred in.

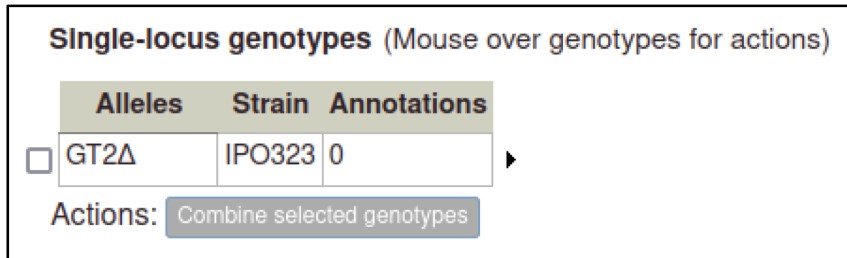

**Appendix 2—figure 12.** After selecting this, the Pathogen–Host Interaction Community Annotation Tool (PHI-Canto) creates a genotype containing a single allele, with the allele name automatically generated from the gene name followed by a delta symbol.

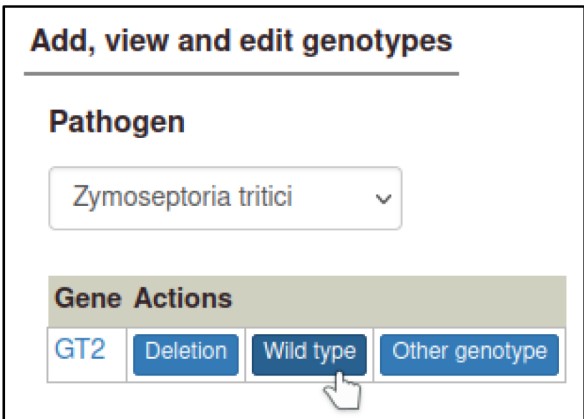

**Appendix 2—figure 13.** The curator will also need to prepare a wild-type genotype for the pathogen *GT2* gene, which can be added to the control metagenotype so that any changes in the phenotype (between the wild-type pathogen and the altered pathogen inoculated onto the host) can be properly annotated. This first requires making a wild-type allele for *GT2*, using the 'Wild-type' allele type.

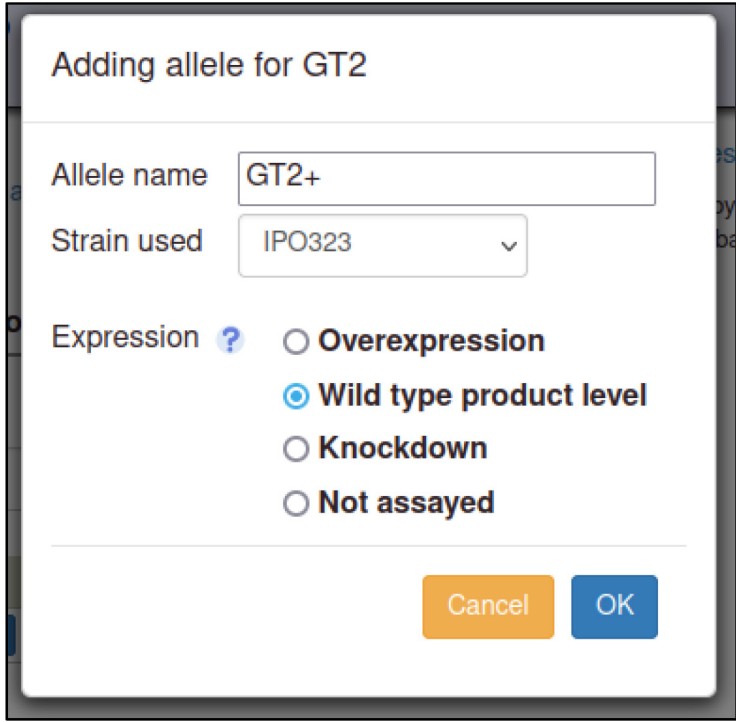

**Appendix 2—figure 14.** Wild-type alleles require the gene expression level to be specified. In this case, there was no change in expression level, so the curator selects 'Wild-type product level.' The Pathogen–Host Interaction Community Annotation Tool (PHI-Canto) automatically creates an allele name by appending a plus symbol to the gene name.

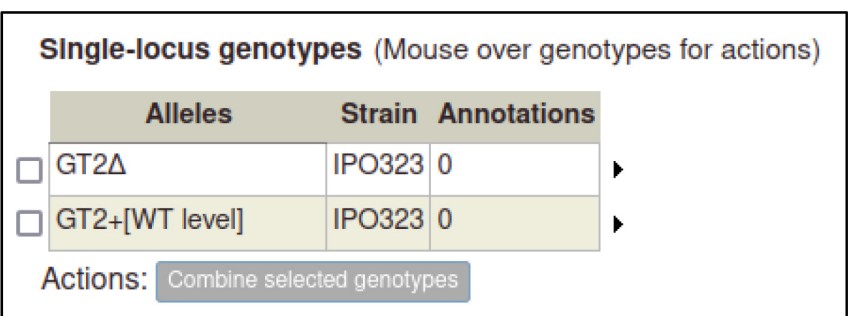

**Appendix 2—figure 15.** As genotypes are created, they are added to a table of genotypes on their respective genotype management page (Pathogen genotype management for pathogens, Host genotype management for hosts).

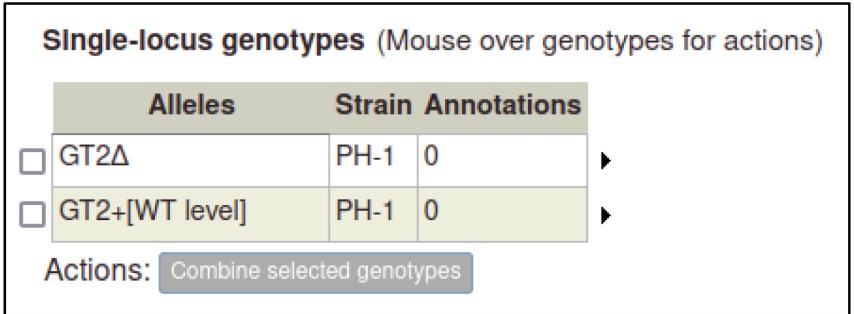

**Appendix 2—figure 16.** The curator can repeat the process above to create pathogen genotypes for *F. graminearum*.

### Creating metagenotypes for pathogen–host interactions

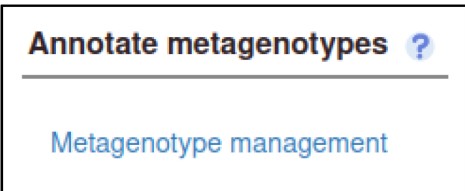

**Appendix 2—figure 17.** Metagenotypes are created using the Metagenotype management page, where genotypes previously added to the curation session can be combined into a metagenotype. The curator can reach this page from the Curation Summary page, or from either the pathogen, or host genotype management page.

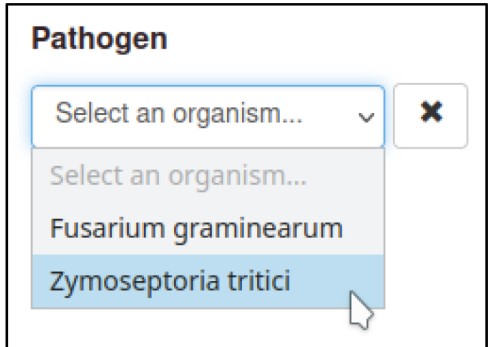

**Appendix 2—figure 18.** The curator starts by selecting a pathogen species from a drop-down menu.

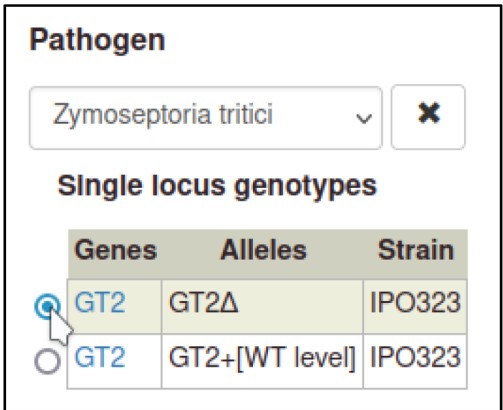

**Appendix 2—figure 19.** Then the curator selects a genotype from the table of pathogen genotypes.

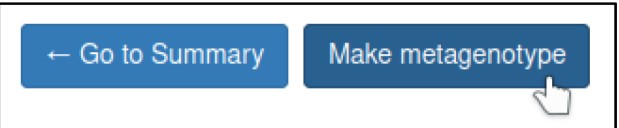

**Appendix 2—figure 20.** Then the curator selects a host genotype. For wild-type hosts, the Pathogen–Host Interaction Community Annotation Tool (PHI-Canto) provides a shortcut where a strain can be selected without needing to create an allele as part of the genotype.

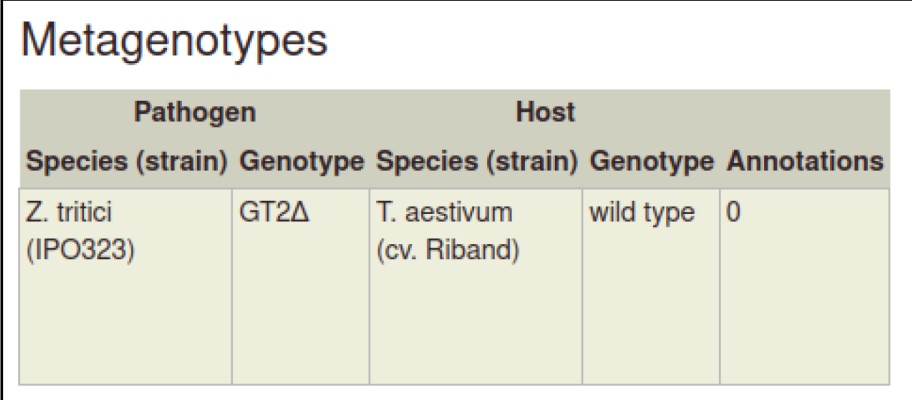

**Appendix 2—figure 21.** The curator selects 'Make metagenotype' to create the metagenotype for the interaction.

## Metagenotypes

| Pathogen | | Host | | |
|---|---|---|---|---|
| **Species (strain)** | **Genotype** | **Species (strain)** | **Genotype** | **Annotations** |
| Z. tritici (IPO323) | GT2Δ | T. aestivum (cv. Riband) | wild type | 0 |

**Appendix 2—figure 22.** The metagenotype is displayed in a table as a combination of pathogen and host genotype.

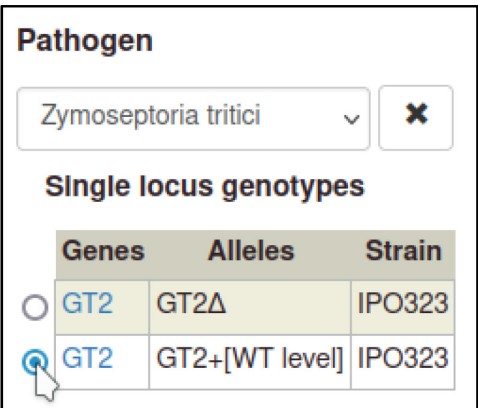

**Appendix 2—figure 23.** This process can be repeated to create the metagenotype for the wild-type interaction between *Z. tritici* and *T. aestivum*. In this case, the pathogen genotype containing the wild-type *GT2* is selected instead of the deletion allele.

## Metagenotypes

| Pathogen | | Host | | |
|---|---|---|---|---|
| Species (strain) | Genotype | Species (strain) | Genotype | Annotations |
| Z. tritici (IPO323) | GT2Δ | T. aestivum (cv. Riband) | wild type | 0 |
| Z. tritici (IPO323) | GT2+[WT level] | T. aestivum (cv. Riband) | wild type | 0 |

**Appendix 2—figure 24.** The additional metagenotype is now displayed in the table.

## Metagenotypes

| Pathogen | | Host | | |
|---|---|---|---|---|
| Species (strain) | Genotype | Species (strain) | Genotype | Annotations |
| F. graminearum (PH-1) | GT2Δ | T. aestivum (cv. Bobwhite) | wild type | 0 |
| F. graminearum (PH-1) | GT2+[WT level] | T. aestivum (cv. Bobwhite) | wild type | 0 |

**Appendix 2—figure 25.** Creating the corresponding metagenotypes for *F. graminearum* and *T. aestivum* simply requires changing the pathogen species and selecting cv. Bobwhite for the host strain.

## Annotating pathogen–host interactions with phenotypes

## Metagenotypes

| Pathogen | | Host | | | |
|---|---|---|---|---|---|
| Species (strain) | Genotype | Species (strain) | Genotype | Annotations | |
| Z. tritici (IPO323) | GT2Δ | T. aestivum (cv. Riband) | wild type | 0 | Annotate pathogen-host interaction phenotype<br>Annotate gene-for-gene phenotype<br>Annotate disease name<br>View phenotype annotations<br>Delete |

**Appendix 2—figure 26.** Metagenotypes can be annotated with phenotypes by selecting the 'Annotate pathogen-host interaction phenotype' action.

## Phenotype and evidence

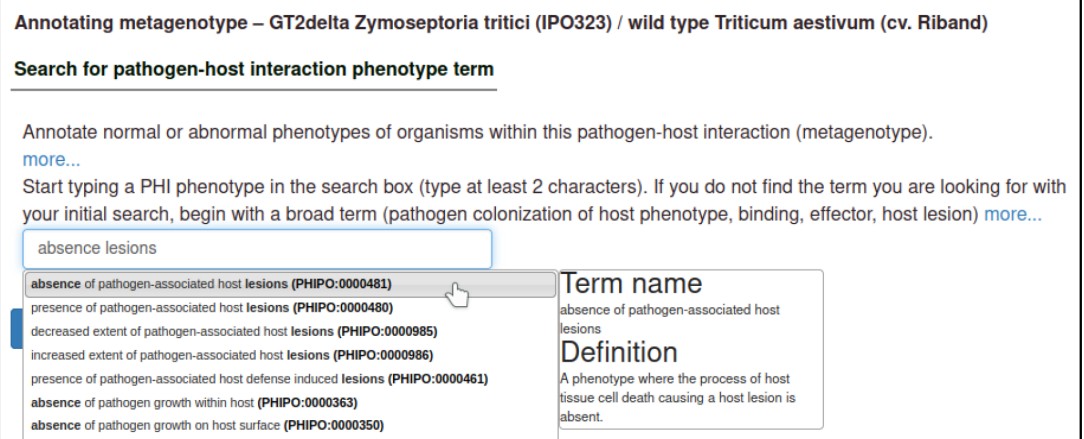

**Appendix 2—figure 27.** The first step is to select a term from a controlled vocabulary that describes the phenotype of the interaction. The Pathogen–Host Interaction Community Annotation Tool (PHI-Canto) uses terms from the Pathogen–Host Interaction Phenotype Ontology (PHIPO) for this purpose. The primary observed phenotype, in this case, is the *absence of pathogen-associated host lesions* (PHIPO:0000481).

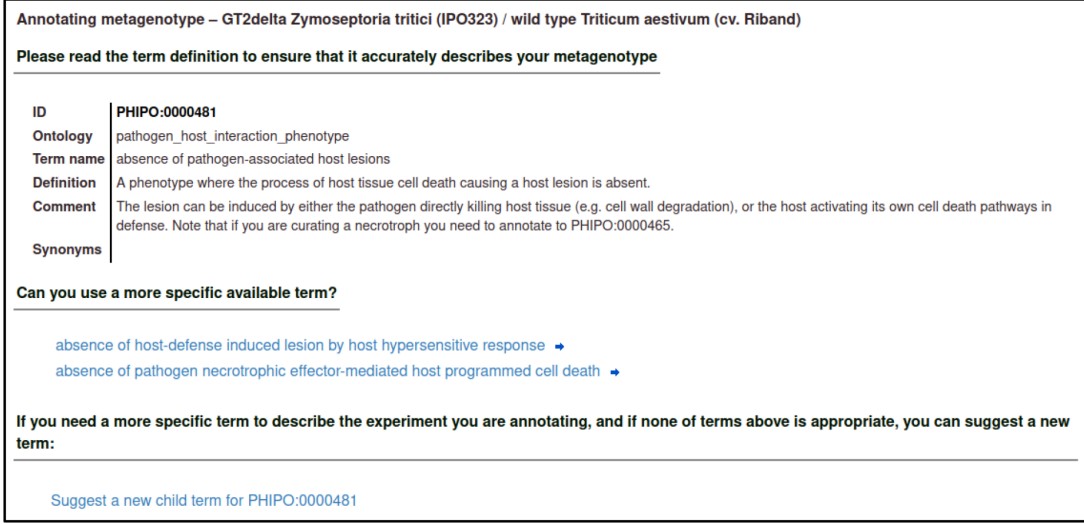

**Appendix 2—figure 28.** Upon selecting the term, the curator is shown a description of the term and its synonyms to help confirm that their chosen term is appropriate.

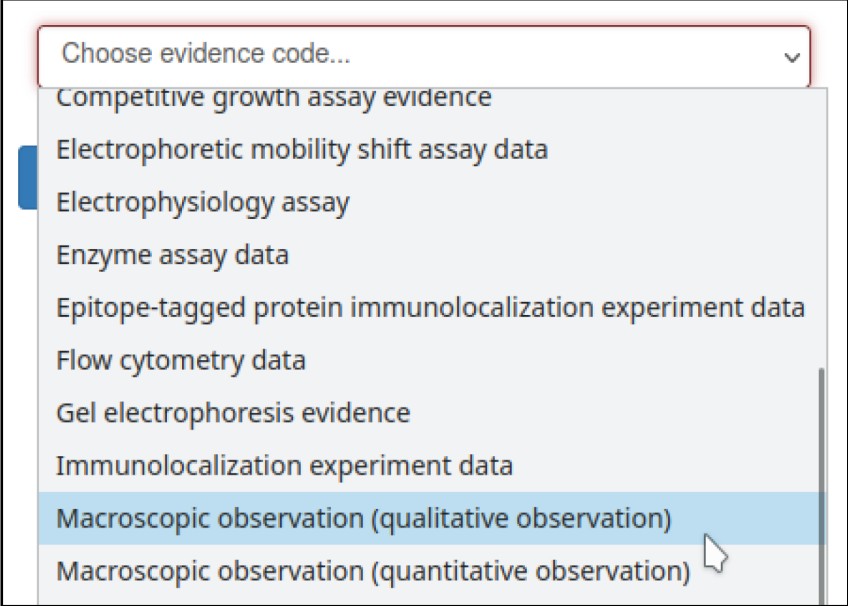

**Appendix 2—figure 29.** The curator must select an evidence code for the observation of the phenotype. In this case, the phenotype was observed macroscopically, and measured qualitatively.

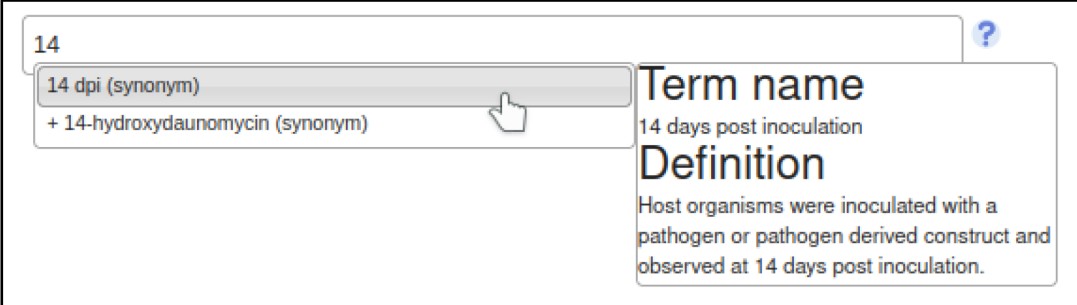

**Appendix 2—figure 30.** The curator may also specify experimental conditions for the experiment – such as the growth medium, or days elapsed after inoculation of the host. This annotation specifies that the assay was performed 14 days after inoculation with the *Z. tritici GT2* deletion mutant.

## Annotation extensions

**Annotation extensions**

These extension types are available for *absence of pathogen-associated host lesions* (PHIPO:0000481):

compared to control genotype
penetrance
severity
extent of infectivity
host tissue infected
outcome of interaction

**Appendix 2—figure 31.** The Pathogen–Host Interaction Community Annotation Tool (PHI-Canto) uses annotation extensions to provide additional information about the conditions and outcome of the pathogen–host interaction. Of particular note are the host tissue infected, the changes to the infective ability of the pathogen, the presence (or absence) of disease, and the interaction used as a control for the interaction involving a mutant pathogen.

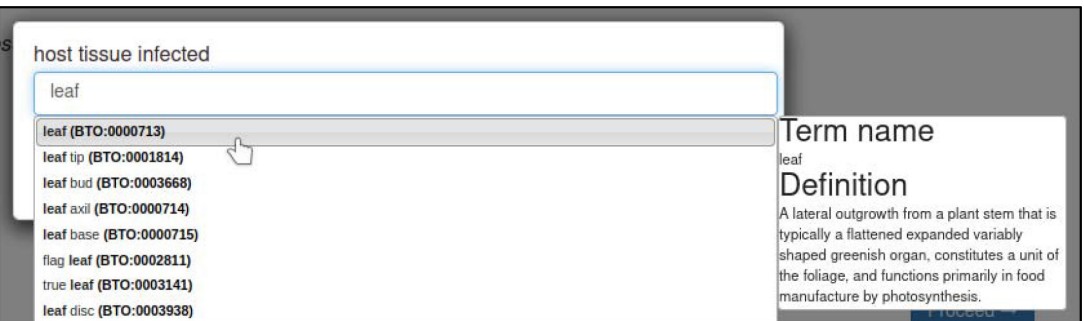

**Appendix 2—figure 32.** The host tissue that was infected during the interaction is annotated with the 'host tissue infected' annotation extension. This extension uses ontology terms from the BRENDA Tissue Ontology (BTO). In this case, the curator specifies that the *leaf* (BTO:0000713) of *T. aestivum* was infected.

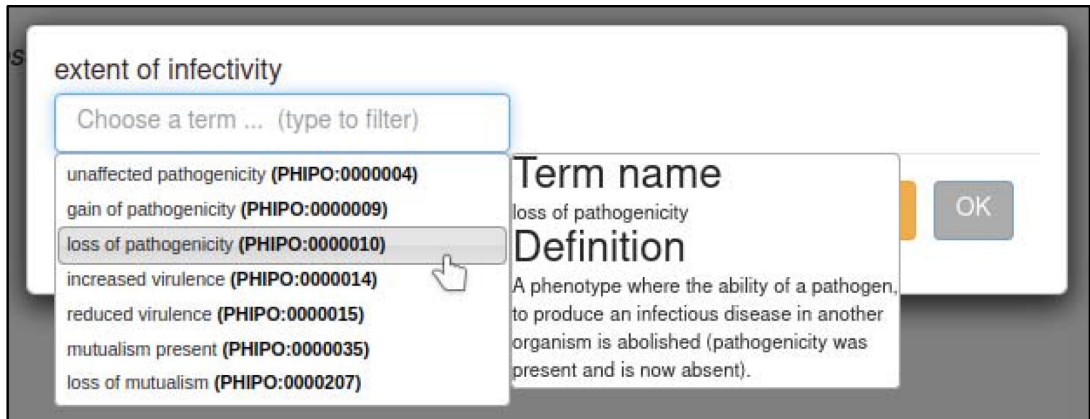

**Appendix 2—figure 33.** Changes in the infective ability of the pathogen are annotated with the 'extent of infectivity' annotation extension. This extension uses a subset of ontology terms from the Pathogen–Host Interaction Phenotype Ontology (PHIPO). In this case, the curator specifies that the interaction resulted in a *loss of pathogenicity* (PHIPO:0000010).

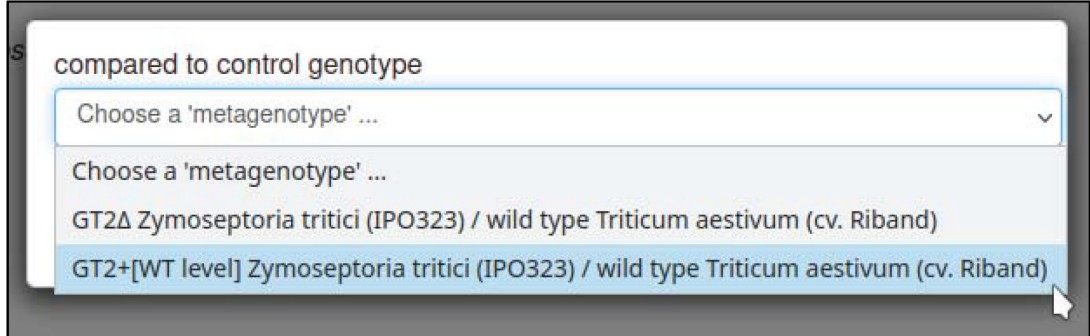

**Appendix 2—figure 34.** The control interaction (to which the interaction being annotated should be compared) can be annotated with the 'compared to control genotype' annotation extension. This annotation allows any metagenotype in the curation session to be designated as a control. In this case, the curator selects the wild-type metagenotype that was created earlier.

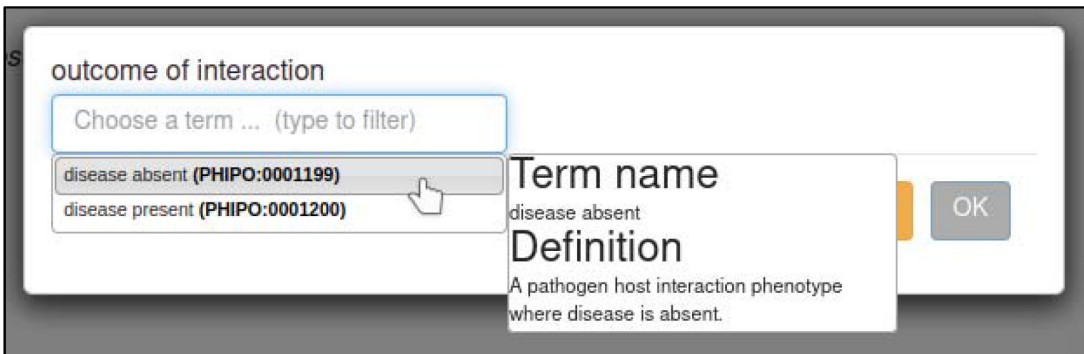

**Appendix 2—figure 35.** The presence or absence of disease resulting from the interaction can be annotated with the 'outcome of interaction' annotation extension. This extension uses a subset of ontology terms from the Pathogen–Host Interaction Phenotype Ontology (PHIPO). In this case, the curator specifies that no disease was observed as a result of the interaction: *disease absent* (PHIPO:0001199).

## Figure numbers and comments

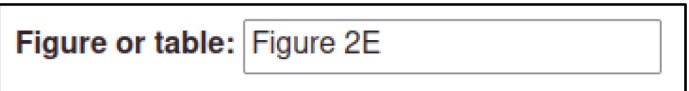

**Appendix 2—figure 36.** After adding annotation extensions, the curator has the option to provide the figure number from the publication (if any) that illustrates the phenotype. In this case, the figure was Figure 2E.

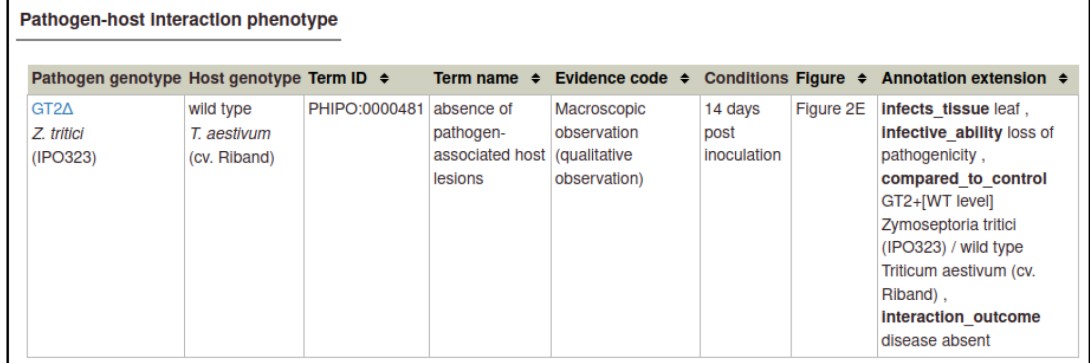

**Appendix 2—figure 37.** The curator can also provide additional information in a comments field, in case of details that are not appropriate for any other field. Once the above steps are completed, the phenotype annotation is created.

## Copying annotations

| ◆ | Conditions | Figure ◆ | Annotation extension ◆ | |
|---|---|---|---|---|
| | 14 days post inoculation | Figure 2E | **Infects_tissue** leaf , **infective_ability** loss of pathogenicity , **compared_to_control** GT2+[WT level] | View metagenotype<br>Edit<br>Copy and edit<br>Delete |

**Appendix 2—figure 38.** The above annotation can be used as a template for the interaction between the wild-type pathogen and host, since many of the variables are the same. The Pathogen–Host Interaction Community Annotation Tool (PHI-Canto) provides a 'Copy and edit' feature that allows curators to use one annotation as a template for creating another.

| Pathogen genotype | Host genotype | Term ID ◆ | Term name ◆ | Evidence code ◆ | Conditions | Figure ◆ | Annotation extension ◆ |
|---|---|---|---|---|---|---|---|
| GT2+[WT level] *Z. tritici* (IPO323) | wild type *T. aestivum* (cv. Riband) | PHIPO:0000480 | presence of pathogen-associated host lesions | Macroscopic observation (qualitative observation) | 14 days post inoculation | Figure 2E | **Infects_tissue** leaf , **interaction_outcome** disease present |

**Appendix 2—figure 39.** For the wild-type interaction, the pathogen genotype is changed to wild-type *GT2*, the phenotype term is changed to *presence of pathogen-associated host lesions* (PHIPO:0000480), the interaction outcome is changed to *disease present* (PHIPO:0001200), and the extensions for infective ability and control metagenotypes are removed, since they are not applicable.

| Pathogen genotype | Host genotype | Term ID ◆ | Term name ◆ | Evidence code ◆ | Conditions | Figure ◆ | Annotation extension ◆ |
|---|---|---|---|---|---|---|---|
| GT2Δ *F. graminearum* (PH-1) | wild type *T. aestivum* (cv. Bobwhite) | PHIPO:0000481 | absence of pathogen-associated host lesions | Macroscopic observation (qualitative observation) | 13 days post inoculation | Figure 4E | **Infects_tissue** inflorescence , **infective_ability** loss of pathogenicity , **compared_to_control** GT2+[WT level] Fusarium graminearum (PH-1) / wild type Triticum aestivum (cv. Bobwhite) , **interaction_outcome** disease absent |

**Appendix 2—figure 40.** The interaction between *Z. tritici* and *T. aestivum* can also be used as a template for the interaction between *F. graminearum* and *T. aestivum*. Here, the pathogen genotype is changed to the *GT2* deletion *F. graminearum*, the host strain is changed to cv. Bobwhite, the experimental condition is changed to '13 days post inoculation,' the host tissue infected is changed to *inflorescence* (BTO:0000628), the control metagenotype is updated accordingly, and the figure number is changed to 4E.

| Pathogen genotype | Host genotype | Term ID ⬍ | Term name ⬍ | Evidence code ⬍ | Conditions | Figure ⬍ | Annotation extension ⬍ |
|---|---|---|---|---|---|---|---|
| GT2+[WT level] *F. graminearum* (PH-1) | wild type *T. aestivum* (cv. Bobwhite) | PHIPO:0000480 | presence of pathogen-associated host lesions | Macroscopic observation (qualitative observation) | 13 days post inoculation | Figure 4E | **Infects_tissue** leaf , **interaction_outcome** disease present |

**Appendix 2—figure 41.** The changes required for the wild-type interaction between *F. graminearum* and *T. aestivum* are the same as those required for *Z. tritici* and *T. aestivum*, since the interaction outcome is the same (presence of pathogen-associated host lesions, and presence of disease).

**Pathogen-host interaction phenotype**

| Pathogen genotype | Host genotype | Term ID ⬍ | Term name ⬍ | Evidence code ⬍ | Conditions | Figure ▲ | Annotation extension ⬍ |
|---|---|---|---|---|---|---|---|
| GT2Δ *Z. tritici* (IPO323) | wild type *T. aestivum* (cv. Riband) | PHIPO:0000481 | absence of pathogen-associated host lesions | Macroscopic observation (qualitative observation) | 14 days post inoculation | Figure 2E | **Infects_tissue** leaf , **infective_ability** loss of pathogenicity , **compared_to_control** GT2+[WT level] Zymoseptoria tritici (IPO323) / wild type Triticum aestivum (cv. Riband) , **interaction_outcome** disease absent |
| GT2+[WT level] *Z. tritici* (IPO323) | wild type *T. aestivum* (cv. Riband) | PHIPO:0000480 | presence of pathogen-associated host lesions | Macroscopic observation (qualitative observation) | 14 days post inoculation | Figure 2E | **Infects_tissue** leaf , **interaction_outcome** disease present |
| GT2Δ *F. graminearum* (PH-1) | wild type *T. aestivum* (cv. Bobwhite) | PHIPO:0000481 | absence of pathogen-associated host lesions | Macroscopic observation (qualitative observation) | 13 days post inoculation | Figure 4E | **Infects_tissue** inflorescence , **infective_ability** loss of pathogenicity , **compared_to_control** GT2+[WT level] Fusarium graminearum (PH-1) / wild type Triticum aestivum (cv. Bobwhite) , **interaction_outcome** disease absent |
| GT2+[WT level] *F. graminearum* (PH-1) | wild type *T. aestivum* (cv. Bobwhite) | PHIPO:0000480 | presence of pathogen-associated host lesions | Macroscopic observation (qualitative observation) | 13 days post inoculation | Figure 4E | **Infects_tissue** leaf , **interaction_outcome** disease present |

**Appendix 2—figure 42.** Shown here is a table of all the pathogen–host interaction phenotypes from this curation example.

## Disease annotation

## Metagenotypes

| Pathogen | | Host | | | |
|---|---|---|---|---|---|
| Species (strain) | Genotype | Species (strain) | Genotype | Annotations | |
| Z. tritici (IPO323) | GT2+[WT level] | T. aestivum (cv. Riband) | wild type | 1 | Annotate pathogen-host interaction phenotype Annotate gene-for-gene phenotype Annotate disease name View phenotype annotations Delete |

**Appendix 2—figure 43.** The Pathogen–Host Interaction Community Annotation Tool (PHI-Canto) provides the 'Disease name' annotation type, which is used to annotate a disease to a pathogen–host interaction. These annotations highlight the fact that two different pathogens infecting different tissue types of the same host have

*Appendix 2—figure 43 continued*
been used in experiments within this publication. Disease name annotations are made on the Metagenotype Management page, via the 'Annotate disease name' link.

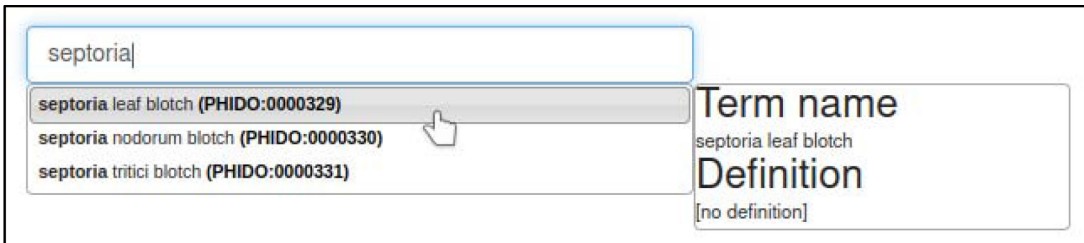

**Appendix 2—figure 44.** The curator can select a disease from a list of disease names provided by the Pathogen–Host Interactions database (PHI-base) Disease List (PHIDO). For *Z. tritici*, the disease is *septoria leaf blotch* (PHIDO:0000329).

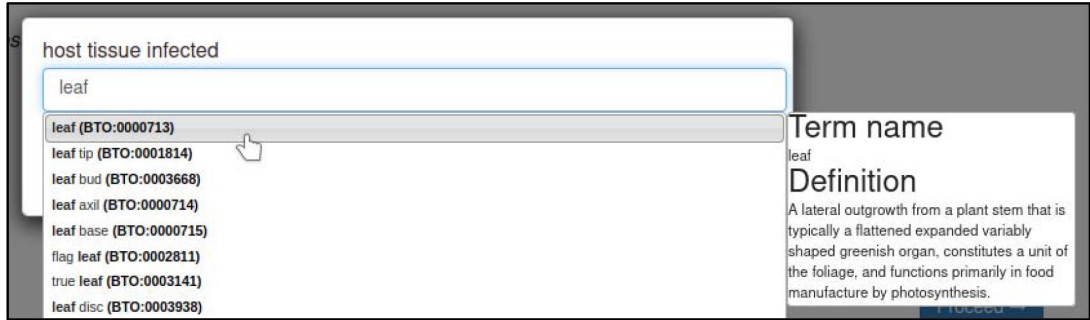

**Appendix 2—figure 45.** Disease name annotations also allow the host tissue infected to be specified. In this case, the tissue is the *leaf* (BTO:0000713).

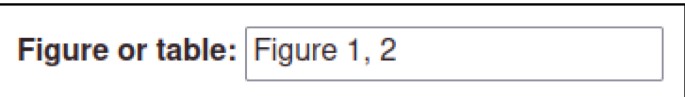

**Appendix 2—figure 46.** The curator has the option to provide the figure number and additional comments. In this case, the figure numbers are 1 and 2.

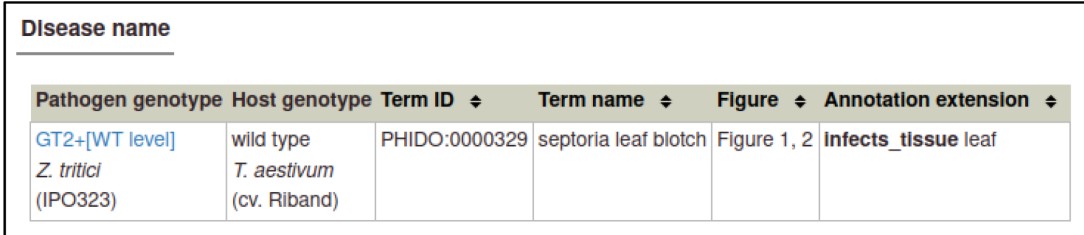

**Appendix 2—figure 47.** Once this step is completed, the disease name annotation is created.

| Disease name | | | | | |
| --- | --- | --- | --- | --- | --- |
| **Pathogen genotype** | **Host genotype** | **Term ID** ⬍ | **Term name** ⬍ | **Figure** ⬍ | **Annotation extension** ⬍ |
| GT2+[WT level] *F. graminearum* (PH-1) | wild type *T. aestivum* (cv. Bobwhite) | PHIDO:0000162 | fusarium ear blight | Figure 4 | **infects_tissue** inflorescence |

**Appendix 2—figure 48.** The same process can be followed to create the Disease name annotation for *F. graminearum*: the genotype is the wild-type *GT2*, the host cultivar is *cv. Bobwhite*, the disease is *fusarium ear blight* (PHIDO:0000162), the host tissue infected is the *inflorescence* (BTO:0000628), and the figure number is 4.

## Gene Ontology annotation

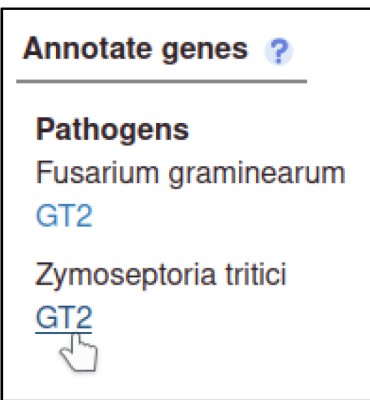

**Appendix 2—figure 49.** The Pathogen–Host Interaction Community Annotation Tool (PHI-Canto) also provides the ability to annotate biological processes, molecular functions, and cellular components associated with wild-type versions of genes, using terms from the Gene Ontology (GO). In this publication, GT2 is described as having glycosyltransferase activity as its molecular function, so the curator can annotate this. Gene Ontology annotations are made by selecting the gene from the Curation Summary page.

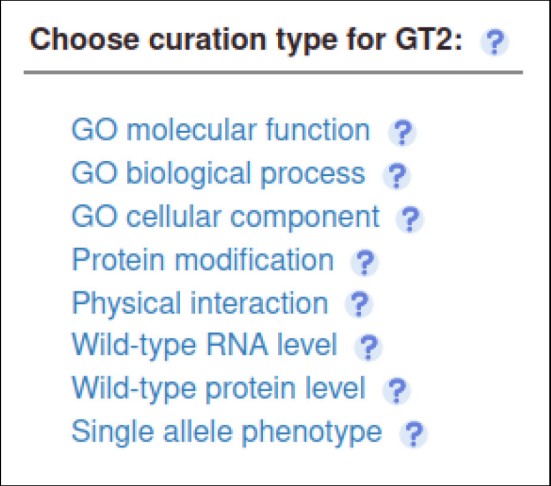

**Appendix 2—figure 50.** The gene details page has a list of available annotation types.

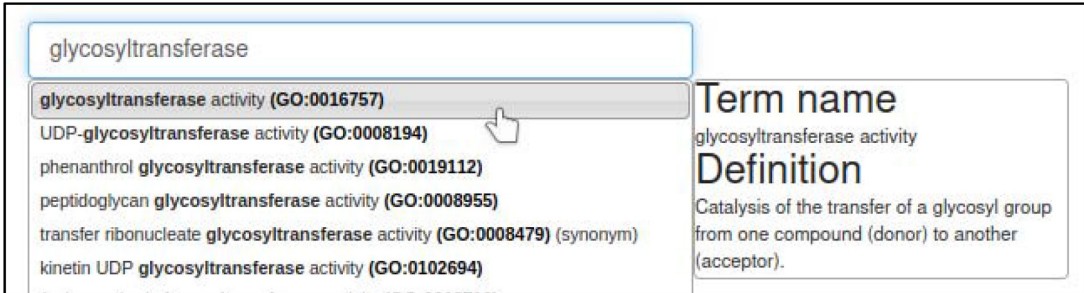

**Appendix 2—figure 51.** The curator selects the Gene Ontology (GO) Molecular Function annotation type and is prompted for a term from the Gene Ontology. In this case, the correct term is *glycosyltransferase activity* (GO:0016757).

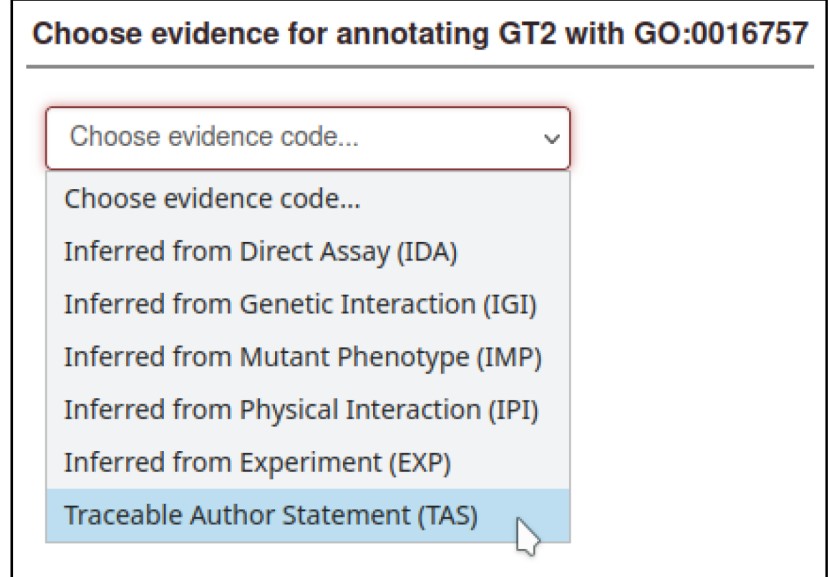

**Appendix 2—figure 52.** The curator must provide an evidence code from a controlled list specified by the Gene Ontology. The appropriate evidence code in this case is a *Traceable Author Statement* in the publication.

**Annotation extensions**

These extension types are available for *glycosyltransferase activity* (GO:0016757):

with host species
has function during
physical location
involved in biological process
PR:nnn ID for gene product form
qualifier

**Appendix 2—figure 53.** here are many annotation extensions available for Gene Ontology (GO) annotations, but in this case, none of them are applicable (or required), so the curator skips this step.

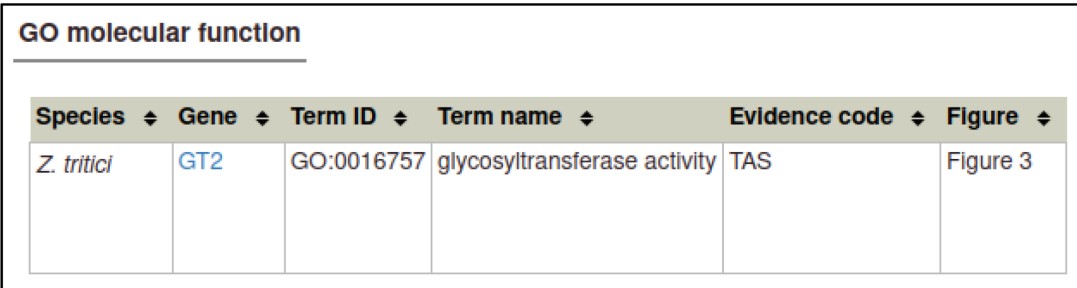

**Appendix 2—figure 54.** Figure numbers can be specified for Gene Ontology (GO) annotations: in this case, the relevant figure is Figure 3.

| GO molecular function | | | | | |
| --- | --- | --- | --- | --- | --- |
| **Species** ⬍ | **Gene** ⬍ | **Term ID** ⬍ | **Term name** ⬍ | **Evidence code** ⬍ | **Figure** ⬍ |
| *Z. tritici* | GT2 | GO:0016757 | glycosyltransferase activity | TAS | Figure 3 |

**Appendix 2—figure 55.** Once this step is completed, the molecular function annotation is created.

## Other annotation types

The publication contains other information which is not included in this worked example for the sake of brevity. In the real curation session, this other information is captured as the following annotations:

- GO biological process annotations indicate that GT2 is involved in the hyphal growth process.
- GO cellular component annotations indicate that GT2 is located in the hyphal cell wall.
- Pathogen phenotype annotations capture information about the pathogen *in vitro*, specifically normal and altered phenotypes for unicellular population growth, hyphal growth, cellular melanin accumulation, filament morphology, and so on.

All these annotation types use the same annotation process as the annotation types described above.

## Submitting the curation session

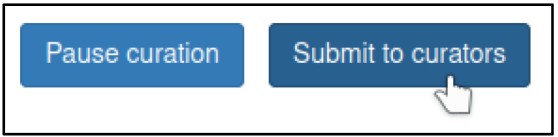

**Appendix 2—figure 56.** Once the curator has made all their annotations, the curation session is submitted to the PHI-base team for review. The curator can use a text box to provide any information that is outside the scope of the curation process before finishing the submission process. Once the submission process is finished, the curation session can no longer be edited except by members of the Pathogen–Host Interactions database (PHI-base) team, who have the option to reactivate the session in case changes are required by the original curator.

# Appendix 3

## Author checklist prior to publication

Here, we have developed a list of important points for an author to consider prior to submitting a manuscript for publication. Nine key points are displayed in *Appendix 3—table 1*.

**Appendix 3—table 1.** Author checklist prior to publication.

| Point number | Point for the author to consider |
|---|---|
| 1 | Use the UniProtKB assigned gene name. Synonyms can be recorded in addition to the gene name. Prefix the gene name with the genus and species initials if the same genes from multiple species are used. |
| 2 | If reporting on a new (gene) sequence, submit your sequence to NCBI GenBank or the European Nucleotide Archive (ENA), then obtain an accession number prior to publication. Record this accession number within the manuscript. If reporting on a gene with an existing accession number, make sure this is reported in the manuscript. Please record the UniProtKB accession number for the protein of the gene, where available. Provide or use any existing informative allele or line designations for mutations and transgenes. |
| 3 | Provide a binomial species name for pathogen and host organisms, not just a common name. If possible, please also include NCBI Taxonomy IDs for the pathogen and host organisms at the rank of species. |
| 4 | Describe the tissue or organ in which the experimental observations were made (controlled language can be found in the BRENDA Tissue Ontology, see https://www.ebi.ac.uk/ols/ontologies/bto). |
| 5 | Describe any experimental techniques used, and accurately record any chemicals or reagents used. |
| 6 | When writing an article, try to keep the use of descriptive language as accurate and controlled as possible. For example, do not use 'reduced pathogenicity' or 'loss of virulence,' as these terms can be misleading: it would be more accurate to use 'reduced virulence' and 'loss of pathogenicity,' respectively. Ideally, try to follow the terminology of an existing ontology: this will make the data easier to extract and reuse. Relevant ontologies include PHIPO and GO (https://www.ebi.ac.uk/ols/ontologies/phipo, https://www.ebi.ac.uk/ols/ontologies/go). |
| 7 | Document all the key information for the paper: do not rely on citing past papers for information on the pathogen used, or the strain used, and so on. |
| 8 | Think carefully when choosing keywords for your manuscript to ensure that the publication can be located by PHI-base's keyword searches. One example of an ideal keyword is 'pathogen-host interaction.' |
| 9 | Record the provenance of the pathogen strain: for example, whether it is a lab strain or a field isolate, or if the strain was obtained from a stock center or as a gift from another lab. |

