## [Editor Report]

Focused on host-pathogen interactions, this valuable study presents a useful resource for unifying language(s) and rules used in biology experiments, with a new ontology and tool called PHI-Canto. The framework enables using UniProtKB IDs to curate proteins and eventually derive 'metagenotypes', an important concept that may incidentally help shrinking proliferating names and acronyms for genes, processes, and interactions. This important framework builds on established standards and methods and was rigorously tested with a variety of publications, providing a system that may eventually capture complex information hidden in the data, such as metagenotypes.

---

## [Decision Letter]

**Decision letter after peer review:**

Thank you for submitting your article "A framework for community curation of interspecies interactions literature" for consideration by *eLife*. Your article has been reviewed by 3 peer reviewers, one of whom is a member of our Board of Reviewing Editors, and the evaluation has been overseen by Meredith Schuman as the Senior Editor. The following individual involved in the review of your submission has agreed to reveal their identity: Lorena Etcheverry (Reviewer #2).

Essential revisions:

1. Specify where the contributions go, how curation is done, and how they are made available.

2. Describe or reference the complete data model behind annotations, namely: concepts, methods, eventual algorithms, as well as formats for information storage and retrieval.

3. Expand on how data display and interoperability are implemented, for example, to link related information from new and existing publications. Address the possible use of graph representation to link complex information. (See more detailed comments from Reviewers 2 and 3.)

4. Explain why PHIDO has been generated rather than using existing disease terminologies (such as Mondo or DO).

5. The authors should briefly comment on the possible extension of this approach beyond pathogen-host interactions, which could increase the broader relevance of the study.

*Reviewer #1 (Recommendations for the authors):*

Many readers may wrongly think they will find tones of centralized information about, e.g. their present-day favorite gene. This does not seem to be the case, leading to a related question: where do the contributed curations go and how are they made available? Is there a final control from within the resource team to filter wrong curations due to bad procedures, or even directly fraudulent data treatments?

On the model extension side, feeding on new interactions proposed by users: it is not clear what kind of follow-up would be made in order to encompass the usage with appropriate growth and amelioration. Are there plans for this? Also, given that most authors are in the industry, a short statement about conflicts of interest would be desirable. Related to this, it is not clear to the reader if new, useful added curations obtained by a user, will be added to the resource and made publicly available. In an ideal world, the ten examples forming the basis of the resource should grow to thousands. I might be missing something, though.

*Reviewer #2 (Recommendations for the authors):*

Given my area of expertise, I am not in a position to assess the relevance of the work from a biological point of view. However, I feel that some points relating to data management and problem modeling deserve some comment.

In particular, there is no mention of the complete data model of each annotation or the formats in which this information is stored. This omission is probably because this is part of the Canto project. Still, to make this publication self-contained, it would be desirable to include this information or at least a reference to it, especially to measure the changes required.

Something that would also improve the work is more detail on how to use or visualize the data generated. Although there are a few brief lines in the section on "Display and interoperability of data", Figure 4 raises the question of how the system will behave in cases where there are already curated publications that refer to the pathogens and hosts of the new publication to be curated. It would be desirable that they are not treated in isolation but that the system allows them to be linked and then navigated in the network resulting from curating a set of publications.

Finally, in the introduction to the paper, the authors make the reckless assertion that manual biocuration is the only way to reliably represent information about functions and phenotypes. I would question this assertion given the current state of the art in LNP tools. While it is likely that, in many cases, automated annotation or curation using these techniques will not yield such accurate results, I believe it would be desirable to explore these techniques. It is also possible to think of hybrid human-in-the-loop systems where automatic techniques assist experts and simplify repetitive tasks. I believe that the paper should at least include a discussion of these issues.

*Reviewer #3 (Recommendations for the authors):*

Here are some specific points of confusion, questions, or suggestions for improvement:

Page 5 of the manuscript (page 6 of the full pdf):

– Lines 103-105 talk about changes in pathogenicity and virulence. It would be useful to readers to have a brief explanation of how these differ from each other and why one only applies to the pathogen while the other can apply to either the host or the pathogen.

– bottom of the page talks about "annotation types". The term "annotation type" seems to be used in a way that allows confusion with the entity being annotated. I believe that the authors intend to say that gene, genotype, and metagenotype are types of biological features (to use their term) that are annotated within PHI-Canto, each with its own set of accompanying annotation types as outlined in Table 1. If that is the correct interpretation, then I suggest modifying the text to make this more explicit with a particular focus on the sentence on lines 111-113.

Page 6 of the manuscript

Line 114 – "curators use annotation extensions" is referenced to a GOA paper. Perhaps then the sentence should specify that GO annotation curators use annotation extensions or indicate the reference is an example of this practice, using "e.g." perhaps.

Page 11 of the manuscript

Last paragraph – the text mentions that ECO terms are used to capture evidence. However, the annotation examples in Appendix 1 appear to use a combination of GO evidence codes and terms/phrases that are related to ECO term names. If ECO is being used, why not use the ECO term ids and/or term names across the board?

Page 12 of the manuscript

Top of page – what is the relationship between PHIDO and other existing disease ontologies such as Mondo or DO?

Figure 5

The NCBI taxonomy is listed in the databases section, however, it is more of a cv – it's certainly not a database like UniProtKB or PHI-base are. The Evidence and Conclusion Ontology is mentioned in the text but is not in the list of OBO Ontologies. The curated list of strains (line 247, page 11) is not shown as a PHI-base CV, although perhaps the list of strains is stored in the form of the mapping file that is shown in the figure. Is that the case? If so, then perhaps rename that box to make that more clear.

Table 1

– In the gene section, GO annotation type – are the host species and symbiont species extensions meant to indicate the interacting species? Or the species from which the gene comes? I'm assuming it means the interacting species, but this could be made more explicit.

– In the genotype section under "single species phenotype" should it not say "(Pathogen phenotype or Host phenotype)" rather than "and"?

Appendix 1

– In general, I find the header/spacing organization made it difficult to follow where one part ended and the next began. Perhaps giving letters to the sections starting "If you have…" such that there would be Section 1A, 1B, etc. might help. Also perhaps the use of some indentation so that separate sections referring to each publication will be easier to see.

– Section 1, the section on "If you have a metagenotype phenotype recording "a pathogen effector' (corresponds to footnote 5 in table 2)" – annotations for PMID:31804478: I'm confused why in the gene level GO process annotations that the gene is annotated to the specific child GO:0052034 'effector-mediated suppression of host pattern-triggered immunity', however, when the 'protein binding' and 'enzyme inhibitor activity' GO function annotations are made, the part_of annotation extension is to a grandparent of GO:0052034 that is GO:0140590 'effector-mediated suppression of host defenses. I would have thought that the part_of annotations would have been to GO:0052034. Why is this not the case?

Table 1 and Appendix 1 – both refer to the example paper involving mutualism. By definition, in a mutualist relationship, neither partner is a pathogen as disease is not caused. I realize this issue is likely beyond the scope of this paper to discuss, but I wonder about the inclusion of this annotation in the PHI resource since it does not involve a pathogen. Is it because the species has been seen to be a pathogen in other cases? Or is it that PHI includes some non-pathogenic interactions as well? If the first case, it brings to bear how to define something as a pathogen when (as is true for almost all organisms that cause disease in another organism) it only causes disease in some situations and not others (which is of course relevant to the host-pathogen-environment disease triangle mentioned in the text and Figure 1). If it is the second case, might it make sense to think about the scope of the resource as related to its name? Since mechanisms of colonization are often shared between pathogens and beneficial commensals alike, including annotations for symbionts beyond known pathogens would be useful. However, if these are regularly included, more prominent statements to that effect made on the PHI website and in publications would inform users.

---

## [Author Response]

Essential revisions:1. Specify where the contributions go, how curation is done, and how they are made available.

Points – Where the contributions go and how they are made available.

Once data curated in PHI-Canto has been checked by a species expert, the data is added to the Pathogen-Host Interactions Database (PHI-base), which is freely available at www.phi-base.org. Information on PHI-base version 5 is given in this manuscript, and we also cite our recent publication by Urban et al. (2022; doi.org/10.1093/nar/gkab1037), which includes a detailed explanation and screenshots of the new gene-centric web display.

Point – How curation is done.

The approach taken to article curation is fully described in Appendix 2, where the reader is guided through a step-by-step worked example for PHI-Canto for the fully curated article by King et al. (2017), ‘A conserved fungal glycosyltransferase facilitates pathogenesis of plants by enabling hyphal growth on solid surfaces (PMID:29020037)’. In addition, we have published ten tutorial videos on YouTube that cover different aspects of the PHI-Canto curation process, plus three introductory videos on what PHI-base is, the criteria for selecting curatable publications into PHI-base, and the value of curating your publications into PHI-base. These videos are available at https://www.youtube.com/@PHI-base.

In addition to the above, the following detailed author response highlights the areas in the manuscript already covering the requested information and the newly added link to the YouTube channel videos on PHI-Canto and PHI-base.

In the section “Display and interoperability of data” the following original text explains where the contributions go:

“All data curated in PHI-Canto will be displayed in PHI-base version 5, introduced in (Urban et al., 2022).”

This text has now been modified for further clarification.

“All data curated in PHI-Canto will be displayed in the new gene-centric version 5 of PHI-base, introduced in Urban et al., 2022.”

This point is further reinforced in the Figure 4 legend with original text:

“After all annotations have been made, the session is submitted to PHI-base.”

Which has now been amended to:

“After all annotations have been made, the session is submitted into PHI-base version 5.”

In the worked curation example in Appendix 2 there are already instructions on how to view the example annotations on our new PHI-base gene-centric display pages:

“The information curated from this publication is available on the new gene centric PHI-base 5 website (http://phi5.phi-base.org, search for PHIG:308 and PHIG:307).”

In the Introduction we introduce PHI-base, its URL and state how the curated data is freely available for use (note: we have since added the additional reference for Urban et al., 2020):

“The pathogen–host interaction research communities are an example of a domain of the biological sciences exhibiting a literature deluge (Figure 1). The Pathogen–Host Interactions Database, PHI-base (phi-base.org), is an open-access FAIR biological database containing data on bacterial, fungal and protist genes proven to affect (or not to affect) the outcome of pathogen–host interactions (Rodriguez-Iglesias et al., 2016; Urban et al., 2020; Urban et al., 2022).”

Regarding how curation is done, we have a section titled “Summary of the PHI-Canto curation process” which thoroughly explains how to do curation as stated in the opening sentence of the original text:

“The PHI-Canto curation process is outlined in Figure 4, Figure 4 —figure supplement 1, the PHI-Canto user documentation and a detailed worked example is provided in Appendix 2.”

This text has further been updated in the revised manuscript to include information on the location of PHI-Canto video training tutorials:

“The PHI-Canto curation process is outlined in Figure 4, Figure 4 —figure supplement 1, the PHI-Canto user documentation, a detailed worked example provided in Appendix 2 and curation tutorials on the PHI-base YouTube channel (https://www.youtube.com/@PHI-base), under the playlist ‘PHI-Canto tutorial videos’.*”*

The text at the end of this section states:

“Once the curation process is complete, the curator submits the session for review by a nominated species expert.”

This step will enable quality control of curated data before it is publicly available in PHI-base (which we also noted in Urban et al. 2022).

A link to our comprehensive PHI-Canto user documentation is also provided in the “Code availability” section:

“The source code for PHI-Canto’s user documentation is available on GitHub, at https://github.com/PHI-base/canto-docs. The user documentation is licensed under the MIT license. The published format of the user documentation is available online at https://canto.phi-base.org/docs/index.”

In summary, data curated using PHI-Canto will be displayed in our new PHI-base version 5 gene-centric web display. This data is freely available for searches, for use and/or for downloading into other applications. Prior to display in PHI-base the curated data is checked by species experts. Extensive PHI-Canto training materials are available.

2. Describe or reference the complete data model behind annotations, namely: concepts, methods, eventual algorithms, as well as formats for information storage and retrieval.

We requested feedback from the editor regarding the above revision, as we felt some of the information requested (e.g. ‘eventual algorithms’) was not requested in the written communication by the reviewers and was not relevant. The editor clarified that we should “clearly explain the data models and what is behind them including any algorithms”, but that we should inform them if “any of the examples we listed under data model clarification are not relevant”. We have further described our data model and its related topics below.

Point – entity-relationship models.

We have included a basic entity–relationship model in Figure 3 – supplement 1 illustrating the new entities that were created for PHI-Canto. We have now also included two further Figure 3 supplements that expand on this information, referenced in the ‘Changes to the Canto data model and configuration’ section of the manuscript.

“To implement PHI-Canto several new entities were added to the Canto data model in order to support pathogen–host curation, as well as new configuration options (the new entities are illustrated in Figure 3 —figure supplement 1). These entities were ‘strain’, ‘metagenotype’ and ‘metagenotype annotation’. The complete data model for PHI-Canto is illustrated in Figure 3 ­­­—figure supplements 2 and 3.”

These new supplements are Figure 3 – supplement 2, which illustrates the entity–relationship model for the main Canto database, and Figure 3 – supplement 3, which illustrates the entity–relationship model for a curation session database in Canto.

Point – concepts.

We understand the reviewer’s use of the word ‘concepts’ to be equivalent to our use of the word ‘entities’. The entities that are used in PHI-Canto are included in the Figure 3 supplements as described in ‘Point – entity-relationship models’.

Point – methods.

We note that the annotation process is described at a conceptual level in Figure 4. With regards to the implementation of the process in code, PHI-Canto’s complete source code can be viewed at the Canto repository on GitHub (see the ‘Code Availability section’), where it is made available under the GNU General Public License, version 3 (GPLv3).

Point – eventual algorithms.

With regards to 'eventual algorithms', there is no particular algorithm of note that underlies PHI-Canto's data model, so we did not believe that describing the code that supports PHI-Canto's data model would be relevant. Much of PHI-Canto's code exists simply to validate the data entered by the curator and to store said data in the correct location in the database. The code supporting PHI-Canto's data model is not amenable to description as pseudocode, since it involves multiple source code modules (many from externally developed software libraries) and totals thousands of lines of code.

Point – Information storage and retrieval methods.

Text has been added to the Methods section ‘Changes to the Canto data model and configuration’ providing a general description of how PHI-Canto stores its data.

“PHI-Canto stores its data in a series of relational databases using the SQLite database engine. A primary database stores data shared across all curation sessions, and each curation session also has its own database to store data related to a single publication (such as genes, genotypes, metagenotypes, etc.). PHI-Canto can export its data as a JSON file or more specialized formats, for example the GO Annotation File (GAF) format.”

3. Expand on how data display and interoperability are implemented, for example, to link related information from new and existing publications. Address the possible use of graph representation to link complex information. (See more detailed comments from Reviewers 2 and 3.)

Point – How data display and interoperability are implemented.

The PHI-base version 5 gene centric pages will display all curated information for a gene from multiple publications – both from new and existing publications.

For example

The curated data is made available within the new gene centric version 5 of PHI-base, where on a single gene page all published information from multiple publications, i.e. both new and existing are presented. For interoperability within PHI-base version 5, if the first host target is known for a pathogen gene/protein/other entities, there is a direct link-out to this related gene centric PHI-base page. Similarly, if there is already or will occur in the future a double gene and/or multi-gene functional analysis for either the pathogen or the host or both, then there is/will be a direct link-out(s) to this/these related gene centric PHI-base page(s).

We have amended the text in the Discussion to describe this further

“Here, we have described the development of PHI-Canto to allow the curation of the interspecies pathogen–host interaction literature by professional curators and publication authors. This curated data is then made available on the new gene-centric version 5 of PHI-base, where all information (i.e. new and existing) on a single gene from several publications is presented on a single page, with links to external resources providing information on interacting genes, proteins and other entities.”

Point – Addressing the use of graph representation.

With regards to a graph representation of the data, we are aware of the examples the reviewer described, and we agree that this type of representation could be preferable. However, our data model is currently constrained by the developers of Canto (Rutherford et al., 2014; doi: 10.1093/bioinformatics/btu103), who use a relational data model and currently have no plans to implement a graph data model or a graph representation. We acknowledge that query languages like GraphQL can provide a graph-based interface to an existing relational data model, but we believe this would require a significant technological investment. For PHI-base, we plan to enable a graph representation of the data by integrating with existing knowledge graph tools, such as KnetMiner (www.knetminer.com; doi.org/10.1111/pbi.13583), which will provide graph-based queries on PHI-base (albeit only on select species for which knowledge graphs will be provided, i.e. *Arabidopsis*, rice, wheat, eight plant and human infecting fungal ascomycete pathogens, and two non-pathogenic yeast species). We will also use KnetMiner integration to embed subgraphs of the complete knowledge graph into the gene-centric pages on the PHI-base 5 website.

We have amended the text in our discussion to include a reference to graph-based representation.

“Our future intentions are two-fold: firstly, a graph-based representation of the data will be enabled by integration with knowledge network generation tools, such as Knetminer (Hassani-Pak et al., 2021), where subgraphs of the knowledge graph could be embedded into each gene-centric page on the PHI-base 5 website.”

4. Explain why PHIDO has been generated rather than using existing disease terminologies (such as Mondo or DO).

Point – Why we have developed PHIDO.

We would like to clarify that PHIDO is not intended to compete with existing disease ontologies: it is instead being used as a placeholder, until the time when its terms can be replaced with terms from existing disease ontologies. PHIDO was an expedient solution, in the sense that it provided the fastest way for us to test the process of curating diseases with PHI-Canto. This is because we only had to convert the existing list of disease names already in PHI-base into a controlled vocabulary, thus removing the need to wait for maintainers of other ontologies to add terms for us (as reported in Urban et al., 2022).

Additionally, we were required to use terms from PHIDO due to the lack of representation for plant and animal diseases in existing ontologies or vocabularies. Plant disease, in particular, is very underrepresented, with the ontologies we surveyed having either inappropriate semantics (e.g. the Plant Trait Ontology focusing on traits related to disease, rather than the diseases themselves) or still being in development (e.g. the Plant Stress Ontology). The majority of source ontologies used by MONDO are human-centric, and DO is exclusively for human disease, yet human disease represents only part of the focus of PHI-base (~35%). Furthermore, our choice of vocabularies is limited by the fact that Canto currently only supports ontologies in OBO format (for historical reasons).

We have begun the process of harmonizing disease names in PHI-base with terms from existing disease ontologies – such as MONDO, DO, and the National Cancer Institute Thesaurus – with the ultimate aim of using terms from those ontologies in curation, instead of terms from PHIDO. As general vocabularies for animal and plant disease emerge or are identified, we will extend this procedure to those diseases.

We have also added additional text to the manuscript to make this clearer as recorded below in our detailed response to Reviewer #3 recommendations for the authors.

5. The authors should briefly comment on the possible extension of this approach beyond pathogen-host interactions, which could increase the broader relevance of the study.

Point – Extending PHI-Canto beyond pathogen-host interactions.

We acknowledge the lack of discussion about extending the tool for broader interspecies interactions. These examples may have been omitted from a previous draft due to word count restrictions. We have included additional text in the discussion to suggest some possible extended use cases.

“PHI-Canto, PHI-base and PHIPO were devised and built over the past seven years to serve the research needs of a specific international research community interested in exploring the wide diversity of common and species-specific mechanisms underlying pathogen attack and host defense in plant, animals, humans and other host organisms caused by fungi, protists and bacteria. However, it should be noted that the underlying developments to Canto’s data model – especially the concept of annotating metagenotypes – could be of use to communities focused on different types of interspecies interactions. Possible future uses of the PHI-Canto schema could include insect–plant interactions (both beneficial and detrimental), endosymbiotic relationships such as mycorrhiza–plant rhizosphere interactions, nodulating bacteria–plant rhizosphere interactions, fungi–fungi interactions, plant–plant interactions or bacteria–insect interactions, and non-pathogenic relationships in natural environments such as bulk soil, rhizosphere, phyllosphere, air, freshwater, estuarine water or seawater, and human–animal, animal–bird, human–insect, animal–insect, bird–insect interactions in various anatomical locations (e.g. gut, lung, and skin). The schema could also be extended to situations where phenotype–genotype relations have been established for predator–prey relationships or where there is competition in herbivore–herbivore, predator–predator or prey–prey relationships in the air, on land or in the water. Finally, the schema could be used to explore strain to strain interactions within a species when different biological properties have been noted. Customizing Canto to use other ontologies and controlled vocabularies is as simple as editing a configuration file, as shown in Source code 1.”

Reviewer #1 (Recommendations for the authors):Many readers may wrongly think they will find tones of centralized information about, e.g. their present-day favorite gene. This does not seem to be the case, leading to a related question: where do the contributed curations go and how are they made available? Is there a final control from within the resource team to filter wrong curations due to bad procedures, or even directly fraudulent data treatments?On the model extension side, feeding on new interactions proposed by users: it is not clear what kind of follow-up would be made in order to encompass the usage with appropriate growth and amelioration. Are there plans for this? Also, given that most authors are in the industry, a short statement about conflicts of interest would be desirable. Related to this, it is not clear to the reader if new, useful added curations obtained by a user, will be added to the resource and made publicly available. In an ideal world, the ten examples forming the basis of the resource should grow to thousands. I might be missing something, though.

Points – centralized information, where contributed curations go, how they are made available and how curation is checked.

We believe that the Reviewer #1 has overlooked several key pieces of information regarding the display of curated data within a gene-centric web page in PHI-base version 5, that PHI-Canto curated display is displayed in the freely accessible PHI-base, and that PHI-Canto curated data is checked by a species expert prior to display in PHI-base. More detailed responses to reviewer #1 first point recommendations to authors have already been addressed in the author response to the Essential revisions point 1.

Point – extension of the model beyond pathogen-host interactions.

Regarding the model extensions a section of text has now been added containing some example communities as noted in point 6 of the essential revisions. Implementing these model extensions would be followed up by the individual communities where they would be able to use the freely able Canto source code and load up either newly developed or existing ontologies as required.

Point – conflict of interest.

The following text in the manuscript declares no conflicts of interest

“Ethics declarations, Competing interests.The authors declare no competing interests.”

Point – adding new curations to PHI-base.

The reviewer states that “*it is not clear to the reader if new, useful added curations obtained by a user, will be added to the resource and made publicly available.*” Responses to this point have already been addressed in the author response to the Essential revisions point 1.

In summary the curated data will be added to the publicly available PHI-base. If an author has more data on an interaction, this will only be added to PHI-base once this data has been through the peer review process.

Additionally the reviewer states “In an ideal world, the ten examples forming the basis of the resource should grow to thousands. I might be missing something, though.” It appears that the reviewer has overlooked the text that the ten publications were selected for ‘*trial*’ curation and that we expect many thousands to be curated in the future.

“Ten publications covering a wide range of typical plant, human, and animal pathogen-host interactions were selected for trial curation in PHI-Canto (Table 2).”

To clarify this points in the manuscripts for future readers we have added ‘Trial’ to the section heading of the manuscript which now reads:

“Trial curation of interspecies interaction publications”

and some additional text to describe that we expect curation to expand beyond the initial ten trial publications with publication authors curating their own data.

“Ten publications covering a wide range of typical plant, human, and animal pathogen–host interactions were selected for trial curation in PHI-Canto before the tool was made available to publication authors and communities to add further publications (Table 2).”

Reviewer #2 (Recommendations for the authors):Given my area of expertise, I am not in a position to assess the relevance of the work from a biological point of view. However, I feel that some points relating to data management and problem modeling deserve some comment.In particular, there is no mention of the complete data model of each annotation or the formats in which this information is stored. This omission is probably because this is part of the Canto project. Still, to make this publication self-contained, it would be desirable to include this information or at least a reference to it, especially to measure the changes required.Something that would also improve the work is more detail on how to use or visualize the data generated. Although there are a few brief lines in the section on "Display and interoperability of data", Figure 4 raises the question of how the system will behave in cases where there are already curated publications that refer to the pathogens and hosts of the new publication to be curated. It would be desirable that they are not treated in isolation but that the system allows them to be linked and then navigated in the network resulting from curating a set of publications.Finally, in the introduction to the paper, the authors make the reckless assertion that manual biocuration is the only way to reliably represent information about functions and phenotypes. I would question this assertion given the current state of the art in LNP tools. While it is likely that, in many cases, automated annotation or curation using these techniques will not yield such accurate results, I believe it would be desirable to explore these techniques. It is also possible to think of hybrid human-in-the-loop systems where automatic techniques assist experts and simplify repetitive tasks. I believe that the paper should at least include a discussion of these issues.

Point – data models.

We thank the reviewer for identifying the lack of a mention of a complete data model of each annotation and for the formats in which it is stored. The reviewer assumes correctly that this was omitted due to being part of the Canto project. However, to make this information clearer to our readers we have amended the manuscript and addressed these points in more detail in Essential revision point 3.

Point – data visualization.

The reviewer has queried how data will be visualized and what happens when older and newer publications looking at the same gene interactions are curated. The data will be visualized on gene-centric pages within the new version of PHI-base 5. These gene-centric pages will collate all the information i.e. new and existing from multiple publications into one gene page. Again more detail is provided in response to the Essential revision point 3.

Point – manual techniques compared to automated techniques for curation.

The reviewer has identified our lack of discussion regarding manual techniques compared to automated techniques for curation. We acknowledge that our initial statement in favor of manual curation may have been too bold and have amended the following text in the Introduction section:

“Due to the complexity of the biology, manual biocuration is currently the only way to reliably represent information about function and phenotype in databases and knowledge bases (Wood, Sternberg, & Lipshitz, 2022).”

To the updated text:

“Due to the complexity of the biology and the specificity of the curation requirements, manual biocuration is currently the most reliable way to capture information about function and phenotype in databases and knowledge bases (Wood, Sternberg, & Lipshitz, 2022). For pathogen–host interactions, the original publications do not provide details of specific strains, variants, and their associated genotypes and phenotypes, nor the relative impact on pathogenicity and virulence, in a standardized machine-readable format. The expert curator synergizes knowledge from different representations (text, graphs, images) into clearly defined machine-readable syntax.”

We have also updated the Discussion section with a discussion comparing manual curation to automated curation with regards to PHI-Canto.

“With regards to our focus on manual curation, we recognize that great progress has been made with machine learning (ML) approaches in recent times. However, Wood, Sternberg & Lipshitz (2022) note that the data being curated from publications are “categorical, highly complex, and with hundreds of thousands of heterogeneous classes, often not explicitly labeled”. There are no published examples of ML approaches outperforming an expert curator in accuracy, which is paramount in the medical field. However, curation by experts could provide a highly reliable corpus that could be used for training ML systems. Our aspiration is that ML and expert curators can collaborate in a virtuous cycle whereby expert curators continually review and refine the ML models, while the manual work of finding publications and entity recognition is handled by the ML system.”

Reviewer #3 (Recommendations for the authors):Here are some specific points of confusion, questions, or suggestions for improvement:Page 5 of the manuscript (page 6 of the full pdf):– Lines 103-105 talk about changes in pathogenicity and virulence. It would be useful to readers to have a brief explanation of how these differ from each other and why one only applies to the pathogen while the other can apply to either the host or the pathogen.

Point – Description of pathogenicity and virulence.

This is an observant point. In fact, this information was included in an earlier version of the manuscript, but later omitted to keep the word count down. New text has been added to clarify how pathogenicity and virulence differ from each other.

“Each metagenotype can be annotated with pathogen–host interaction phenotypes to capture changes in pathogenicity (caused by alterations to the pathogen) and changes in virulence (caused by alterations to the host and/or the pathogen). Pathogenicity is a property of the pathogen that describes the ability of the pathogen to cause an infectious disease in another organism. When a pathogenic organism causes disease, the severity of the disease that occurs is referred to as ‘virulence’ and this can also be dependent upon the host organism.”

– Bottom of the page talks about "annotation types". The term "annotation type" seems to be used in a way that allows confusion with the entity being annotated. I believe that the authors intend to say that gene, genotype, and metagenotype are types of biological features (to use their term) that are annotated within PHI-Canto, each with its own set of accompanying annotation types as outlined in Table 1. If that is the correct interpretation, then I suggest modifying the text to make this more explicit with a particular focus on the sentence on lines 111-113.

Point – How we describe different “annotation types”.

We thank the reviewer for this observation and understanding of our intended meaning that gene, genotype, and metagenotype are types of biological features that are annotated within PHI-Canto, each with its own set of accompanying annotation types as outlined in Table 1. We have made changes to Table 1 and the manuscript text to help clarify this. Further details below.

Changes made to Table 1

We have changed the title from

‘Annotation types and selected Annotation extensions used in PHI-Canto’

to

‘Annotation types and selected annotation extensions used in PHI-Canto, grouped by the biological feature being annotated.’

We have changed the Table 1 section headings from

‘Gene annotation type’

to

‘Annotation types for the *gene* biological feature’,

‘Genotype annotation type’

to

‘Annotation types for the *genotype* biological feature’,

and ‘Metagenotype annotation type’

to

‘Annotation types for the *metagenotype* biological feature’.

The following changes in the main text have also been made

“In PHI-Canto, ‘annotation’ is the task of relating a specific piece of knowledge to a biological feature. To curate a wide variety of experiment types, three groupings of annotation types are available in PHI-Canto, covering ‘metagenotype’, ‘genotype’ (of a single species) and ‘gene’ annotation types (Table 1).”

Has been changed to

“In PHI-Canto, ‘annotation’ is the task of relating a specific piece of knowledge to a biological feature. Three types of biological features can be annotated in PHI-Canto: genes, genotypes and metagenotypes. Genotypes can be further specified as pathogen genotypes or host genotypes. Each of these biological features has a corresponding set of annotation types. The relation between biological features, annotation types and the values that can be used for annotation are shown in Table 1.”

Page 6 of the manuscriptLine 114 – "curators use annotation extensions" is referenced to a GOA paper. Perhaps then the sentence should specify that GO annotation curators use annotation extensions or indicate the reference is an example of this practice, using "e.g." perhaps.

Point – Use of GO annotation curators as an example.

We thank the reviewer for this observation and have added further detail to the text as suggested to clarify that GO curators are an example of curators using an annotation extension.

“To capture additional biologically relevant information associated with an annotation, curators use the concept of annotation extensions (which include Gene Ontology annotations described by Huntley et al., 2014) to extend the primary annotation.”

Page 11 of the manuscriptLast paragraph – the text mentions that ECO terms are used to capture evidence. However, the annotation examples in Appendix 1 appear to use a combination of GO evidence codes and terms/phrases that are related to ECO term names. If ECO is being used, why not use the ECO term ids and/or term names across the board?

Point – Different evidence codes are used for GO annotations and phenotype annotations.

GO annotations in Canto are configured to use GO evidence codes. We use ECO (Experiment and Conclusion Ontology) terms to curate the experimental evidence used in making a phenotype annotation and PHI-ECO (PHI-Experimental Conditions Ontology) terms to curate experimental conditions. As we understand it, GO does not permit any evidence codes to be used on its annotations except the GO evidence codes, so terms from ECO would not be applicable for GO annotations. To make this clearer, the following text has been added to Section 1D under the first example GO annotation.

“Note: GO annotations use GO evidence codes (http://geneontology.org/docs/guide-go-evidence-codes/).”

For the phenotype annotation examples the following text has been added to Section 1A

“Note: Phenotype annotations use evidence codes modeled on the Evidence & Conclusion Ontology (ECO). Evidence code ‘Cell growth assay’ corresponds to ‘cell growth assay evidence’ (ECO:0001563).”

Section 1B

“Note: Phenotype annotations use evidence codes modeled on ECO. Evidence code ‘Macroscopic observation (qualitative observation)’ corresponds to the new ECO term ‘qualitative macroscopy evidence’ (ECO:0006342).”

Section 1C

“Note: Phenotype annotations use evidence codes modeled on ECO. Evidence code ‘Microscopy’ corresponds to ‘microscopy evidence’ (ECO:0001098).”

Point – How we have developed and used ECO terms.

With regards to the evidence codes that appear to be related to ECO term names, the discrepancy between the evidence codes used in PHI-Canto and the terms in ECO is due to the fact that, for historical reasons, Canto does not load terms directly from ECO, but rather uses text that is closely aligned to the term names in ECO. In light of this, we have been working to ensure that any new evidence codes added to PHI-Canto also have a corresponding ontology term added to ECO. This is described in the following original text of the manuscript.

“Annotations in PHI-Canto include experimental evidence, which is specified by a term from a subset of the Evidence & Conclusion Ontology (ECO) (Giglio et al., 2019). Experimental evidence codes specific to pathogen-host interaction experiments have been developed and submitted to ECO.”

We have been active in this area and the following ECO terms have now been newly created (or edited) and were recently released by ECO (05_08_2022).

sporulation assay evidence (ECO:0006344)

asexual sporulation assay evidence (ECO:0006345)

sexual sporulation assay evidence (ECO:0006346)

quantitative macroscopy evidence (ECO:0006343)

qualitative macroscopy evidence (ECO:0006342)

host penetration assay evidence (ECO:0006348)

host-surrogate penetration assay evidence (ECO:0006347)

host colonization assay evidence (ECO:0001830) (the comment was edited)

The current images and training materials under development for this manuscript have not yet been updated with these small text change discrepancies between the final term names used by ECO and the initial evidence names used in Canto. A note has been added to the examples of phenotype annotation evidence used Appendix 1 to illustrate more recent updates and slight changes in phrase. Details have been recorded in our response to the ‘Point – Different evidence codes are used for GO annotations and phenotype annotations.’

Page 12 of the manuscriptTop of page – what is the relationship between PHIDO and other existing disease ontologies such as Mondo or DO?

We have described this relationship in detail in Essential revision 4. covered in the ‘Point – why we have developed PHIDO.’ and we have also added further clarification to the manuscript text by altering the original text from

“Diseases are specified by a controlled vocabulary of disease names (called PHIDO), which was derived from disease names curated in previous versions of PHI-base.”

to

“Diseases are specified by a controlled vocabulary of disease names (called PHIDO), which was derived from disease names curated in previous versions of PHI-base (Urban et al., 2022). PHIDO was developed as a placeholder to allow disease names to be annotated on a wide variety of pathogen interactions, including those on plant, human, animal and invertebrate hosts, especially where such diseases were not described in any existing ontology.”

Figure 5The NCBI taxonomy is listed in the databases section, however, it is more of a cv – it's certainly not a database like UniProtKB or PHI-base are. The Evidence and Conclusion Ontology is mentioned in the text but is not in the list of OBO Ontologies. The curated list of strains (line 247, page 11) is not shown as a PHI-base CV, although perhaps the list of strains is stored in the form of the mapping file that is shown in the figure. Is that the case? If so, then perhaps rename that box to make that more clear.

Point – Why we have listed NCBI taxonomy as a database.

We believe that the reviewer is mistaken in this comment and will continue to list NCBI taxonomy in the databases section of Figure 5. We have taken this information from https://www.ncbi.nlm.nih.gov/taxonomy where the landing page states that it is a database. “The Taxonomy Database is a curated classification and nomenclature for all of the organisms in the public sequence databases. This currently represents about 10% of the described species of life on the planet.”

Point – Adding The Evidence and Conclusion Ontology to Figure 5.

We thank the reviewer for spotting this omission and The Evidence and Conclusion Ontology has now been added to the list of OBO ontologies within Figure 5.

Point – Adding the list of strains as a controlled vocabulary to Figure 5.

This is a helpful suggestion and we have added the strain lists to the Controlled Vocabulary box in Figure 5.

Table 1– In the gene section, GO annotation type – are the host species and symbiont species extensions meant to indicate the interacting species? Or the species from which the gene comes? I'm assuming it means the interacting species, but this could be made more explicit.

Point – How the annotation extensions refer to the interacting species.

The reviewer is correct, the extensions indicate the interacting species. A primary GO annotation is made and then an annotation extension can be used to indicate the interacting species, hence ‘with host species’ or ‘with symbiont species’.

– In the genotype section under "single species phenotype" should it not say "(Pathogen phenotype or Host phenotype)" rather than "and"?

Point – How the “single species phenotype” can refer to a “Pathogen phenotype or a Host phenotype”.

We agree with the reviewer that in Table 1, in the genotype section under "single species phenotype" it would be clearer to say "(Pathogen phenotype or Host phenotype)". The text has been changed.

Appendix 1– In general, I find the header/spacing organization made it difficult to follow where one part ended and the next began. Perhaps giving letters to the sections starting "If you have…" such that there would be Section 1A, 1B, etc. might help. Also perhaps the use of some indentation so that separate sections referring to each publication will be easier to see.

Point – Layout of Appendix 1

We thank the reviewer for this helpful suggestion and apologize that the organization of Appendix 1 was difficult to follow. Following the reviewer’s guidance we have added the suggested subsections names to each section within Appendix 1 and also a new contents section just after the opening paragraph in Appendix 1. Here is a list of the Appendix 1 contents

“Contents:

Section 1: Annotation Extensions for curating pathogen-host interaction phenotypes on metagenotypes

Section 1A: If you have a metagenotype phenotype recording ‘unaffected pathogenicity’ (corresponds to footnote ^‡^ in Table 2)

Section 1B: If you have a metagenotype phenotype recording ‘altered pathogenicity or virulence’ (corresponds to footnote ^§^ in Table 2)

Section 1C: If you have a metagenotype phenotype recording ‘mutualism’ (corresponds to footnote ^**^ in Table 2)

Section 1D: If you have a metagenotype phenotype recording ‘a pathogen effector’ (corresponds to footnote ^††^ in Table 2)

Section 2: Annotation Extensions for curating gene-for-gene phenotypes on metagenotypes

Section 2A: If you have a metagenotype phenotype recording ‘a gene-for-gene interaction’ (corresponds to footnote ^‡ ‡^ in Table 2)

Section 2B: If you have a metagenotype phenotype recording ‘an inverse gene-for-gene interaction’ (corresponds to footnote ^¶ ¶^ in Table 2)

Section 3: Annotation Extensions for curating single species phenotypes (pathogen phenotypes or host phenotypes)

Section 3A: Example of an *in vitro* pathogen phenotype (corresponds to footnote ^¶^ in Table 2)

Section 3B: Example of an *in vitro* pathogen chemistry phenotype (corresponds to footnote ^***^ in Table 2)

Section 3C: Example of an *in vivo* host phenotype (corresponds to footnote ^§ §^ in Table 2)”

– Section 1, the section on "If you have a metagenotype phenotype recording "a pathogen effector' (corresponds to footnote 5 in table 2)" – annotations for PMID:31804478: I'm confused why in the gene level GO process annotations that the gene is annotated to the specific child GO:0052034 'effector-mediated suppression of host pattern-triggered immunity', however, when the 'protein binding' and 'enzyme inhibitor activity' GO function annotations are made, the part_of annotation extension is to a grandparent of GO:0052034 that is GO:0140590 'effector-mediated suppression of host defenses. I would have thought that the part_of annotations would have been to GO:0052034. Why is this not the case?

Point – Aligning the part_of GO annotation extension to the primary GO BP annotation

We thank the reviewer for noting this annotation oversight, which may have occurred as new GO terms were being newly developed at the time of curation and PHI-Canto development. The 'protein binding' and 'enzyme inhibitor activity' GO function annotations should also use the part_of annotation extension GO:0052034 'effector-mediated suppression of host pattern-triggered immunity'. These annotations have now been altered in Appendix 1.

Table 1 and Appendix 1 – both refer to the example paper involving mutualism. By definition, in a mutualist relationship, neither partner is a pathogen as disease is not caused. I realize this issue is likely beyond the scope of this paper to discuss, but I wonder about the inclusion of this annotation in the PHI resource since it does not involve a pathogen. Is it because the species has been seen to be a pathogen in other cases? Or is it that PHI includes some non-pathogenic interactions as well? If the first case, it brings to bear how to define something as a pathogen when (as is true for almost all organisms that cause disease in another organism) it only causes disease in some situations and not others (which is of course relevant to the host-pathogen-environment disease triangle mentioned in the text and Figure 1). If it is the second case, might it make sense to think about the scope of the resource as related to its name? Since mechanisms of colonization are often shared between pathogens and beneficial commensals alike, including annotations for symbionts beyond known pathogens would be useful. However, if these are regularly included, more prominent statements to that effect made on the PHI website and in publications would inform users.

Point – Inclusion of a mutualistic interaction to illustrate the extension potential of PHI-Canto to curate alternative interactions.

We thank the reviewer for this observation. We decided to include this publication within the ten trial papers to illustrate that PHI-Canto could be used to curate mutualism relationships. The PHI-base release 4.14 contains curated data for 4847 publications and only two of these publications are for a mutualism interaction. In these two examples there is an interaction shift from mutualism to enhanced antagonism/loss of mutualism (PHI:576 and PHI:578). We were able to successfully curate this shift within PHI-base 4 as a proof of concept idea. Whilst developing PHI-Canto one of these mutualist interaction publications was also included to confirm that curation of this type of interaction was also possible with the new tool. One of the reasons we wish to publish in *eLife* is so that knowledge of the curation tool can be shared and extended to additional communities (see author response to Essential revisions point 6). We have added the following additional text to the footnote ^**^ of Table 2 which links out to the example of curating 'mutualism' that is available in Appendix 1.

^“**^Example of curating 'mutualism' available in Appendix 1. Although ‘mutualism interactions’ are generally out of scope for PHI-base, PHI-Canto can be used to curate these publications if required. In this study, the fungal gene mutation altered the interaction from mutualistic to antagonistic.”